

# Evaluating the value of a network of cosmic-ray probes for improving land surface modelling

Roland Baatz[1,2], Harrie-Jan Hendricks Franssen[1,2], Xujun Han[1,2], Tim Hoar[3], Heye R. Bogena[1] and Harry Vereecken[1,2]

[1]Agrosphere (IBG-3), Forschungszentrum Jülich GmbH, 52425 Jülich, Germany.

[2]HPSC-TerrSys , 52425 Jülich, Germany.

[3]NCAR Data Assimilation Research Section, Boulder, CO, USA.

*Correspondence to*: Roland Baatz (r.baatz@fz-juelich.de)





**Abstract:** Land surface models can model matter and energy fluxes between the land surface and atmosphere, and provide a lower boundary condition to atmospheric circulation models. For these applications, accurate soil moisture quantification is highly desirable but not always possible given limited observations and limited subsurface data accuracy. Cosmic-ray probes (CRPs) offer an interesting alternative to indirectly measure soil moisture and provide an observation that can be assimilated into land surface models for improved soil moisture prediction. Synthetic studies have shown the potential to estimate subsurface parameters of land surface models with the assimilation of CRP observations. In this study, the potential of a network of CRPs for estimating subsurface parameters and improved soil moisture states is tested in a real-world case scenario using the local ensemble transform Kalman filter with the Community Land Model. The potential of the CRP network was tested by assimilating CRP-data for the years 2011 and 2012 (with or without soil hydraulic parameter estimation), followed by the verification year 2013. This was done using (i) the regional soil map as input information for the simulations, and (ii) an erroneous, biased soil map. For the regional soil map, soil moisture characterization was only improved in the assimilation period but not in the verification period. For the biased soil map, soil moisture characterization improved in both periods strongly from a $E_{RMS}$ of 0.11 cm$^3$/cm$^3$ to 0.03 cm$^3$/cm$^3$ (assimilation period) and from 0.12 cm$^3$/cm$^3$ to 0.05 cm$^3$/cm$^3$ (verification period) and the estimated soil hydraulic parameters were after assimilation closer to the ones of the regional soil map. Finally, the value of the CRP network was also evaluated with jackknifing data assimilation experiments. It was found that the CRP network is able to improve soil moisture estimates at locations between the assimilation sites from a $E_{RMS}$ of 0.12 cm$^3$/cm$^3$ to 0.06 cm$^3$/cm$^3$ (verification period), but again only if the initial soil map was biased.

## 1  Introduction

Soil water content (SWC) is a key variable of land surface hydrology and has a strong control on the partitioning of net radiation between latent and sensible heat flux. Knowledge of SWC is relevant for the assessment of plant water stress and agricultural production, as well runoff generation as a response to precipitation events (Vereecken et al., 2008;Robinson et al., 2008). In atmospheric circulation models, SWC is important as a lower boundary condition and is calculated as a state variable in land surface models. Coupling of atmospheric circulation models and land surface models allows quantifying the role of soil moisture on atmospheric processes such as soil moisture-precipitation feedbacks (Koster et al., 2004;Eltahir, 1998) and summer climate variability and drought (Oglesby and Erickson, 1989;Sheffield and Wood, 2008;Seneviratne et al., 2006;Bell et al., 2015). It is therefore important to improve the modelling and prediction of SWC, but this is hampered by model deficiencies and lack of high quality data (Vereecken et al., 2016). Soil moisture measured by space-born remote sensing technologies provides information over large areas (e.g. Temimi et al., 2014). However, space-born remote sensing supplies only information on the upper few centimeters, and data are not reliable for areas with dense vegetation. Therefore, in this paper an alternative source for soil moisture information is explored. Cosmic-ray probes (CRPs) measure fast neutron intensity which allows estimating SWC at an intermediate scale (Zreda et al., 2008;Desilets et al., 2010;Cosh et al., 2016;Lv et al., 2014) which is closer to the desired application scale of land surface models (Ajami et al., 2014;Chen et al.,



2007;Shrestha et al., 2014). Fast neutrons originate from moderation of secondary cosmic particles from outer space by terrestrial atoms. These particles are mainly fast neutrons, which are moderated most effectively by hydrogen because of the similar atomic mass. Therefore, the corresponding neutron intensity measured by CRPs strongly depends on the amount of hydrogen within the CRP footprint, allowing for a continuous non-invasive soil moisture estimate over an area of ~15 ha

(Kohli et al., 2015). The spatial extend of this measurement is desirable as it matches with the desired grid cell size of a land surface model (Crow et al., 2012) and small scale heterogeneities are averaged over a larger area (Franz et al., 2013a;Desilets and Zreda, 2013). Vertical measurement depth ranges from a maximum of ~70 cm under completely dry conditions and decreases to roughly ~12 cm under wet conditions (e.g. 40 vol. % soil moisture) (Bogena et al., 2013). Worldwide several CRP networks exist, like the North American COSMOS network (Zreda et al., 2012), the German CRP network (Baatz et

al., 2014) installed in the context of the TERENO infrastructure measure (Zacharias et al., 2011) and the Australian COSMoZ network (Hawdon et al., 2014).

Soil moisture data assimilation provides a way to improve imperfect land surface model predictions with measured soil moisture data by merging model predictions and data, and can consider the uncertainty of initial conditions, model

parameters and model forcings. Ensemble Kalman Filtering (EnKF) is one of the most commonly applied data assimilation methods (Evensen, 1994;Burgers et al., 1998). Soil moisture data assimilation has been the subject of intensive study for more than a decade now. An early contribution was provided by Houser et al. (1998) who assimilated remotely sensed soil moisture observations from a microwave radiometer into a land atmosphere transfer scheme using four-dimensional variational data assimilation. Rhodin et al. (1999) assimilated soil moisture data for a four-day period in order to obtain an

improved characterization of the lower boundary condition for an atmospheric circulation model. They also used a variational data assimilation approach. More recently, the Ensemble Kalman Filter (Reichle et al., 2002a;Dunne and Entekhabi, 2005;Crow, 2003), the Extended Kalman Filter (Draper et al., 2009;Reichle et al., 2002b), four-dimensional variational methods (Hurkmans et al., 2006) and the Local Ensemble Transform Kalman Filter (Han et al., 2015;Han et al., 2013) were applied for updating soil moisture states in land surface models. Reichle et al. (2002a) performed a synthetic

experiment using L-band microwave observations of the Southern Great Plains Hydrology Experiment (Jackson et al., 1999) to analyze the effect of ensemble size and prediction error. Dunne and Entekhabi (2005) showed that an Ensemble Kalman Smoother approach, where data from multiple time steps was assimilated to update current and past states, can yield a reduced prediction error compared to a pure filtering approach.

More recent work addressed joint state-parameter estimation in hydrologic land surface models with data assimilation methods. Joint state-parameter estimation with EnKF is possible by an augmented state vector approach (Chen and Zhang, 2006), a dual approach (Moradkhani et al., 2005) or an approach with an additional external optimization loop (Vrugt et al., 2005). Pauwels et al. (2009) optimized soil hydraulic parameters of a land surface model with synthetic aperture radar data. Lee (2014) used Synthetic Aperture Radar soil moisture data to estimate soil hydraulic properties at the Tibetan plateau



using the EnKF and a Soil Vegetation Atmosphere Transfer model. Bateni and Entekhabi (2012) assimilated land surface temperature with an Ensemble Kalman Smoother and achieved a better estimate of the partitioning of energy between sensible and latent heat fluxes. Han et al. (2014b) updated soil hydraulic parameters of the Community Land Model (CLM) by assimilation of synthetic brightness temperature data with the Local Ensemble Transform Kalman Filter (LETKF) (Hunt

et al., 2007). Shi et al. (2014) used the Ensemble Kalman Filter for a synthetic multivariate data assimilation problem with a land surface model and then applied it to real data (Shi et al., 2015). The cases illustrated a way to use real world data for estimating several parameters in hydrologic land models. (Kurtz et al., 2016) developed a particular CPU-efficient data assimilation framework for the coupled land surface-subsurface model TerrSysMP (Shrestha et al., 2014). They successfully updated $2 \times 10^7$ states and parameters in a synthetic experiment. Whereas these studies were made with land surface models,

also in soil hydrological applications recently data assimilation was used to estimate soil hydraulic parameters. Early work was by Wu and Margulis (2011, 2013) in the context of real-time control of waste water reuse in irrigation. Erdal et al. (2014) investigated the role of bias in the conceptual soil model and explored bias aware EnKF as a way to deal with it. Erdal et al. (2015) focused on handling of strong non-Gaussianity of the state variable in EnKF under very dry conditions. Montzka et al. (2013;2011) explored the role of the particle filter for handling non-Gaussianity in soil hydrology data

assimilation. They showed that the ability of a data assimilation system to correct the soil moisture state and estimate hydraulic parameters strongly depends on the nonlinear character of the soil moisture retention characteristic. Song et al. (2014) worked on a modified iterative filter to handle the non-linearity and non-Gaussianity of data assimilation for the vadose zone. For a further literature review on data assimilation in the context of hydrological and land surface models we refer to Reichle (2008) and Montzka et al. (2012).

Shuttleworth et al. (2013) developed the Cosmic Ray Soil Moisture Interaction Code (COSMIC), which is a forward operator to be applied for assimilating neutron intensity observations from CRPs. The COSMIC code was evaluated for several sites (Baatz et al., 2014;Rosolem et al., 2014). Its capability to propagate surface soil moisture information into the deeper soil column was analyzed by Rosolem et al. (2014). The COSMIC operator was successfully implemented in the Data

Assimilation Research Testbed (Rosolem et al., 2014) to allow for state updating by the Ensemble Adjustment Kalman Filter (Anderson, 2001). The COSMIC operator was implemented in a python interface that couples the land surface model CLM and the LETKF for joint state parameter updating (Han et al., 2015). Neutron counts measured by CRP have been used in data assimilation studies to update model states (Han et al., 2015;Rosolem et al., 2014). Soil hydraulic parameters were also updated by assimilation of neutron counts (Han et al., 2016), but only for a synthetic study which showed its feasibility.

CRPs were also used for inverse estimation of soil hydraulic parameters of the Hydrus-1D model (Villarreyes et al., 2014).

This work further explores the value of measured neutron intensity by CRPs to improve modelling of terrestrial systems at the catchment scale (Simmer et al., 2015) using a land surface model. Compared to existing work the main novelties are:





(i) Data from a network of nine CRPs were assimilated in the Community Land Model version 4.5 (CLM) with an evaluation of the information gain by this assimilation at the larger catchment scale. Until now evaluations with CRPs were made for a single location, but not for a complete network of CRPs. It is a very important question whether CRPs can also improve the soil moisture characterization at the larger catchment scale and how dense the CRP network should be. The high variability of soil moisture at a short distance could potentially limit the CRP measurement value and make updating of soil moisture contents further away from the sensor meaningless. On the other hand, soil maps and atmospheric forcings show spatial correlations over larger distances which suggests that CRP measurements potentially carry important information to update soil moisture contents for larger regions. If it is found that CRP networks with a density like in this study (10 stations per 2354 km$^2$) can improve soil moisture content characterization at the larger catchment scale, this is of high relevance and importance for agricultural applications, flood prediction and protection, and regional weather prediction (Whan et al., 2015;Koster et al., 2004;Seneviratne et al., 2010). The main research question addressed in this paper is therefore whether a CRP network of the density in this study can improve large scale soil moisture characterization by state and parameter updates.

(ii) Soil hydraulic parameters were updated together with the soil moisture states in a real-world case study at the larger catchment scale. The study in this paper also allows some evaluation of the feasibility of the updated large scale soil hydraulic parameters.

In the following paragraphs are presented the model site and the measurements (2.1), the Community land Model and its parameterization (2.2), the COMIC forward model (2.3) and the data assimilation procedure (2.4).

## 2        Materials and methods

### 2.1        Site description and measurements

The model domain, the Rur catchment (2354 km$^2$), is situated in western Germany and illustrated in Fig. 1. Most prominent vegetation types are agricultural land use (mainly in the North), grassland, and coniferous and deciduous forest. The altitude varies between 15 m a.s.l. in the flat northern part and 690 m a.s.l. in the hilly southern part. Precipitation, evapotranspiration and land use follow the topography. Annual precipitation ranges between less than 600 mm in the North to 1200 mm in the hilly South (Montzka et al., 2008). Annual potential evapotranspiration varies between 500 mm in the South and 700 mm in the North (Bogena et al., 2005). The Rur catchment CRP network comprises nine CRPs (CRS1000, HydroInnova LLC, 2009) which were installed in 2011 and 2012 (Baatz et al., 2014). Climate and soil texture of the CRP sites can be found in Table 1. The CRPs were calibrated in the field using gravimetric soil samples. At each site, 18 soil samples were taken along three circles with distances of 25, 75 and 175 meters from the CRP, six samples evenly distributed along each circle. Each sample was extracted with a 50.8 x 300 mm round HUMAX soil corer (Martin Burch AG, Switzerland). The samples were split into 6 sub-samples with 5 cm length each and oven dried at 105 °C for 48 hours to measure dry soil bulk density and soil moisture. Lattice water was determined for each site using a heat conductivity detector. Soil bulk density, soil moisture,





lattice water and 12 hour averaged measured neutron intensity were used to determine calibration parameters specific for each CRP and the COSMIC operator.

## 2.2 Community land model and parameterization

The Community Land Model version 4.5 (CLM) was the land surface model of choice for simulating water and energy exchange between the land surface and the atmosphere (Oleson et al., 2013). Some of the key processes which are solved by CLM are radiative transfer in the canopy space, interception of precipitation by the vegetation and evaporation from intercepted water, water uptake by vegetation and transpiration, soil evaporation, photosynthesis, as well as water and energy flow in the subsurface. SWC in CLM is influenced by precipitation, infiltration into the soil, water uptake by vegetation,

surface evaporation and surface and subsurface runoff. Oleson et al. (2013) provide further details on CLM4.5. To limit the scope and complexity of this study, CLM was run using satellite phenology, e.g. prescribed leaf area index data and the biogeochemical module turned off.

The spatial domain is discretized by rectangular grid cells by CLM. Each grid cell may have several types of land cover:

Lake, urban, vegetated, wetland, and glacier. The vegetated part of the grid cell can be covered by several plant functional types which are all linked to a single soil column. The soil column is vertically discretized by ten soil layers and five bedrock layers. Layer thickness increases exponentially from 0.007 m at the surface to 2.86 m for layer 10. Vertical water flow in soils is modelled by the 1D Richards equation. Soil hydraulic parameters are determined from sand and clay content using pedotransfer functions for the mineral soil fraction (Clapp and Hornberger, 1978;Cosby et al., 1984), and organic matter

content for the organic soil fraction (Lawrence and Slater, 2008).

The following equations describe how soil texture and organic matter define the soil hydraulic properties in CLM such as porosity, hydraulic conductivity, the empirical exponent $B$ and soil matric potential. Hydraulic conductivity ($k[z]$ in mm/s) at the depth z between two layers ($i$ and $i+1$) is a function of soil moisture ($\theta$ in m³/m³ in layers $i$ and $i+1$), saturated

hydraulic conductivity ($k_{sat}$ in mm/s at $z$), saturated soil moisture ($\theta_{sat}$ in m³/m³) and the empirical exponent $B$:

$$k[z] = \begin{cases} \phi_{ice}k_{sat,z}\left[\frac{0.5(\theta_i+\theta_{i+1})}{0.5(\theta_{sat,i}+\theta_{sat,i+1})}\right]^{2B_i+3} & 1 \leq i \leq N_{levsoi}-1 \\ \phi_{ice}k_{sat,z}\left(\frac{\theta_i}{\theta_{sat,i}}\right)^{2B_i+3} & i = N_{levsoi} \end{cases} \qquad 1$$

where $\phi_{ice}$ is the ice impedance factor. The ice impedance factor was implemented to simplify an increased tortuosity of water flow in a partly frozen pore space. It is calculated with $\phi_{ice} = 10^{-\Omega F_{ice}}$ using the resistance factor $\Omega = 6$ and the frozen fraction of soil porosity $F_{ice} = \theta_{ice}/\theta_{sat,i}$. Soil hydraulic properties are calculated separately for the mineral (*min*) and organic matter (*om*) soil components. Total porosity $\theta_{sat,i}$ is calculated using the fraction of organic matter ($f_{om,i}$) with:




$$\theta_{sat,i} = (1 - f_{om,i})\theta_{sat,min,i} + f_{om,i}\theta_{sat,om} \qquad\qquad 2$$

where the organic matter porosity is $\theta_{sat,om} = 0.9$ and sand content in % determines the mineral soil porosity $\theta_{sat,min}$ as:

$$\theta_{sat,min} = 0.489 - 0.00126 \times \%sand \qquad\qquad 3$$

Analogous, the exponent $B$ is calculated with

$$B_i = (1 - f_{om,i})B_{min,i} + f_{om,i}B_{om} \qquad\qquad 4$$

Where $B_{om} = 2.7$ is the organic exponent and the mineral exponent $B_{min,i}$ is determined by clay content in % with:

$$B_{min,i} = 2.91 + 0.159 \times \%clay \qquad\qquad 5$$

Saturated hydraulic conductivity is calculated for a connected and an unconnected fraction of the grid cell with:

$$k_{sat}[z_i] = (1 - f_{perc})k_{sat,uncon}[z_i] + f_{perc,i}k_{sat,om}[z_i] \qquad\qquad 6$$

5   where $f_{perc,i}$ is the fraction of a grid cell where water flows with saturated hydraulic conductivity of the organic matter ($k_{sat,om}[z_i]$ in mm/s) through the organic material only, the so called connected flow pathway, whereas the saturated hydraulic conductivity of the unconnected part ($k_{sat,uncon}[z_i]$ in mm/s) depends on organic and mineral saturated soil hydraulic conductivity:

$$k_{sat,uncon} = (1 - f_{perc})\left(\frac{1 - f_{om}}{k_{sat,min}} + \frac{f_{om} - f_{perc}}{k_{sat,om}}\right)^{-1} \qquad\qquad 7$$

where saturated hydraulic conductivity for mineral soil is calculated from the grid cell sand content as:

$$k_{sat,min}[z_i] = 0.0070556 \times 10^{-0.884 + 0.0153 \times \%sand} \qquad\qquad 8$$

10   The fraction $f_{perc}$ is calculated with:

$$f_{perc} = 0.908 \times (f_{om} - 0.5)^{0.139} \qquad f_{om} \geq 0.5 \qquad\qquad 9$$

$$f_{perc} = 0 \qquad\qquad\qquad\qquad f_{om} < 0.5 \qquad\qquad 10$$

Soil matric potential (mm) is defined as function of saturated soil matric potential (mm) with:

$$\psi_i = \psi_{sat,i}\left(\frac{\theta_i}{\theta_{sat,i}}\right)^{-B_i} = \left[(1 - f_{om,i})\psi_{sat,min,i} + f_{om,i}\psi_{sat,om}\right]\left(\frac{\theta_i}{\theta_{sat,i}}\right)^{-B_i} \qquad\qquad 11$$

where saturated organic matter matric potential is $\psi_{sat,om} = -10.3\ mm$ and saturated mineral soil matric potential is calculated from sand content as:

$$\psi_{sat,min,i} = -10.0 \times 10^{1.88 - 0.0131 \times \%sand} \qquad\qquad 12$$





### 2.3    Cosmic-ray forward model

SWC retrievals were calculated from neutron intensity observations with the Cosmic-ray Soil Moisture Interaction Code COSMIC (Shuttleworth et al., 2013) following calibration results and the procedure of Baatz et al. (2014). COSMIC parameterizes interactions between neutrons and atoms in the subsurface, relevant for soil moisture estimation. COSMIC

was calibrated against the more complex Monte Carlo Neutron Particle model MCNPx (Pelowitz, 2005) and needs considerably less CPU-time than the MCNPx model. The reduced CPU-time need and the physically based parameterization make COSMIC a suitable data assimilation operator. The code was tested at multiple sites for soil moisture determination (Baatz et al., 2014;Rosolem et al., 2014) and analyzed in detail by Rosolem et al. (2014).

COSMIC assumes that a number of high energy neutrons enter the soil. In the soil, high energy neutrons are reduced by interaction with the soil leading to isotropic generation of fast neutrons with less energy in each soil layer. Before resurfacing, fast neutrons are reduced again by soil interaction (Shuttleworth et al., 2013). The number of neutrons $N_{CRP}$ that reaches the CRP can be summarized in a single integral as

$$N_{CRP} = N_{COSMIC} \int_0^\infty \left\{ A(z)[\alpha \rho_S(z) + \rho_w(z)] exp\left(-\left[\frac{m_s(z)}{L_1} + \frac{m_w(z)}{L_2}\right]\right)\right\} \cdot dz \qquad 13$$

where $N_{COSMIC}$ is an empirical coefficient that is CRP specific and needs to be estimated by calibration, $A(z)$ is the

integrated average attenuation of fast neutrons, $\alpha = 0.404 - 0.101 \times \rho_S$ is the site specific empirical coefficient for the creation of fast neutrons by soil, $\rho_S$ is the dry soil bulk density in g/cm$^3$, $\rho_w$ is the total soil water density in g/cm$^3$, $m_s$ and $m_w$ are the mass of soil and water, respectively, per area in g/cm$^2$. $L_1 = 162.0$ g cm$^{-2}$ and $L_2 = 129.1$ g cm$^{-2}$ are empirical coefficients that were estimated using the MCNPx code (Shuttleworth et al., 2013). The integrated average attenuation of fast neutrons $A(z)$ can be found numerically by solving

$$A(z) = \left(\frac{2}{\pi}\right)\int_0^{\pi/2} exp\left(\frac{-1}{cos(\theta)}\left[\frac{m_s(z)}{L_3} + \frac{m_w(z)}{L_4}\right]\right) \cdot d\theta \qquad 14$$

where $\theta$ is the angle along a vertical line below the CRP detector to the element that contributes to the attenuation of fast neutrons, $L_3 = -31.65 + 99.29 \times \rho_S$ is determined from soil bulk density and $L_4 = 3.16$ g cm$^{-2}$ is another empirical coefficient estimated using the MCNPx code (Shuttleworth et al., 2013). The COSMIC operator is discretized into 300 vertical layers of one cm thickness up to a depth of three meters. For each CLM grid cell in the model domain, simulated SWC is used to generate a SWC retrieval using the COSMIC code. Simulated SWC is handed from the CLM simulation

history files to the COSMIC operator. Given the vertical SWC distribution of the individual CLM soil column, COSMIC internally calculates the contribution of each layer to the simulated neutron intensity signal at the COSMIC soil surface. In this study, the contribution of each layer was used to calculate the weighted CLM SWC retrieval corresponding to the vertical distribution of simulated SWC in each grid cell. Measured neutron intensity of CRPs was used to inversely





determine a CRP SWC retrieval as by Baatz et al. (2014) assuming a homogeneous vertical SWC distribution. Then, the weighted CLM SWC retrieval is used in the data assimilation scheme to relate the CRP SWC retrieval to the model state.

## 2.4    Data assimilation

This study uses the local ensemble transform Kalman filter (LETKF) (Hunt et al., 2007) to assimilate SWC retrievals by CRPs into the land surface model CLM. Besides other Ensemble Kalman Filter variants, the LETKF is applied in atmospheric sciences (Liu et al., 2012;Miyoshi and Kunii, 2012), ocean science (Penny et al., 2013) and also in land surface hydrology (Han et al., 2014a;Han et al., 2015). Updates were calculated either for states or jointly for states and parameters. For state updates only, the LETKF was used as proposed by Hunt et al. (2007). Calculations were made for an ensemble of model simulations which differed related to variations in model forcings and input parameters. The states of the different ensemble members are indicated by $\mathbf{x}_i^f$ where $i$=1, …., $N$ and $N$ is the number of ensemble members. The individual state vectors $\mathbf{x}_i^f$ contain the CLM-simulated SWC of the ten soil layers and the vertically weighted SWC retrieval obtained with the COSMIC operator. For each grid cell, a vector $\mathbf{X}^f$ can be constructed which contains the deviations of the simulated states with respect to the ensemble mean $\bar{\mathbf{x}}^f$ :

$$\mathbf{X}^f = \left[\mathbf{x}_1^f - \bar{\mathbf{x}}^f, …, \mathbf{x}_N^f - \bar{\mathbf{x}}^f\right] \qquad 15$$

In case of joint state-parameter updates, a state augmentation approach was followed (Hendricks Franssen and Kinzelbach, 2008;Han et al., 2014b). In this case, the augmented model state vector $\mathbf{X}^f$ is constructed from the weighted SWC, and the grid cell's sand, clay and organic matter content.

In order to relate the measured neutron intensity with the simulated SWC of CLM, the observation operator $\mathbf{H}$ (COSMIC) is applied on the measured neutron intensity in order to obtain the expected weighted SWC retrieval for each of the stochastic realizations:

$$\mathbf{y}_i^f = \mathbf{H}\left(\mathbf{x}_i^f\right) \qquad 16$$

The ensemble realizations of the modelled SWC retrieval $\mathbf{y}_1^f$ to $\mathbf{y}_N^f$ with respect to the ensemble mean $\bar{\mathbf{y}}^f$ are stored in the vector $\mathbf{Y}^f$:

$$\mathbf{Y}^f = \left[\mathbf{y}_1^f - \bar{\mathbf{y}}^f, …, \mathbf{y}_N^f - \bar{\mathbf{y}}^f\right] \qquad 17$$

The observation error correlation was reduced in space by the factor $f_{red}$ using the spherical model:

$$f_{red} = 1 - (1.5 \times d/d_{max}) + (0.5 \times [d/d_{max}]^3) \qquad 18$$

where $d$ is the distance to the observation and $d_{max} = 40km$ is the maximum observation correlation length, about half the size of the catchment. Only SWC retrievals within the maximum observation correlation length were used for assimilation.





This leads to a 'localized' size of $\mathbf{Y}^f$ and the observation error covariance matrix $\mathbf{R}$. The intermediate covariance matrix $\mathbf{P}^a$ (also called analysis error covariance matrix) is calculated according to:

$$\mathbf{P}^a = [(N-1)\mathbf{I} + \mathbf{Y}^{fT}\mathbf{R}^{-1}\mathbf{Y}^f] \qquad\qquad 19$$

In addition, the mean weight vector $\overline{\mathbf{w}}^a$ is obtained as follows:

$$\overline{\mathbf{w}}^a = \mathbf{P}^a\mathbf{Y}^{fT}\mathbf{R}^{-1}(\mathbf{y}^0 - \overline{\mathbf{y}}^f) \qquad\qquad 20$$

where $\mathbf{y}^0$ is CRP SWC retrieval. In the ensemble space, a perturbation matrix $\mathbf{W}^a$ is calculated from the symmetric square root of $\mathbf{P}^a$:

$$\mathbf{W}^a = [(N-1)\mathbf{P}^a]^{1/2} \qquad\qquad 21$$

The final analysis $\mathbf{X}^a$ is obtained from:

$$\mathbf{X}^a = \overline{\mathbf{x}}^f + \mathbf{X}^f[\overline{\mathbf{w}}^a + \mathbf{W}^a] \qquad\qquad 22$$

A more detailed description of the LETKF can be found in (Hunt et al., 2007) and details on the implementation of the LETKF in combination with CLM are given by (Han et al., 2015).

## 3 Model and Experiment Setup

### 3.1 Model Setup

In this study, discretization and parameterization of the hydrological catchment was done on the basis of high resolution data. The Rur catchment domain is spatially discretized by rectangular grid cells of 0.008 degree size (~750 m). The model time step was set to hourly. Land cover was assumed to consist of vegetated land units only, and a single plant functional type (PFT) for each grid cell was defined. The plant functional types were derived from a remotely sensed land use map using RapidEye and ASTER data with 15 m resolution (Waldhoff, 2012). Sand content, clay content and organic matter content were derived from the high resolution regional soil map BK50 (Geologischer Dienst Nordrhein-Westfalen, 2009). The BK50 soil map provides the high resolution soil texture for the catchment and is the most detailed soil map available for the defined region. As an alternative, simulations were also performed for a biased soil texture distribution with a fixed sand content of 80 % and clay content of 10 % (S80 soil map). This represents a large error with respect to the expected true soil properties. It allows evaluating the joint state-parameter estimation approach because given the expected bias, we can evaluate whether and to what extend the soil properties are modified by the data assimilation to be closer to the available high resolution soil map. In addition, in many regions across the Earth a high resolution soil map is not available and land surface models which are applied for those regions, for example in the context of global simulations, might be strongly affected by the error in soil properties. It was tested how this impacted the simulation results. Maximum saturated fraction, a surface parameter which is used for runoff generation, was calculated from a 10 meter digital elevation model (scilands GmbH, 2010). Leaf area index data were derived from monthly averaged Moderate Resolution Imaging Spectrometer data



(MODIS). CLM was supplied with hourly atmospheric forcing data from a reanalysis data set for the years 2010 to 2013 from the German Weather Service (DWD). The data was downscaled from a resolution of 2.8 km$^2$ to the CLM resolution using linear interpolation based on Delaunay triangulation. Forcing data include precipitation in mm/s, incident solar and longwave radiation in W/m$^2$, air temperature in K, air pressure in hPa, wind speed in m/s and relative humidity in kg/kg at the lowest atmospheric level.

### 3.2 Model ensemble

Uncertainty was introduced into the regional CLM model by perturbed soil parameters and forcings. Contents of sand, clay and organic matter were perturbed with spatially correlated noise from a uniform sampling distribution with mean zero and standard deviation 10 % or 30 % (Han et al., 2015). By perturbing texture, soil parameters are also perturbed through the pedotransfer functions used in CLM as specified in Sect. 2.2. Precipitation ($\sigma = 0.5$ or 1.0; lognormal distribution) and shortwave radiation ($\sigma = 0.3$; lognormal distribution) were perturbed with multiplicative noise with mean equal to one. Longwave radiation ($\sigma = 20$ W m$^{-2}$) and air temperature ($\sigma = 1K$) were perturbed with additive noise. The forcing perturbations were imposed with correlations in space (5 km) using a fast Fourier transform. Correlation in time was introduced with an AR(1)-model with autoregressive parameter=0.33. These correlations and standard deviations were chosen based on previous data assimilation experiments (Reichle et al., 2010;Kumar et al., 2012;De Lannoy et al., 2012;Han et al., 2015). In this work, only results for precipitation perturbation with $\sigma = 0.5$ will be shown as results for $\sigma = 1.0$ were very similar. An ensemble size of 95 realizations was used in the simulations. Based on previous work (Baatz et al., 2015), the SWC retrieval uncertainty for CRPs was estimated to be 0.03 cm$^3$/cm$^3$.

### 3.3 Experiment set-up

All simulation experiments in this study used initial conditions from a single five year spin-up run. For the five year spin-up run, a single forcing data set of the year 2010 was repeatedly used as atmospheric input. The soil moisture regime became stable after five years spin-up period, and additional spin-up simulations would not affect soil moisture in the consecutive years. After this five year spin-up, soil parameters and forcing data of the consecutive years were perturbed. From 1$^{st}$ Jan. 2011 onwards, CLM was propagated forward with an ensemble of 95 realizations. On 20$^{th}$ Mar. 2011, the first SWC retrieval was assimilated and assimilation of SWC retrievals continued until 31$^{st}$ Dec. 2012. From 1$^{st}$ January 2013 to 31$^{st}$ December 2013 the model was propagated forward without data assimilation but with an ensemble of 95 realizations. The year 2013 was used exclusively as evaluation period for data assimilation experiments.

In total, 26 simulation experiments were carried out using different setups (Table 2). Two open loop simulations were run for the BK50 soil map (OL-BK50) and the S80 soil map (OL-S80), respectively, without data assimilation and soil parameter perturbation of 30 %. These simulations are referred to as reference runs for the respective soil map. Simulation results of data assimilation runs were compared to the reference runs for quantification of data assimilation benefits. Simulations were




done with joint state-parameter estimation (PAR-), two for the S80 soil map (PAR-S80-) and two for the BK50 soil map (PAR-BK50-), for which soil texture was perturbed by 10 % and 30 %. Two simulations were done with state updates only for the BK50 soil map (Stt-BK50) and the S80 soil map (Stt-BK50), where soil texture was perturbed by 30 %. These eight simulations form the basic set of experiments.

Besides the data assimilation experiments also a larger number of jackknifing simulations was run to evaluate the data assimilation performance. These simulations allow evaluating the impact of the CRP network to improve SWC characterization at other locations, without CRP. In a jackknife experiment, data from eight CRP locations were assimilated and one CRP was excluded from the assimilation for evaluation purpose. At the evaluation location, simulated SWC (which is affected by the assimilation of the other eight probes) was compared to CRP SWC retrievals. For jackknife simulations, the perturbation of soil texture was set to 30 % and precipitation perturbation was done with $\sigma = 0.5$. States and parameters at these sites were jointly updated, and simulations were made using the BK50 and the S80 soil maps as input. Therefore, a total of 18 jackknife simulations (jk-S80-* and jk-BK50-*) was performed (two soil maps times nine different simulations leaving away one CRP at a time).

Simulation results were evaluated with the root mean square error ($E_{RMS}$):

$$RMSE = \sqrt{\frac{\sum_{t=1}^{n}\left(SWC_{t,CLM} - SWC_{t,CRP}\right)^2}{n}}$$

where $n$ is the total number of time steps, $SWC_{t,CLM}$ is the CLM SWC retrieval at time step $t$ and $SWC_{t,CRP}$ is the CRP SWC retrieval at time step $t$. In case SWC was assimilated at the corresponding time step, $SWC_{t,CLM}$ is SWC prior to assimilation. In the case the $E_{RMS}$ is estimated at a single point in time over all CRPs available, the number of time steps $n$ can be replaced by the number of CRPs available. The second evaluation measurement in this study is the bias:

$$bias = \frac{\sum_{t=1}^{n}\left(SWC_{t,CLM} - SWC_{t,CRP}\right)}{n}$$

## 4 Results

### 4.1 General Results

Table 3 summarizes the performance statistics in terms of $E_{RMS}$ for the assimilation period (2011 and 2012). Presented are results for the open loop scenario and six data assimilation scenarios. Errors of open loop simulations are higher for the S80-simulation than for the BK50-simulation at all sites but Merzenhausen. At Merzenhausen $E_{RMS}$ was 0.054 cm$^3$/cm$^3$ for the S80 soil map and 0.067 cm$^3$/cm$^3$ for the BK50 soil map. Open loop simulations with the S80 soil map resulted in $E_{RMS}$-values above 0.10 cm$^3$/cm$^3$ at five of nine sites. At all sites data assimilation results with the S80 soil map improved SWC compared to the open loop simulations. This was also the case for the BK50 soil map simulations at all sites but Aachen



where $E_{RMS}$ was larger than for the open loop run. In general, data assimilation improved simulations more for the S80 soil map ($E_{RMS}$ reduced by 0.079 cm³/cm³) than for the BK50 soil map ($E_{RMS}$ reduced by 0.01 cm³/cm³). Room for improvement with the BK50 soil map runs was more limited because of the smaller open loop errors. Nevertheless, after state updating alone the BK50 soil map still gave smaller errors than the S80 soil map. However, joint state-parameter estimation further improved simulation results by reducing $E_{RMS}$-values and the parameter updating resulted in similar $E_{RMS}$-values for the BK50 (0.028 cm³/cm³) and S80 soil map (0.03 cm³/cm³). The $E_{RMS}$ for simulations with 10 % and 30 % perturbation of soil texture values did not show very different results.

The temporal course of soil moisture in 2011 at the two sites Gevenich and Merzenhausen is shown in Fig. 2. The figures illustrates that SWC at both sites was lower with the S80 soil map than with the BK50 soil map. In Gevenich and Merzenhausen, mean open loop SWC in 2011 was 0.17 cm³/cm³ for the S80 soil map at both sites and 0.27 cm³/cm³ for the BK50 soil map at both sites. CRP measurements at Merzenhausen started in May 2011. In the data assimilation runs SWC was immediately affected at the Merzenhausen and Gevenich sites as soon as Merzenhausen CRP SWC retrievals were assimilated. The simulated SWC for the PAR-S80-30 data assimilation run increased as compared to the S80 open loop simulation. The first observation at Gevenich was recorded on July 7th, 2011. By that date, the simulated CLM SWC retrieval was already close to the CRP SWC retrieval at the Gevenich site (Fig. 2) due to SWC updates which showed to have a beneficial impact. Fig. 2 also shows that the BK50 open loop run was close to the observed SWC at both sites, even without data assimilation.

Fig. 3 shows the temporal course of SWC from January 2011 to December 2013 at Heinsberg and Wildenrath. Assimilation and evaluation results are shown for the case of joint state-parameter updates (PAR-S80-30), only state updates (Stt-S80), open loop (OL-S80) and CRP SWC retrievals. At Heinsberg, results show that assimilated SWC was closer to the CRP SWC retrieval when both states and parameters were updated (PAR-S80-30) than if only states were updated (Stt-S80). This is the case in the assimilation period and in the evaluation period. At the beginning of the evaluation period, the Stt-S80 simulation shows an increase in bias between modeled CLM SWC retrievals and CRP SWC retrieval within the first few days of 2013. The bias of Stt-S80 remained throughout the evaluation period. In contrast, modeled SWC during the evaluation period was close to the CRP SWC retrieval if parameters were previously updated (PAR-S80-30).

The CRP at Wildenrath started operating on May 7th, 2012. SWC retrievals at other CRPs were assimilated already from May 2011 onwards and affected SWC at Wildenrath (Fig. 3). Until May 2012, Fig. 3 shows assimilated SWC (Stt-S80 and PAR-S80-30) was higher than open loop SWC. However, no SWC retrievals were available at the Wildenrath site for comparison during this period. When SWC retrievals from the CRP at Wildenrath became available and were assimilated into the model, assimilated (Stt-S80 and PAR-S80-30) and open loop (OL-S80) SWC were close to CRP SWC retrievals. This was the case throughout the remaining assimilation and evaluation period. These results suggest that the high sand



content of the biased soil map is not far from the optimal sand content at Wildenrath. Therefore, at Wildenrath, the high sand content of both soil maps (60 % and 80 %) resulted in good modeling results already for the open loop runs. This suggests that before May 2012, simulated SWC of the open loop runs with either soil map represented more realistic SWC than assimilated SWC during this period. This will be discussed further in the discussion section.

## 4.2     Verification period

The year 2013 was the verification year without data assimilation. $E_{RMS}$ values for the evaluation period 2013 are reported in Table 4. On the one hand, BK50 data assimilation runs with joint state-parameter estimation resulted in improved SWC at three out of the nine sites compared to open loop BK50-runs. For the other six sites results worsened compared to the
corresponding BK50 open loop run. $E_{RMS}$ values increased from an average of 0.041 cm$^3$/cm$^3$ (OL-BK50) over all sites to 0.047 cm$^3$/cm$^3$ (PAR-BK50-30). On the other hand, for the S80 soil map, all sites except Wildenrath had significantly reduced $E_{RMS}$ values for the case of data assimilation including parameter updating compared to the S80 open loop run. For the S80 simulations, average $E_{RMS}$ over all sites for 2013 was on average 0.12 cm$^3$/cm$^3$ for the open loop run and 0.04 cm$^3$/cm$^3$ for the run including data assimilation. In case only states were updated (Stt-S80 and Stt-BK50), $E_{RMS}$ was also
slightly reduced (compared to open loop runs) for the majority of sites during the evaluation period in 2013. On average, this reduction was 0.016 cm$^3$/cm$^3$ for the S80 soil map (Stt-S80) and 0.002 cm$^3$/cm$^3$ for the BK50 soil map (Stt-BK50). At sites, where $E_{RMS}$ was larger for data assimilation runs with state updating (compared to open loop runs), the increase was only 0.001 cm$^3$/cm$^3$.

Bias calculated on the basis of a comparison of hourly SWC measured by CRP and simulated for 2013 is reported in Table 5. The average bias for the S80 open loop run is 0.11 cm$^3$/cm$^3$ while it is 0.02 cm$^3$/cm$^3$ for the BK50 open loop run. Bias of the BK50 open loop run was positive at Merzenhausen, Gevenich, Heinsberg, and Aachen, and it was negative at Rollesbroich, Kall, RurAue, and Wuestebach. Bias was zero at Wildenrath for the BK50 open loop run. Bias of the S80 open loop run was negative at all sites indicating that modeled SWC was higher than measured SWC. Joint state-parameter updates reduced the
absolute bias on average to 0.03 cm$^3$/cm$^3$ (PAR-S80-30) and 0.02 cm$^3$/cm$^3$ (PAR-S80-10) for the S80 soil map. In case of the BK50 soil map, the bias in 2013 increased to 0.03 cm$^3$/cm$^3$ by joint state-parameter updates. State updates without parameter updates reduced the biases only marginally to 0.01 cm$^3$/cm$^3$ for the BK50 soil map and to 0.09 cm$^3$/cm$^3$ for the S80 soil map. This indicates that state updates also can slightly improve SWC-characterization in the verification period due to improved initial conditions.

## 4.3     Temporal evolution of mean $E_{RMS}$

Fig. 4 shows the temporal evolution of the hourly $E_{RMS}$ calculated for all nine CRPs. $E_{RMS}$ was highest for the S80 open loop run and lowest for the PAR-S80-30 simulation. State updates did not improve modeled SWC as much as joint state-parameter updates improved modeled SWC. The $E_{RMS}$ in case of Stt-S80 also falls behind the $E_{RMS}$ of the BK50 open loop





run through most of the time. Joint state-parameter updates for the S80 soil map improved the $E_{RMS}$ throughout most of the time compared to the open loop simulations based on the BK50 and S80 soil maps. During the assimilation period 2011-2012, the PAR-S80-30 simulation performed best out of the four simulations. During the evaluation period 2013, OL-BK50 and PAR-S80-30 performed equally well except in summer 2013 when the PAR-S80-30 simulation yielded much higher
$E_{RMS}$-values than the BK50 open loop run.

### 4.4 Jackknife simulations

The jackknife simulations investigated the impact of the network of CRPs for improving estimates of SWC at locations between the CRPs, outside the network. The errors shown in Table 4 refers to the two open loop simulations (for the S80 soil map and the BK50 soil map) and the 18 jackknife simulations. All simulations with the S80 soil map resulted in an improved
$E_{RMS}$ in the jackknife simulations compared to the open loop simulation, except for Wildenrath. In all cases the $E_{RMS}$ was smaller than 0.10 $m^3/m^3$. Error reduction was smaller at sites where the open loop error was smaller. At sites with large open loop $E_{RMS}$, the assimilation could reduce the $E_{RMS}$ by 50 % or more. In case of the BK50 soil map, the jackknife simulations resulted in $E_{RMS}$-values below 0.10 $m^3/m^3$ at all sites. However, in this case only at Merzenhausen the $E_{RMS}$ was reduced
during the data assimilation period. At Wildenrath, the $E_{RMS}$ was highest for jk-BK50 (0.091 $m^3/m^3$) and jk-S80 (0.095 $m^3/m^3$). The average absolute bias for the jackknife experiments was 0.04 $cm^3/cm^3$ for both soil maps, BK50 and S80, in the evaluation period 2013 (Table 5). Hence, bias in the jk-S80-* simulations improved compared to the open loop run but not in the jk-BK50-* simulations, where bias was already small.

### 4.5 Temporal evolution of parameters

The temporal evolution of the percentage sand content during the assimilation period for the nine CRP sites is shown in Fig. 5 for PAR-S80-30, PAR-S80-10, PAR-BK50-30, PAR-BK50-10, jk-S80-30* and jk-BK50-30*. Time series start on March 20th, 2011, the date of the first assimilated CRP SWC retrieval at Wuestebach. Wuestebach and sites within the influence sphere of Wuestebach (Aachen, Kall and Rollesbroich) show also a change in sand content from this date onwards. All other
sites show a change in sand content in May 2012 when Rollesbroich and Merzenhausen start operating and their data is assimilated. All sites show variability in sand content over time. Wuestebach, Kall, RurAue, Rollesbroich and Heinsberg show some peaks in the time series. Merzenhausen, Aachen, Gevenich, and Wildenrath show a smoother course compared to the other sites. Sand levels approach a constant site-specific value for the sites Merzenhausen (45 %), Kall (30 %), Gevenich (41 %), RurAue (30 %), Heinsberg (42 %) and Wildenrath (62 %) with a reasonable spread amongst the experiments. The
spread in estimated sand content for the sites Wuestebach, Aachen and Rollesbroich is larger, and it seems not to have stabilized at the end of the assimilation. Sand content estimates of the jackknife simulations was close to the sand content of the other data assimilation experiments with joint state-parameter estimation at the sites Merzenhausen, Gevenich, RurAue and Heinsberg. Evolution of the sand content for the jackknife simulations showed larger deviations from the sand content estimated by other data assimilation experiments for the sites Wuestebach, Kall, Aachen, Rollesbroich and Wildenrath.



The soil hydraulic parameter B and saturated hydraulic conductivity are shown in Fig. 6 and Fig. 7 for PAR-S80-30, PAR-S80-10, PAR-BK50-30, PAR-BK50-10, jk-S80-30* and jk-BK50-30*. Updates of soil hydraulic parameters start in March and May 2011 with the assimilation of SWC retrievals depending on the location. The B parameter increases for all

simulations. Throughout the whole assimilation period B varies considerably within short time intervals. The total range of the B parameter is between 2.7 and 14 at all sites. At the sites Merzenhausen, Kall, Aachen, Gevenich and Rollesbroich, it generally ranges between 6 and 10. At Wuestebach, Heinsberg and RurAue, B ranges most of the time between 8 and 12, and at Wildenrath, B is below 8. Initial saturated hydraulic conductivity is rather high ($k_{sat}$>0.015 mm/s) in case of high sand content i.e. for the S80 soil map, and rather low ($k_{sat}$<0.005 mm/s) in case of low sand content i.e. for the BK50 soil

map. In case of the S80 soil map, at all sites except Wildenrath, high initial saturated hydraulic conductivity decreases quickly by parameter updates to values below 0.01 mm/s. The initial spread in $k_{sat}$ values amongst the simulation scenarios decreases at most sites. At Wuestebach, Merzenhausen, Aachen, Gevenich, RurAue and Heinsberg, the spread is rather small particularly at the end of the assimilation period, while at Wildenrath $k_{sat}$ ranges from 0.005 to 0.015 for individual experiments at the end of the assimilation period. The discussion section will elaborate more on this.

### 4.6    Latent heat and sensible heat

Latent heat flux or evapotranspiration (ET) is another important diagnostic variable of the CLM model and of importance for atmospheric models. Results of the data assimilation experiments showed that soil texture updates altered soil moisture states significantly. In Fig. 8 it is shown that joint state-parameter estimation also altered ET. Fig. 8 shows ET within the

evaluation period 2013 across the whole catchment for four simulations. On the one hand, ET was similar for both open loop simulations in the South of the catchment. On the other hand, ET in the North was up to 80 mm per year lower for the S80 open loop run compared to the BK50 open loop run. Regarding open loop runs, the differences can be linked to the drier soil conditions in case OL-S80 compared OL-BK50 simulation results. For PAR-S80-10, ET increased by up to 40 mm per year in the Northern part of the catchment through data assimilation. The differences between open loop ET and data assimilation

ET were larger for the S80 soil map than for the BK50 soil map. This could be related to the larger update in SWC in case of the S80 scenario compared to the BK50 scenario.

### 5    Discussion

The applied data assimilation scheme improved soil moisture characterization in the majority of simulation experiments with

the regional Community Land Model (CLM). During 2011 and 2012, the biased S80 soil map gave a $E_{RMS}$ up to 0.17 cm³/cm³ (at Rollesbroich) in the open loop simulation which left plenty of room for improvements. The soil map BK50 led to $E_{RMS}$-values in open loop simulations below 0.05 cm³/cm³ which left little room for error reduction considering the measurement error of 0.03 cm³/cm³. For the simulations starting with 80 % sand content, sand content was closer to the values of the BK50 soil map after joint state-parameter estimation. However, the temporal evolution of the updated soil



texture and the soil hydraulic parameters was not stable. Temporal fluctuations imply that there may be multiple or seasonal optimal parameter values. This is also supported by the findings of the temporal behavior of $E_{RMS}$ during the evaluation period e.g. when in the dry summer 2013 the $E_{RMS}$ peaked in the PAR-S80-30 simulation. Many possible error sources were not subject to calibration in this study but they could be crucial for an even better soil moisture and more stable soil

parameter estimation. In this study we only considered uncertainty of soil parameters, but also vegetation parameters are uncertain. Also a number of other CLM-specific hydrologic parameters (e.g. decay factor for subsurface runoff and maximum subsurface drainage) strongly influence state variables in CLM and hence show also potential for optimization. Considering this uncertainty could give a better uncertainty characterization. Precipitation is an important forcing for the model calculations and its estimate could be improved. For this study, precipitation data from the COSMO_DE re-analysis

were used. A product which optimally combines gauge measurements and precipitation estimates from radar could give better precipitation estimates. This could improve the soil moisture characterization and also potentially lead to better parameter estimates. Also other error sources like the ones related to the model structure play a significant role. This should be subject of future investigation.

Evaluation simulations for 2013 led to partly improved and partly deteriorated $E_{RMS}$ values when the BK50 soil map was used as prior information on the soil hydraulic properties. The simulations with the S80 soil map on the contrary showed an improved soil moisture characterization in all simulation scenarios and the updated soil hydraulic parameter estimates for those simulations approached the values of the BK50 soil map. These results indicate that the soil hydraulic parameters derived from the BK50 soil map were already well suited for soil moisture predictions and updating soil texture and soil

parameters could not improve further the results. $E_{RMS}$ values for simulations with state updates only (Stt-BK50 and Stt-BK50) in 2013 imply the beneficial role of state updates only. However, the improvements in the evaluation period by state updates (without parameter values) are small compared to the improvements obtained by joint state-parameter estimation. This illustrates the benefits of joint state-parameter updates compared to state updates only, and that soil moisture states are strongly determined by soil hydraulic parameters. It also illustrates that the improved characterization of soil moisture states

in the assimilation period which results in improved initial states for the verification period loses its influence in the verification period fast over time.

The jackknife simulations illustrated that a network of CRPs can improve modeled SWC if the soil map information is not sufficient. Temporal evolution of subsurface parameters of the jackknife simulations (e.g. jk-S80-*) was close to the

evolution of parameter estimates by other simulations (e.g. PAR-S80-10). Parameter estimates at jackknife test sites were inferred from multiple surrounding CRP sites, while updates at sites with CRP information were strongly inferred from single site information. A comparison of parameter estimates at the end of the assimilation period indicates that initial soil parameterization has a limited effect on the resulting parameter estimates. Parameter estimates of jk-BK50-30* and jk-S80-30* are close together at the end of the assimilation period. The CRP network led to improved results for the jackknife



evaluation simulations in case of the biased soil map. This suggests that assimilation of CRP data is particularly useful for regions with little information on subsurface parameters. We expect a tradeoff between the initial uncertainty on soil moisture content (related to the quality of the soil map and meteorological data) and the density of a CRP network. In case of a large uncertainty, like in regions with limited information about soils and a low density of meteorological stations, a sparse

network of probes can already be helpful for improving soil moisture characterization. On the other hand, in regions with a high density of meteorological stations and a high resolution soil map it can be expected that a high resolution CRP network is needed to further lower the error of soil moisture characterization. Further experiments in other regions with networks of CRPs are needed to get more quantitative information about this.

A question that remains to be answered is whether it is more beneficial to assimilate neutron counts measured by CRPs directly or to assimilate CRP SWC retrievals derived from the neutron counts, as done in this study. Fast neutron intensity measured by CRPs is also affected by vegetation. Neutron count rate decreases with increasing biomass because of the hydrogen content in vegetation (Baatz et al., 2015). Seasonal biomass changes at a single site have a rather small impact on neutron intensity compared to differences between grass land site and a forest site (Baatz et al., 2015). Therefore, using

measured neutron flux directly in a data assimilation framework in a catchment with different vegetation types would require to account for the effects of vegetation types on neutron intensity. Hence, vegetation estimates for each grid cell would be necessary. At present, there are two methods that include biomass in the CRP calibration process (Baatz et al., 2015;Franz et al., 2013b) but both methods naturally require accurate biomass estimates, which are typically not available. Besides the uncertainty associated with CRP methods using biomass in the calibration process, biomass estimates also come along with

high uncertainties. Therefore, in the case of a catchment with different vegetation types, it is desirable to circumvent the use of biomass estimates, and assimilate directly SWC retrievals obtained at the observation sites instead of assimilating neutron intensity. Therefore, this study uses CRP SWC retrievals in the data assimilation scheme assuming that seasonal changes of biomass can be neglected.

**6       Conclusions and Outlook**

This study demonstrates the benefits of assimilating data from a network of nine cosmic-ray probes (CRP) in the land surface model CLM version 4.5. Although information on neutron flux intensity was only available at few locations in the catchment, the local ensemble transform Kalman filter (LETKF) allows updating of soil water content (SWC) at unmonitored locations in the catchment considering model and observation uncertainties. Joint state-parameter estimates

improved soil moisture estimates during the assimilation and during the evaluation period. The $E_{RMS}$ and bias for the soil moisture characterization reduced strongly for simulations initialized with a biased soil map and approached values similar to the ones obtained when the regional soil map was used as input to the simulations. $E_{RMS}$-values in simulations with a regional soil map were not improved, because open loop simulation results were already close to the observations. The beneficial results of joint state-parameter updates were confirmed by additional jackknife experiments. This real-world case



study on assimilating CRP SWC retrievals into a land surface model shows the potential of CRP networks to improve subsurface parameterization in regional land surface models, especially if prior information on soil properties is limited. In many areas of the world, less detailed soil maps are available than the high resolution regional soil map applied in this study. In these areas, more advanced sub-surface characterization is possible using CRP measurements and the data assimilation

framework presented in this study.

For now, CRP neutron intensity observations were not assimilated directly. In future studies it would be desirable to use the COSMIC operator for assimilating neutron intensity observations directly. However, in this case the impact of biomass on the CRP measurement signal would have to be taken into account. Therefore, it is desirable to further develop the COSMIC

operator to include the impact of biomass on neutron intensities. Using the biogeochemical module of CLM would then allow to characterize local vegetation states as input for the measurement operator. Remotely sensed vegetation states are another option to characterize vegetation states as input for the measurement operator. Both methods require additional field measurements for the verification of vegetation state estimates. The further extension of the data assimilation framework would also enable the estimation of additional sub-surface parameters. The impact of other sub-surface parameters such as

subsurface drainage parameters and the surface drainage decay factor on SWC states and radiative surface fluxes has already been shown (Sun et al., 2013). Estimation of these parameters is desirable because of the inherent uncertainty of these globally tuned parameters. However, estimation of soil texture and organic matter content was demonstrated to be already beneficial for improved SWC modeling. Hence, this study represents a way forward towards the integration of CRP information in the calibration of large scale weather prediction models.

**Data Availabilitiy**

Most data presented in this study are freely available via the TERENO data portal TEODOOR (http://teodoor.icg.kfa-juelich.de/). Atmospheric data were licensed by the German Weather Service (DWD), and the BK50 soil map was licensed by the Geologischer Dienst Nordrhein-Westfalen.

**Acknowledgements**

We gratefully acknowledge the support by the SFB-TR32 "Pattern in Soil-Vegetation-Atmosphere Systems: Monitoring, Modelling and Data Assimilation" funded by the Deutsche Forschungsgemeinschaft (DFG) and TERENO (Terrestrial Environmental Observatories) funded by the Helmholtz-Gemeinschaft. The authors also gratefully acknowledge the computing time granted by the John von Neumann Institute for Computing (NIC) and provided on the supercomputer JURECA at Jülich Supercomputing Centre (JSC).





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





**Figures**

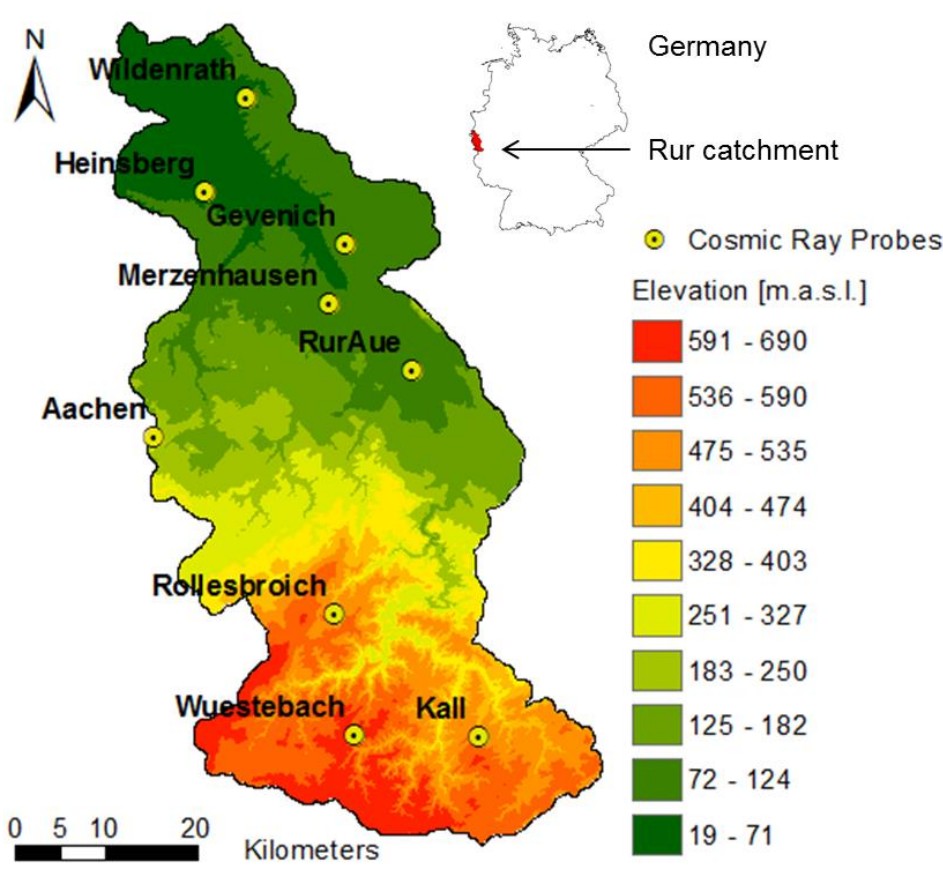

Fig. 1. Map of the Rur catchment and locations of the nine cosmic-ray probes. The hilly South of the catchment is prone to more rainfall,
lower average temperatures and less potential evapotranspiration than the North of the catchment.





Fig. 2. Temporal evolution of simulated SWC, calculated with open loop (OL-*) simulations and data assimilation including parameter updating (PAR-S80-30), together with the CRP soil water content retrieval (SWC) during the first year of simulation at the sites Merzenhausen and Gevenich. Simulated SWC was vertically weighted using the COSMIC operator to obtain the appropriate SWC corresponding to the CRP SWC retrieval.





Fig. 3. Temporal evolution of simulated SWC retrievals, calculated with open loop (OL-S80), data assimilation with state update only (Stt-BK50), and data assimilation including parameter updating (PAR-S80-30), together with the CRP soil water content (SWC) retrieval at the sites Heinsberg and Wildenrath for the data assimilation period 2011 and 2012, and the evaluation period 2013. Simulated SWC was vertically weighted to obtain the appropriate SWC corresponding to the CRP SWC retrieval.





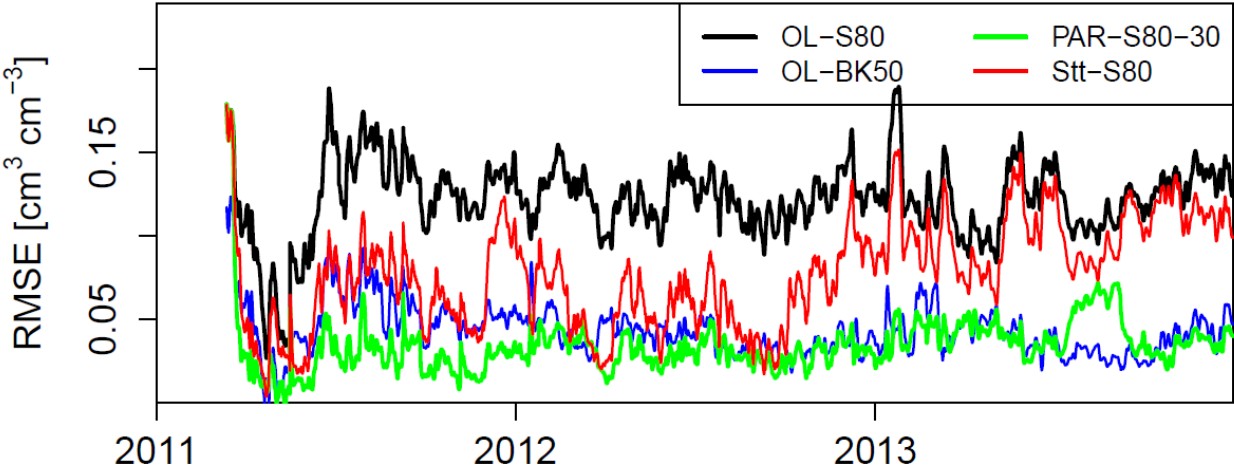

Fig. 4. Temporal evolution of root mean square error ($E_{RMS}$) for hourly SWC retrievals. $E_{RMS}$ is calculated hourly for nine CRP´s for open loop runs for soil maps S80 and BK50, joint state-parameter updates (PAR-S80-30) and state updates only (Stt-S80) during the assimilation period (2011 and 2012) and verification period (2013).





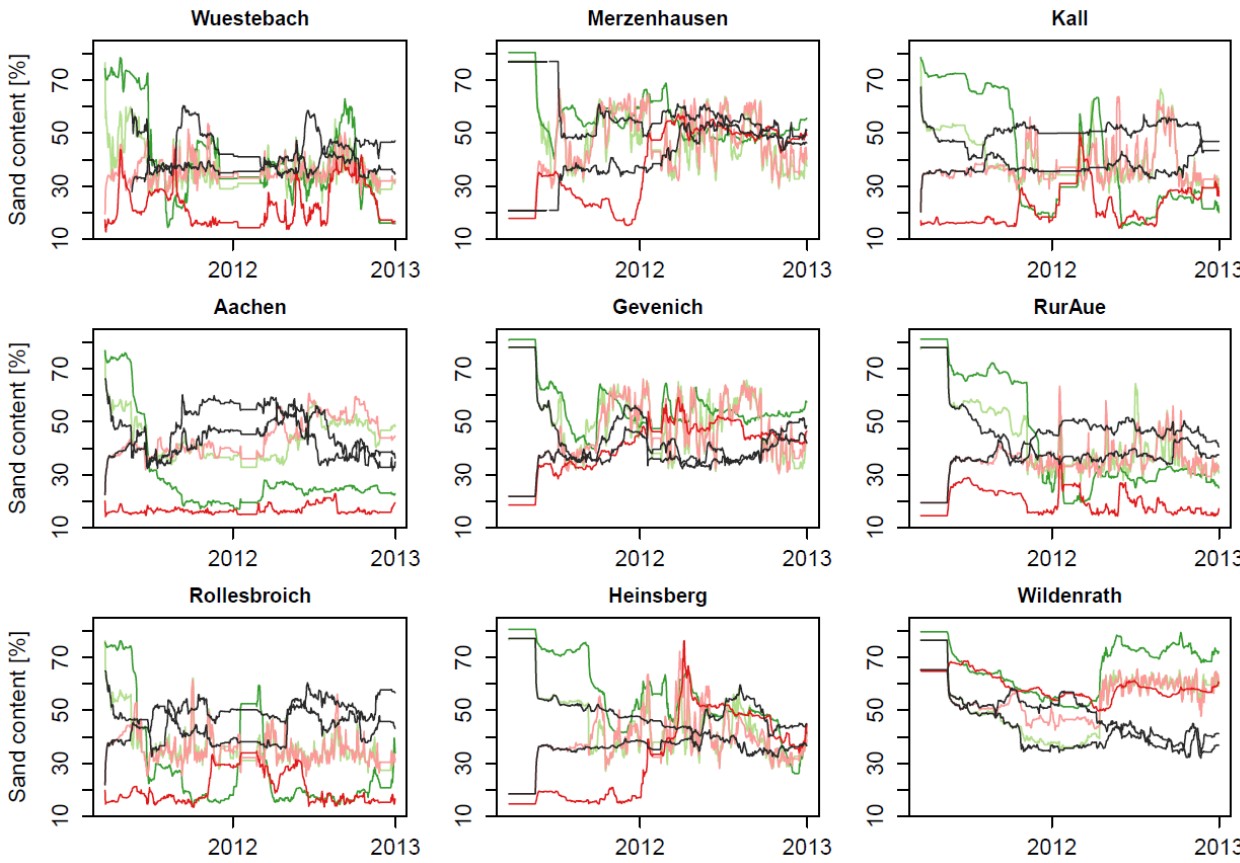

Fig. 5. Temporal evolution of the percentage sand content for simulations with parameter update: PAR-S80-30 (green), PAR-S80-10 (light green), PAR-BK50-30 (red), PAR-BK50-10 (light red), jk-S80-30* (black) and jk-BK50-30* (black).





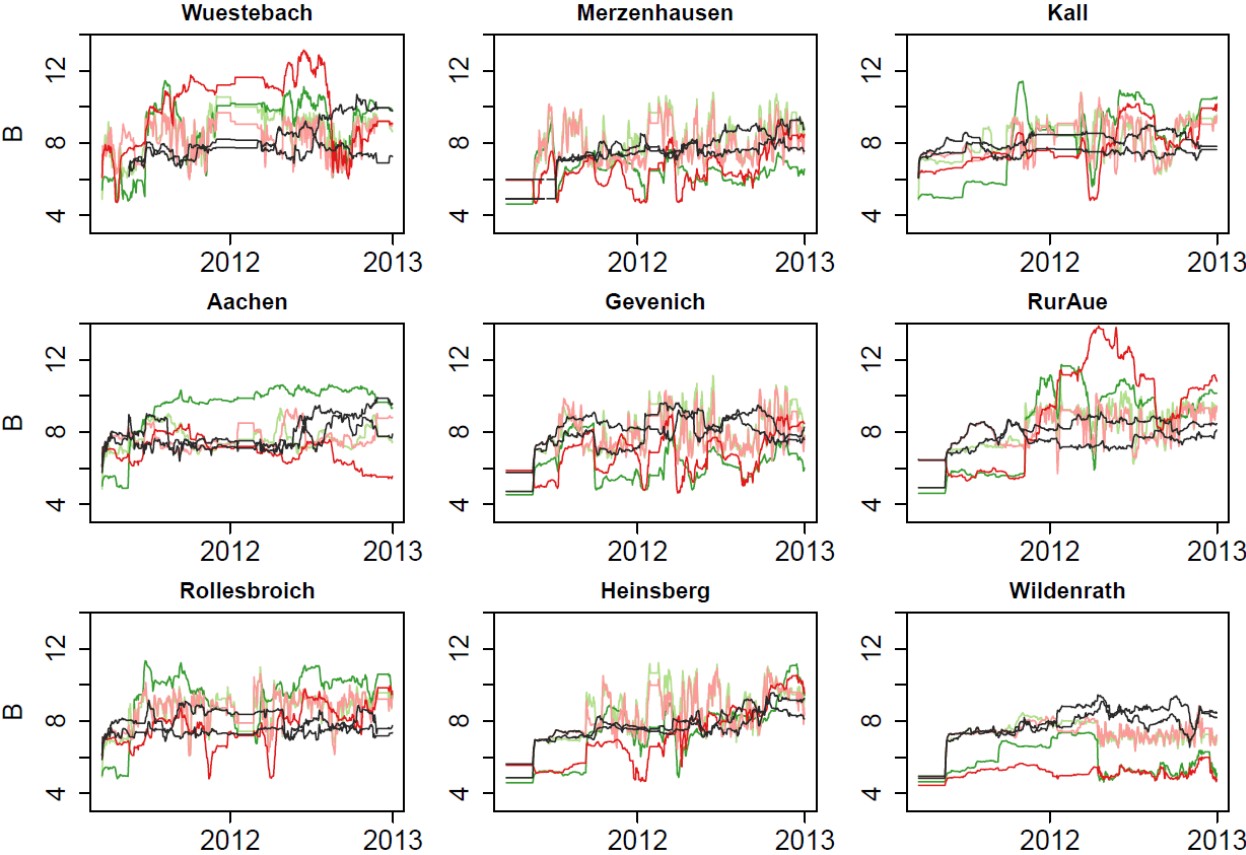

Fig. 6. Temporal evolution of the B parameter (top 15cm) for simulations with parameter update: PAR-S80-30 (green), PAR-S80-10 (light green), PAR-BK50-30 (red), PAR-BK50-10 (light red), jk-S80-30* (black) and jk-BK50-30* (black).



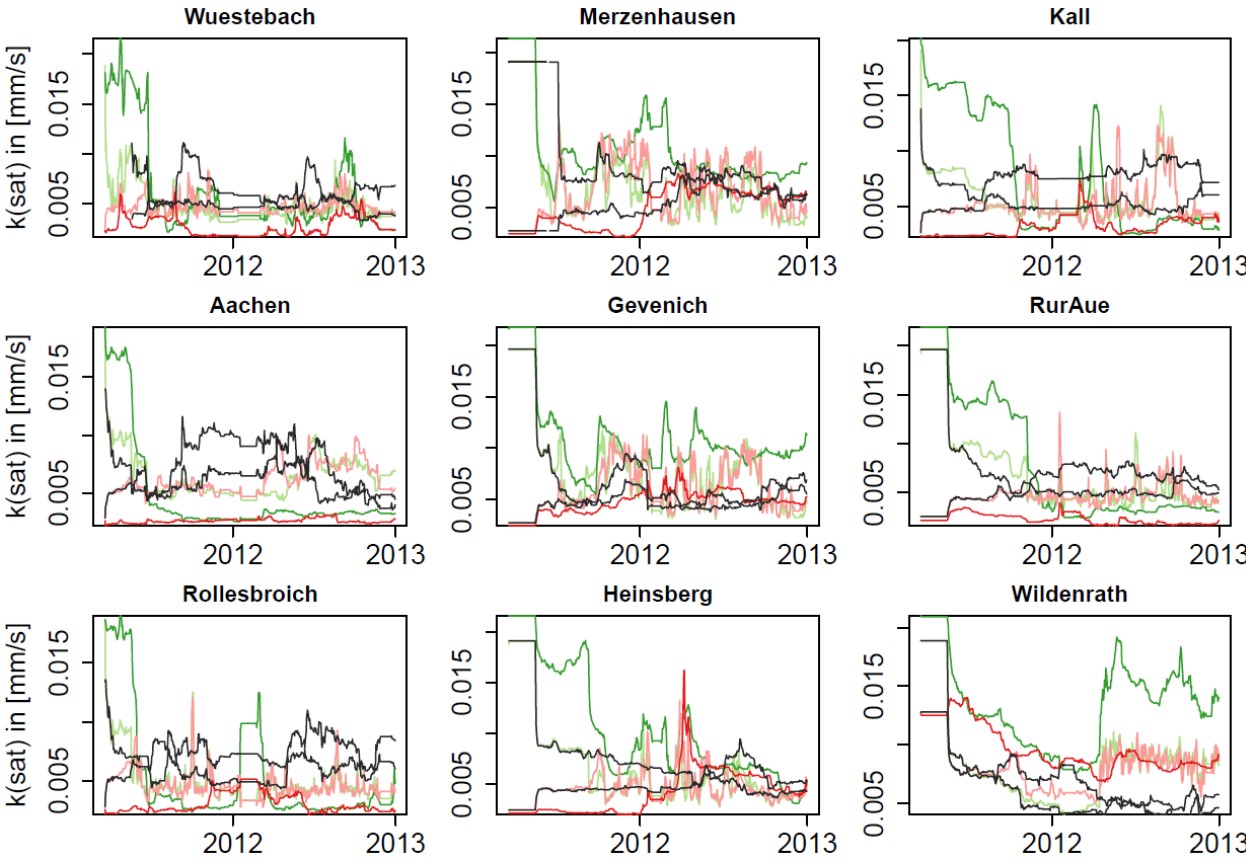

Fig. 7. Temporal evolution of saturated hydraulic conductivity (top 15cm) for simulations with parameter update: PAR-S80-30 (green), PAR-S80-10 (light green), PAR-BK50-30 (red), PAR-BK50-10 (light red), jk-S80-30* (black) and jk-BK50-30* (black).



Fig. 8. Annual evapotranspiration in the evaluation period (year 2013) for simulations OL-S80, OL-BK50, PAR-S80-10 and PAR-BK50-10.





**Tables**

Table 1: Site information on elevation (m.a.s.l.), average annual precipitation (mm/year), CLM plant functional type, sand content (%), clay content (%), and the date of the first SWC retrieval assimilated.

| Name | m.a.s.l. | Precip. | Plant functional type | Sand | Clay | Date of first assimilation |
|------|----------|---------|----------------------|------|------|----------------------------|
| Aachen | 232 | 952 | Crops | 22 | 23 | 13.01.2012 |
| Gevenich | 108 | 884 | Crops | 22 | 20 | 07.07.2011 |
| Heinsberg | 57 | 814 | Crops | 18 | 19 | 09.09.2011 |
| Kall | 504 | 935 | C3 non arctic grass | 20 | 22 | 15.09.2011 |
| Merzenhausen | 94 | 825 | Crops | 21 | 22 | 19.05.2011 |
| Rollesbroich | 515 | 1307 | C3 non arctic grass | 22 | 23 | 19.05.2011 |
| RurAue | 102 | 743 | C3 non arctic grass | 19 | 26 | 08.11.2011 |
| Wildenrath | 76 | 856 | Broadleaf deciduous temperate tree | 65 | 12 | 07.05.2012 |
| Wuestebach | 605 | 1401 | Needleleaf evergreen temperate tree | 19 | 23 | 20.03.2011 |





Table 2: Overview of simulation scenarios: Open loop (OL-*) with variation in the soil map BK50 or S80, data assimilation run with state
update (Stt) or joint state- and parameter update (PAR) with variation in the soil map perturbation (-10 or -30), and jackknife evaluation

5   runs (jk-S80-1 to 9, and jk-BK50-1 to 9).

| Simulation Code | Soil Perturbation | | Sand Content | | Update | |
|---|---|---|---|---|---|---|
| | 10 | 30 | BK50 | 80 % fix | State | Parameter |
| OL-BK50 | | + | + | | | |
| OL-S80 | | + | | + | | |
| Stt-BK50 | | + | + | | + | |
| Stt-S80 | | + | | + | + | |
| PAR-BK50-30 | | + | + | | + | + |
| PAR-BK50-10 | + | | + | | + | + |
| PAR-S80-30 | | + | | + | + | + |
| PAR-S80-10 | + | | | + | + | + |
| jk-BK50-1 to 9 | | + | + | | + | + |
| jk-S80-1 to 9 | | + | | + | + | + |





Table 3: $E_{RMS}$ (cm$^3$/cm$^3$) at CRP sites for open loop runs and different data assimilation scenarios, for the assimilation period (2011 and 2012).

| 2011 & 2012 | Rolles-broich | Merzen-hausen | Geve-nich | Heins-berg | Kall | RurAue | Wueste-bach | Aachen | Wilden-rath | Average $E_{RMS}$ |
|---|---|---|---|---|---|---|---|---|---|---|
| OL-BK50 | 0.054 | 0.067 | 0.039 | 0.035 | 0.042 | 0.027 | 0.041 | 0.032 | 0.017 | 0.039 |
| Stt-BK50 | 0.033 | 0.041 | 0.021 | 0.022 | 0.030 | 0.024 | 0.038 | 0.023 | 0.017 | 0.028 |
| PAR-BK50-10 | 0.036 | 0.036 | 0.019 | 0.021 | 0.033 | 0.025 | 0.035 | 0.045 | 0.015 | 0.029 |
| PAR-BK50-30 | 0.031 | 0.034 | 0.018 | 0.019 | 0.027 | 0.023 | 0.040 | 0.044 | 0.016 | 0.028 |
| jk-BK50-* | 0.070 | 0.058 | 0.073 | 0.035 | 0.048 | 0.050 | 0.053 | 0.050 | 0.091 | 0.059 |
| OL-S80 | 0.170 | 0.053 | 0.081 | 0.117 | 0.149 | 0.158 | 0.065 | 0.169 | 0.020 | 0.109 |
| Stt-S80 | 0.104 | 0.020 | 0.037 | 0.051 | 0.083 | 0.056 | 0.060 | 0.086 | 0.018 | 0.057 |
| PAR-S80-10 | 0.032 | 0.038 | 0.024 | 0.023 | 0.033 | 0.023 | 0.036 | 0.048 | 0.015 | 0.030 |
| PAR-S80-30 | 0.029 | 0.035 | 0.018 | 0.019 | 0.027 | 0.023 | 0.039 | 0.068 | 0.016 | 0.030 |
| jk-S80-* | 0.082 | 0.038 | 0.063 | 0.026 | 0.062 | 0.034 | 0.038 | 0.073 | 0.095 | 0.057 |





Table 4: $E_{RMS}$ (cm³/cm³) at CRP-sites for open loop, data assimilation and jackknife simulations on the basis of a comparison with CRP SWC retrievals during the verification period (2013). For each jackknife simulation only one $E_{RMS}$ is reported: The $E_{RMS}$ of the location that is meant for evaluation.

| Year 2013 | Rolles-broich | Merzen-hausen | Geve-nich | Heins-berg | Kall | RurAue | Wueste-bach | Aachen | Wilden-rath | Average $E_{RMS}$ |
|---|---|---|---|---|---|---|---|---|---|---|
| OL-BK50 | 0.044 | 0.065 | 0.036 | 0.027 | 0.048 | 0.038 | 0.048 | 0.042 | 0.017 | 0.041 |
| Stt-BK50 | 0.041 | 0.054 | 0.034 | 0.027 | 0.049 | 0.038 | 0.048 | 0.041 | 0.018 | 0.039 |
| PAR-BK50-10 | 0.068 | 0.062 | 0.036 | 0.038 | 0.056 | 0.056 | 0.043 | 0.058 | 0.017 | 0.048 |
| PAR-BK50-30 | 0.052 | 0.061 | 0.035 | 0.033 | 0.068 | 0.048 | 0.043 | 0.048 | 0.035 | 0.047 |
| jk-BK50-* | 0.036 | 0.047 | 0.028 | 0.025 | 0.042 | 0.031 | 0.040 | 0.054 | 0.106 | 0.045 |
| OL-S80 | 0.157 | 0.062 | 0.106 | 0.115 | 0.160 | 0.154 | 0.099 | 0.167 | 0.019 | 0.115 |
| Stt-S80 | 0.100 | 0.063 | 0.107 | 0.106 | 0.099 | 0.146 | 0.097 | 0.158 | 0.020 | 0.100 |
| PAR-S80-10 | 0.060 | 0.039 | 0.043 | 0.040 | 0.064 | 0.043 | 0.052 | 0.060 | 0.019 | 0.047 |
| PAR-S80-30 | 0.049 | 0.059 | 0.037 | 0.036 | 0.053 | 0.032 | 0.046 | 0.047 | 0.035 | 0.044 |
| jk-S80-* | 0.079 | 0.046 | 0.042 | 0.036 | 0.059 | 0.038 | 0.063 | 0.044 | 0.105 | 0.057 |




Table 5: Bias (cm$^3$/cm$^3$) at CRP-sites for open loop, data assimilation and jackknife simulations compared to CRP SWC retrievals during the verification period (2013). For each jackknife simulation only one bias is reported: The bias of the location that is meant for evaluation.

| Year 2013 | Rolles-broich | Merzen-hausen | Geve-nich | Heins-berg | Kall | Rur-Aue | Wueste-bach | Aachen | Wilden-rath | Mean absolute bias |
|---|---|---|---|---|---|---|---|---|---|---|
| OL-BK50 | -0.03 | 0.06 | 0.01 | 0.00 | -0.02 | 0.00 | -0.02 | 0.01 | 0.00 | 0.02 |
| Stt-BK50 | -0.01 | 0.04 | 0.00 | 0.00 | -0.01 | -0.01 | -0.02 | 0.00 | 0.00 | 0.01 |
| PAR-BK50-10 | 0.06 | 0.05 | 0.01 | 0.02 | 0.04 | 0.04 | 0.02 | -0.04 | 0.00 | 0.03 |
| PAR-BK50-30 | 0.03 | 0.05 | 0.00 | 0.02 | 0.04 | 0.03 | -0.01 | -0.03 | 0.03 | 0.03 |
| jk-BK50-* | -0.02 | 0.04 | 0.01 | -0.01 | -0.03 | -0.02 | -0.03 | -0.05 | 0.11 | 0.04 |
| OL-S80 | -0.17 | -0.05 | -0.08 | -0.12 | -0.15 | -0.16 | -0.09 | -0.17 | -0.01 | 0.11 |
| Stt-S80 | -0.09 | -0.05 | -0.10 | -0.10 | -0.08 | -0.14 | -0.09 | -0.15 | -0.01 | 0.09 |
| PAR-S80-10 | 0.03 | 0.05 | -0.01 | 0.02 | 0.03 | 0.01 | -0.01 | -0.03 | 0.03 | 0.02 |
| PAR-S80-30 | 0.04 | 0.02 | -0.03 | 0.02 | 0.05 | 0.03 | 0.03 | -0.04 | -0.01 | 0.03 |
| jk-S80-* | -0.07 | 0.03 | 0.02 | 0.02 | -0.04 | -0.02 | -0.04 | -0.03 | 0.10 | 0.04 |

