# Peer review of "Evaluation of a cosmic-ray neutron sensor network for improved land surface model prediction"

_Hydrology and Earth System Sciences, 2016_

## Referee Comment (RC1) · Anonymous Referee #1 · 14 Sep 2016

**OVERVIEW**

The manuscript investigates the value of a network of cosmic-ray probes for improving soil moisture estimation through land surface modelling. Specifically, the assimilation of soil moisture observations from 9 cosmic-ray probes into CLM (Community Land Model) is analysed for the Rur catchment in western Germany (period 2011-2013). The Local Ensemble Transform Kalman Filter is considered as data assimilation technique. Different real-world assimilation experiments were carried out also by considering an erroneous soil map. Results without and with the assimilation of soil moisture data were

analysed in order to assess their impact for improving the simulation of soil moisture.

**GENERAL COMMENTS**

The manuscript is well written and clear. The topic is of interest for the HESS readership as cosmic-ray probes represent a relatively new technology for ground measuring soil moisture over large areas. Therefore, we need to assess the impact of this new technology for improving land surface modelling. The paper describes several assimilation experiments in which soil moisture data from cosmic-ray probes are used for improving soil moisture modelling through CLM land surface model. Results are (quite) well described and clearly structured. However, in my opinion, several aspects should be improved/changed before the publication. I reported below a list of the general comments to be addressed with also the specification of their relevance.

1) **MAJOR**: Some of the results shown in the paper are well-known. I am aware that it is important to show real-world experiments, mainly by considering new technology, but the main results given in the paper were already reported in several previous studies: a) the assimilation of ground-based soil moisture data is able to improve soil moisture modelling, b) the joint state-parameter assimilation is better than the state assimilation only, and c) the assimilation is more effective when soil texture information are wrong (i.e., there's larger room for improvement). I believe that the paper results need to be published, but I would like to see some new findings that can be obtained by using the same material (data and modelling) presented in the paper.

For instance: A) What are the results if only one (or two) cosmic-ray probes are assimilated? In the real world it is expected that the number of probes will be limited and, hence, the use of a limited number of probes is surely of great interest. B) What is the impact in terms of fluxes? In section 4.6 a comparison of annual evapotranspiration maps without and with the assimilation is carried out, but simply showing that

the resulting maps are different. However, it is obvious that changing soil moisture will change evapotranspiration. Is it possible to perform an independent validation by using data about the actual evapotranspiration in the basin? Or likely by using discharge observations?

I believe that some new results should be included in the paper (even though I am aware that authors are usually reluctant to perform additional analyses). Moreover, the results in terms of soil moisture simulation should be synthetized (see Comment 5).

2) **MAJOR**: The description of the data assimilation experiments should be improved. As usually, a number of subjective choices were made in the setup of the data assimilation experiments, and these choices may have a significant impact on the results. For instance, a fixed error for soil moisture estimates from cosmic-ray probes is considered (0.03 cm$^3$/cm$^3$). Similarly, the perturbation factors for input data (precipitation and shortwave radiation) and parameters (10 and 30%) are arbitrarily selected. A sensitivity analysis on these choices should be carried out. It might be that different choices produce very different results.

3) **MODERATE**: Similarly as above, the selection of one single biased soil texture map is arbitrary. Why only one soil map? Why 80% of sand content and 10% of clay content is selected? What is the average sand and clay content percentage in the basin? Again, a sensitivity analysis is needed. Otherwise, it might be that a very large error in the soil map is used to highlight the positive impact of assimilating soil moisture data. What happens for a less biased map? This aspect should be clarified.

4) **MINOR**: In CLM the subsurface lateral flow is not considered. It has an impact on soil moisture simulation and, mainly, on the capability to modify soil moisture simulations at unmonitored locations. Therefore, I expect that the assimilation of in situ soil moisture data will have a local effect. However, the jackknifing data assimilation experiments show that the assimilation produces significant changes also at unmonitored locations. Why does it happen? I believe it should be clarified in the paper.

[Figure]

5) **MINOR**: In sections 4.2 and 4.5, too many details are provided in the description of the results for each single site. I suggest focusing on the most important results to improve their readability. Also, discussion section is too generic, especially the first paragraph.

In the specific comments, I added some corrections and suggestions that should be implemented.

On this basis, I believe the paper deserves to be published only after a major revision.

**SPECIFIC COMMENTS (P: page, L: line or lines)**

Title: The paper, in the current version, demonstrates that the assimilation of soil moisture data from cosmic-ray probes is able to improve soil moisture modelling, not "land surface modelling" (e.g., evapotranspiration or discharge fluxes). Therefore, I suggest changing the title.

Abstract: The abstract should include information on the location of the study area and on the employed data assimilation technique.

P4, L7: Formatting error for Kurtz et al. (2016). Please correct.

P5, L19: It should be COSMIC in place of COMIC.

P12, L11: Is sigma=0.5 considered for perturbing precipitation in all the assimilation experiments? Please clarify.

P12, L25: It should be "four data assimilation scenarios" in place of "six assimilation scenarios".

P19, L18-19: This sentence is too broad, please modify.

---

## Referee Comment (RC2) · Anonymous Referee #2 · 23 Sep 2016

The authors present an interesting and important study on investigating the benefits of integrating CRNP in data assimilation to improve atmospheric/land surface modeling. While I am not a data assimilation expert, this clearly seems to be a path forward on showing the importance and utilizing long-term monitoring networks for societal benefits. This is a novel study on using a network of CRNP to improve catchment water and energy balance. The authors show the utility of a network on CRNP on improving SWC states and soil parameter estimates in areas with poor or low meteorological coverage and soil information. While the paper is generally well written the authors missed some key references to put this work into proper context. In particular, the recent paper on the Plumber experiment of LSMs (Best 2015) and follow up paper on information content (Nearing 2016) should be discussed in light of this papers major findings. With these additional modifications the paper is appropriate for publication in

HESS.

Major Comments.

1. The recent paper by Best (2015) on the plumbing of LSMs needs to be discussed in the introduction and discussion. In addition, the follow up paper by Nearing (2016) on discussing the information content of LSMs is critical. Most notably, their findings on the importance parameterization, model physics, and boundary conditions affecting the partitioning of sensible and latent heat, and comparisons between a SWC benchmark are important. The authors need to discuss these results and how their findings agree or disagree with Best (2015) and Nearing (2016). (e.g. Pg. 3 L 12, Pg. 17 L 13, conclusions). Without this it is hard to place this work in its proper context for critical evaluation. 2. The work of Avery (2016) should also be discussed given the importance of soil and vegetation parameters discussed in this manuscript. This and point 1 will help update the referencing to be most up to date. (e.g. Pg. 10 L 25, Pg. 18 L 18).

Minor Comments:

Pg 4. L 6. The cases illustrate a way...

Pg 4. L 15-16. Sentence is awkward please revise.

Referecnes:

Avery, W., C. Finkenbiner, T. E. Franz, T. Wang, A. L. Nguy-Roberston, A. Suyker, T. Arkebauer, and F. Munoz-Arriola. 2016. Incorporation of globally available datasets into the roving cosmic-ray neutron probe method for estimating field-scale soil water content. HESS 20: 3859-3872. doi:10.5194/hess-20-3859-2016.

Best, M. J., G. Abramowitz, H. R. Johnson, A. J. Pitman, G. Balsamo, A. Boone, M. Cuntz, B. Decharme, P. A. Dirmeyer, J. Dong, M. Ek, Z. Guo, V. Haverd, B. J. J. Van den Hurk, G. S. Nearing, B. Pak, C. Peters-Lidard, J. A. Santanello, L. Stevens, and N. Vuichard. 2015. The Plumbing of Land Surface Models: Benchmarking Model Performance. J. Hydrometeorol. 16:3: 1425-1442. doi:10.1175/jhm-d-14-0158.1.

Nearing, G. S., D. M. Mocko, C. D. Peters-Lidard, S. V. Kumar, and Y. L. Xia. 2016. Benchmarking NLDAS-2 Soil Moisture and Evapotranspiration to Separate Uncertainty Contributions. J. Hydrometeorol. 17:3: 745-759. doi:10.1175/jhm-d-15-0063.1.
* * *

---

## Referee Comment (RC3) · Anonymous Referee #3 · 27 Sep 2016

The presented manuscript applies time-series data from 9 cosmic-ray neutron stations to the land-surface model CLM in the Rur catchment. The authors assimilate the data using an ensemble Kalman filter technique to update states and parameters of their model. The added value of training data in years 2011 to 2012 is assessed (1) by testing the model performance in year 2013, (2) by testing the model adaption capabilities to an invalid soil map, and (3) by jackknifing single stations from the training period. The application of the cosmic-ray neutron method in large-scale models is one of the challenges in state-of-the-art hydrology and thus the present study is worth to be published in the scope of HESS after major revision.

[Figure]

**1 Evaluating the overall quality**

Large parts of the manuscript are written in the style of a protocol, by listing lots of other publications who used similar approaches, by mentioning tools that were used, and by reporting every step of the performed analysis. However, I believe that scientific articles should be entitled to challenge their own strategy by discussing alternative methods, by justifying their selection of tools and decisions, and by explaining the corresponding implications. I would thus recommend to rewrite and extend major parts of the introduction and method section. Therein, some literature reviews are unnecessary and probably unrelated to the study and can be omitted or need further explanation (see line-by-line comments). Here, I would suggest to follow the guideline that cited papers should be discussed, and not just mentioned. Other parts concerning the data integration and the SWC model need to be described in more detail. I would further recommend to reduce the detail of the results section, which is hard to follow without proper discussion, and thus to merge it with the discussion section.

The results of the study were well structured and decribed, but are not entirely novel, and not sufficient to provide answers to all research questions raised by the authors. For example, to assess the value of a CRNS network of certain density to the performance of a land-surface model, various fractions of the 9 stations should be tested as requested by Referee #1.

Furthermore, the study uses some questionable assumptions, like a constant error of soil moisture data (although neutron measurement uncertainty highly varies with wetness condition), or the assimilation of SWC data assuming homogeneous vertical profiles and no changes of seasonal biomass (see line-by-line comments for details). Another questionable approach is to allow static site-specific parameters to be variable in time, e.g., hydraulic conductivity or soil porosity. This is highly counter-intuitive and should be discussed with respect to uncertain data and/or model conceptualization.

I agree to most of the general comments made by Referee #1 and will thus compensate

the previous reviews by detailed line-by-line comments below.

**2 Line-by-line comments, scientific questions/issues, and technical corrections**

- Title ("Evaluating the value of a network of cosmic-ray probes for improving land surface modelling"): I'd suggest to remove "the value of" to simplify the title. Furthermore, state observations usually do not improve a *model*, they rather improve *model results*, e.g. predictions.

- L1: "Land surface models can model": bad phrasing, replace "can model" e.g. by "describe".

- L3: "CRP", please use the newly accepted abbreviation CRNS (cosmic-ray neutron sensing/sensor) with regards to the recent 5th COSMOS workshop.

- L14: improve readability, split in two sentences.

- L18: please add a statement about the impact of your findings for the scientific community.

- L21-22: this sentence needs a reference.

- L24: " and is " → "while it is"

- L27-28: the given number of references here appears to overwhelm the statement and its low relevance to your paper. Please use only the 1 or 2 most important citations.

- L30-31: Please discuss the alternatives in more detail to strengthen your decision to use CRNS technology. Were space-borne remote-sensing products assimilated to LSMs before? Why wasn't it successful? What about the use of airborne products with higher resolution and depth? You could also mention point-scale or large-scale soil moisture monitoring networks which have been used for evaluation of land surface models.

- L31: "not reliable for areas with dense vegetation": a paper by the same first author recently found that CRNS is also influenced by dense vegetation. Is it more reliable?

- L33: the selection of citations for this statement appears to be random/unrelated. If you want to provide references for the "intermediate scale", Zreda 2008 and Köhli 2015 might be appropriate.

- L34: "desired application scale of land surface models": please make the reader happy by finally providing concrete information. What is the scale? Are you talking about centimeters or lightyears? Please do not use citations inflationary and do not keep them untouched. How do the three citations help you to support your argumentation?

- L1: omit "fast" as it repeats with the next sentence.

- L3: add "fast" to make clear that the sensor measures the non-moderated neutrons.

- L4: "15 ha", your SWC range seems to be 10 to 40%, which leads to an approximate CRNS footprint of 7 to 14 ha following Köhli 2015, excluding vegetation

and altitude influence. You could write "maximum area of 15 ha" to circumvent mentioning this variability.

- L6: omit "Desilets and Zreda, 2013" as it does only marginally address heterogeneous averaging. Franz 2013a is already a great reference to this topic, Köhli 2015 also touched this.

- L8: Bogena et al. 2013 did not perform simulations to the penetration depth. Instead, Franz et al. 2012 (`doi:10.1029/2012WR01187`) and Köhli et al. 2015 provided simulations that both support these values.

- L11: add a reference for COSMOS-UK, Evans et al. 2016, `10.1002/hyp.10929`

- L13-15: please rephrase to make clear what data assimilation is and is not.

- L15: It is not clear why you choose EnKF. Please at least mention other techniques and provide reasons for your choice. The sentence further should be moved to the end of the paragraph after you have introduced the history of DA.

- L16-34: This historical overview appears to be unnecessary in the context of your study. Neither do you explain what things like " four-dimensional variational DA" are, nor is the relation to your work described. Furthermore, citations are used inflationary again. Please reduce this paragraph to the key publications which support your study. Also think about moving certain studies about ensemble size, multiple time steps, and other filtering approaches to the methodology section, where you need justification for your approach.

- L30-32: Just to emphasize the previous comment, these lines particularly carry no information for non-experts due to the lack of explanation.

[Figure]

- L1-19: As stated before, the whole literature review appears to be random and irrelevant to your work. Or at least the relations are not explained. For example, work from Montzka 2011;2013 and Han 2014b appear to be of some relevance for you, prior to others.

- L23-24: "Its capability to propagate surface soil moisture information into the deeper soil column was analyzed by Rosolem et al. (2014)", what does this sentence mean?

- L26: "The COSMIC operator", third repetition as a sentence starter.

- L27-29: combine those sentences: "neutron observations have been used to update states (...) and hydraulic parameters (...)"

- L29: "showed" → "demonstrated"

- L29-30: be more correct in phrasing. Rephrase that Villarreyes 2014 used a different model, but also estimated hydr. parameters by inversion. Han 2016 did so too, using support from neutron data, but neutron assimilation alone does not "update" a hydraulic parameter.

- L31: "This work further explores", omit "further". Until now it is not clear what *this work* does, you only told stories about work of others. Please summarize which of the presented approaches you are picking up and what scientific novelty you add.

- L3-4: "the soil moisture characterization at the larger catchment scale", what exactly is meant by these terms, and how do you measure improvement?

[Figure]

- L4: "how dense the CRP network should be", do you answer this question?

- L7-8: "soil maps and atmospheric forcings show spatial correlations over larger distances", this is an interesting point, please provide reference. Isn't the large-scale heterogeneity of soil maps only an artefact of soil data scarcity?

- L9: "10 stations", do you assimilate all 10, or just 9?

- L15: "feasibility of the updated large scale soil hydraulic parameters", how can a parameter be feasible? Please clarify your novel research question.

- L18-19: The sentences can be omitted as being obvious.

- L6: correct wording, a "process" can not be "solved"

- L10: "Oleson et al. (2013) provide further details on CLM4.5", redundant information with regard to L5-6.

- L10-12: provide reasons why you artificially limit the scope and complexity of your study. What process would a "biogeochemical module" have added and why are they not important here compared to a prescribed LAI?

- L14: please finally (after lots of references in the introduction) provide concrete information about the grid size in your study (the reader is still lost between centimeters and lightyears)

- L23: use standard format for functions, $k[z] \rightarrow k(z)$

- L24: format $z \rightarrow z$
- L24: what is the difference between "soil moisture" and SWC? Why are you using the expression $\theta$ here, while SWC is used elsewhere (e.g., eqs. 24 and 25)?

- L25: use the more convenient expression $k_{\text{sat}}(z)$,

- L26 (eq. 1):

    – format $k[z] \rightarrow k(z)$,
    – rewrite $k_{sat,z} \rightarrow k_{\text{sat}}(z)$ as this is a functional relationship. In contrast, indexing a state variable $\theta_i$ is ok.
    – omit occurrences of $0.5$ since $\frac{0.5}{0.5} \approx 1$,
    – case conditions (e.g., $1 < i < N$ ...) are usually preceded by a comma in each line
    – the curly bracket on the right is not common in multi-case equations.

- eq. 3 and 5: reformat $sand \rightarrow$ sand, same for clay.

- L6: "whereas", split sentence here.

- eqs. 9 and 10: this is a single equation, requiring only a single equation number, and a multi-case alignment using a curly bracket

- L12: reformat $mm \rightarrow$ mm,

- whole page: please motivate the reader why these details are important for your research question. Also provide information where all these empirical (fixed) parameters (or regression coefficients) are coming from. Is the underlying theory so well understood that no uncertainties or further dependencies are required?

[Figure]

- L4: "COSMIC parameterizes interactions". The *interactions are parameterized* by the underlying physical cross-section data. COSMIC rather *parameterizes the neutron transport*.

- L7-8: Repetition from the introduction.

- L10: "high energy neutrons are reduced" → "the number of high energy neutrons is reduced"

- L11: "with less energy in each soil layer", misleading/unphysical. Fast neutrons typically evaporate with constant energy.

- L12: rewrite "soil interaction", as fast neutrons predominantly interact with the water.

- L16-22 and eq. 14: this part can be omitted, since it is already well described in papers from Shuttleworth and Baatz, and does not add to the message of this paper. If you decide not to omit it, replace $\theta$ in eq. 14 to avoid confusion with soil moisture.

- L22: explain to the reader how the 300 soil layers in COSMIC communicate with the 10 soil layers from CLM.

- L26: what is a "COSMIC soil surface"?

- L25ff: it looks like you are not assimilating neutrons, but reiterating SWC from neutron data. The whole paragraph creates a great confusion about what the difference is between SWC, CLM SWC, weighted CLM SWC, and CRP SWC. In contrast to other less relevant paragraphs in this section, this part is highly unclear and simultaneously highly important to understand the most important

part of your model. Please rephrase the whole paragraph and clarify to the reader what exactly you do, and why (i.e., why not assimilating $N$ directly?)

- L5-10: Your paper is not a protocol. Again, it is described *what* you are using and *who else* used it, but the reader is left with the question *why* you (and others) made this decision. Shortly explain advantages of your strategy and why it serves your research question better than others.

- L11: what is $f$ in $\vec{x}^f$ ?

- L19: confusing typesetting. Is it $\vec{H}$ as a function of the COSMIC model, or is $\vec{H}$ identical with the COSMIC operator?

- eq. 19: do not use $T$ as a symbol for transposition, there is a reserved symbol for this: $\vec{Y}^\top$.

- L18: redundant sentence.

- L20: why these values? is it comparable with the catchment-mean texture? If your question is, what impact a rough and uncertain soil map in data scarce region would have, wouldn't it be more reasonable to *smooth out* the existing soil map to a very rough degree, rather than using a completely arbitrary soil map?

- L3-5: omit physical units (they are irrelevant in this context).
- L8-11: How do you justify the perturbation of physical soil parameters like porosity and texture? Does the uncertainty of the soil map justify the huge variation ranges applied in this work? Are models allowed to adapt their physical basement to hydrological data (which also show uncertainty)?

- L15: omit "="

- How was the CRP SWC uncertainty determined? Assuming a constant CRNS error is not physical and might have substantial influence on the results (to be tested). For example, the error of neutron observations $N$ is $\sqrt{N}$, while $N$ can almost double from very wet to very dry conditions, which leads to a variation of the neutron uncertainty by 30%. This can propagate through the non-linear relation to soil moisture in such a way that your observed SWC is significantly more uncertain in wet periods compared to dry periods. Consequently, the DA approach should give more weight to dry periods during assimilation.

- L16: Why do you use RMSE, although many alternative measures are accepted as state-of-the-art measures for time series evaluation, e.g., KGE or NSE, in order to assess bias, deviation, and correlation simultaneously?

- eqs. 23 and 24: reformat $SWC \rightarrow$ SWC, same with $RMSE$ and $bias$, as those are single multi-letter variables, not products of multiple single-letter variables. Following this style guide, rewrite $\mathrm{E}_{RMS} \rightarrow E_{\mathrm{RMS}}$. You can even omit "RMS" since $E$ is the only error used in this work. This would improve readability of the results section.

- L17-26: It is argued that changes in SWC states have impact to simulated ET flux. However, only for state-parameter updates (L19). Why is ET not affected by (SWC) state updates only?

- L32: "to $E_{RMS}$-values", omit "-"

- L9: if precipitation data from COSMO_DE was used, why was this information omitted in the method section (only mentioning DWD)?

- L26: replace "fast" with "quickly".

- L10-23: This question already needs an answer in the method section, I'd suggest to move the whole paragraph.

- L10-23: I cannot follow the argumentation. Baatz et al. 2014 suggested a correction function for neutron counts based on vegetation estimates. In your model, you already have LAI data every month, implementation of the correction functions in the model would probably be straight forward. Furthermore, to convert neutron data to SWC, some vegetation correction would be necessary, too. Third, assimilating CRP SWC assumes homogeneous vertical SWC profiles (before iteration), this assumption would be unnecessary if neutrons would be assimilated directly. I am afraid that this topic is more complex and needs further discussions and tests. It would be most convincing if you could show that neutron assimilation indeed gives different results than CRP SWC assimilation.

- L27: "neutron flux intensity", do you mean flux or intensity or both?

- L27: "Although ... only available at few locations", write more positively. Neutron data was available at up to 9 locations, which was intended to be the amazing novelty compared to other catchments!

Figures

1. South → south, same with North.

2. Please add grid lines

3. Please add grid lines

4. Please add grid lines

5. It is hard to distinguish two black lines with different meaning. Further indication of the expected "true" sand content (given by the soil map or soil samples) would be helpful to evaluate these plots.

6. This figure is not understandable without the text. Please shortly provide information about the $B$ parameter in the caption to understand the message of this figure.

7. replace k(sat) → $k_{\text{sat}}$. It would be interesting to also show the evolution of the soil porosity parameter together with an indication of its measured value. Why does hydraulic conductivity (and probably also porosity) vary over time at individual sites? Those are expected to be constant physical parameters of the sites. In my opinion this is a serious flaw of the DA approach used here.

8. The purpose of this figure is not clear, as no observation data is provided to evaluate the model performance with respect to simulated latent heat.

Tables

1. what is C3? Replace "non arctic" with "non-arctic", probably add a citation to the caption for plant functional types.

2. •

3. improve readability by increasing font weight (boldness) for particularly good cases below an RMSE threshold, which is a common strategy in many journals.

4. same as 3. Rephrase the last sentence.

5. same as 4.

---

## Author Comment (AC1) · 7 Oct 2016

Text formats:

**Referee – bold, non-italic**

Answer of the author – non-bold, non-italic

*Cited text – non-bold, italic*

**Reviewer 1**

**GENERAL COMMENTS**

**The manuscript is well written and clear. The topic is of interest for the HESS readership as cosmic-ray probes represent a relatively new technology for ground measuring soil moisture over large areas. Therefore, we need to assess the impact of this new technology for improving land surface modelling. The paper describes several assimilation experiments in which soil moisture data from cosmic-ray probes are used for improving soil moisture modelling through CLM land surface model. Results are (quite) well described and clearly structured. However, in my opinion, several aspects should be improved/changed before the publication. I reported below a list of the general comments to be addressed with also the specification of their relevance.**

Thank you for your positive evaluation and for your time reviewing this manuscript. We carefully address your comments in the following indicating the planned revision of the manuscript.

**1) MAJOR: Some of the results shown in the paper are well-known. I am aware that it is important to show real-world experiments, mainly by considering new technology, but the main results given in the paper were already reported in several previous studies: a) the assimilation of ground-based soil moisture data is able to improve soil moisture modelling, b) the joint state-parameter assimilation is better than the state assimilation only, and c) the assimilation is more effective when soil texture information are wrong (i.e., there's larger room for improvement). I believe that the paper results need to be published, but I would like to see some new findings that can be obtained by using the same material (data and modelling) presented in the paper.**

As pointed out later in this response, the present study describes a way to propagate CRP measurements into horizontal space using the local ensemble transform Kalman Filter:

*"Although information on neutron flux intensity was only available at few locations in the catchment, the local ensemble transform Kalman filter (LETKF) allows updating of soil water content (SWC) at unmonitored locations in the catchment considering model and observation uncertainties."*

Another key finding was pointed out in the abstract, where we quantified the possible improvement of soil moisture prediction using a land surface model:

*"For the biased soil map, soil moisture characterization improved in both periods strongly from a ERMS of 0.11 cm3/cm3 to 0.03 cm3/cm3 (assimilation period) and from 0.12 cm3/cm3 to 0.05 cm3/cm3 (verification period) and the estimated soil hydraulic parameters were after assimilation closer to the ones of the regional soil map. "*

This is a major novelty. In the discussion section, the limited land surface model precision was estimated and related to the measurement error:

*"The soil map BK50 led to ERMS-values in open loop simulations below 0.05 cm3/cm3 which left little room for error reduction considering the measurement error of 0.03 cm3/cm3."*

We would like to draw your attention to the conclusion section of this manuscript. The conclusion adds important findings of this work based on the results and discussion Sections. This work stresses the direction of research which should be taken for the use of CRP data in parameter estimation of land surface models:

*"For now, CRP neutron intensity observations were not assimilated directly. In future studies it would be desirable to use the COSMIC operator for assimilating neutron intensity observations directly. However, in this case the impact of biomass on the CRP measurement signal would have to be taken into account. Therefore, it is desirable to further develop the COSMIC operator to include the impact of biomass on neutron intensities."*

**For instance:**

**A) What are the results if only one (or two) cosmic-ray probes are assimilated? In the real world it is expected that the number of probes will be limited and, hence, the use of a limited number of probes is surely of great interest.**

Thank you for the suggestion. We will present in the revised version of the manuscript additional simulation experiments with a smaller number (4) of cosmic ray probes. This allows verification at five other cosmic ray probes.

**B) What is the impact in terms of fluxes? In section 4.6 a comparison of annual evapotranspiration maps without and with the assimilation is carried out, but simply showing that the resulting maps are different. However, it is obvious that changing soil moisture will change evapotranspiration. Is it possible to perform an independent validation by using data about the actual evapotranspiration in the basin? Or likely by using discharge observations? I believe that some new results should be included in the paper (even though I am aware that authors are usually reluctant to perform additional analyses). Moreover, the results in terms of soil moisture simulation should be synthetized (see Comment 5).**

In the revised version of the manuscript we will present a comparison of simulated ET (before and after data assimilation) and measured ET by eddy covariance at a few locations. We will discuss in detail the implications of the comparison. However, a comparison for many EC-stations is not possible as the correction of the energy balance gaps of the EC-data and many data gaps in the EC-data are not trivial to be resolved.

**2) MAJOR: The description of the data assimilation experiments should be improved. As usually, a number of subjective choices were made in the setup of the data assimilation experiments, and these choices may have a significant impact on the results. For instance, a fixed error for soil moisture estimates from cosmic-ray probes is considered (0.03 cm3/cm3). Similarly, the perturbation factors for input data (precipitation and shortwave radiation) and parameters (10 and 30%) are arbitrarily selected. A sensitivity analysis on these choices should be carried out. It might be that different choices produce very different results.**

Thank you for these constructive suggestions. We agree that a number of assumptions needed to be made. The assumption on the measurement error for the soil water content was thoroughly evaluated and more details can be found in our earlier papers on this (e.g., Bogena et al., 2013; Baatz et al., 2014; Baatz et al., 2015). We feel that there is no need to repeat experiments with other values for the measurement error.

However, we agree that perturbation of meteorological forcings and soil hydraulic parameters are subject to larger uncertainty. However, the perturbation of soil hydraulic parameters is done in CLM by perturbing soil texture. The currently applied perturbations of up to 30% are already large and we feel that this covers adequately the uncertainty with respect to texture. We also already tested two different magnitudes of perturbation of precipitation, including a 50% and 100% error. We feel that the applied perturbations are realistic and the difference in the two applied perturbations was already large.

In addition, simulations were not strongly affected by the magnitude of the perturbations. The simulations are also CPU-intensive. We expect that an extension of the experiments for further magnitudes of perturbation would not supply significant additional insights.

**3) MODERATE: Similarly as above, the selection of one single biased soil texture map is arbitrary. Why only one soil map? Why 80% of sand content and 10% of clay content is selected? What is the average sand and clay content percentage in the basin? Again, a sensitivity analysis is needed. Otherwise, it might be that a very large error in the soil map is used to highlight the positive impact of assimilating soil moisture data. What happens for a less biased map? This aspect should be clarified.**

Thank you for this suggestion. In the revised version, we intend to include a simulation with a third soil map that may be close to the expected values of the BK50 soil map. Table 1 shows the sand content at the nine locations assimilated for the BK50 soil map. However, we would like to stress that the high resolution BK50 soil map represents not the *true* sand and clay content but is rather the expected sand and clay content with errors represented by the perturbation. Considering this, the BK50 soil map already is a "less biased map". We will add information on catchment wide average sand and clay content, and what was the motivation to select the biased soil map as initial soil map in part of the simulation experiments:

*"The BK50 soil map provides the initial high resolution soil texture for the catchment and is the most detailed soil map available for the defined region. Average sand and clay content of the catchment are 22.5% and 21.4%, respectively. As an alternative, simulations were also performed for a biased soil texture distribution with a fixed sand content of 80 % and clay content of 10 % (S80 soil map). This represents a large error with respect to the expected soil properties. However, perturbations of 10 and 30 % guarantee some variability in the initial soil properties. The S80 soil map simulations allow evaluating the joint state-parameter estimation approach because given the expected bias, we can evaluate whether and to what extend the soil properties are modified by the data assimilation to be closer to the available high resolution soil map."*

**4) MINOR: In CLM the subsurface lateral flow is not considered. It has an impact on soil moisture simulation and, mainly, on the capability to modify soil moisture simulations at unmonitored locations. Therefore, I expect that the assimilation of in situ soil moisture data will have a local effect. However, the jackknifing data assimilation experiments show that the assimilation produces significant changes also at unmonitored locations. Why does it happen? I believe it should be clarified in the paper.**

Thank you for this detail. It is correct that CLM does not consider subsurface lateral flow. The updates of soil moisture in space depend on spatial correlations of soil moisture. In this study, spatial correlations of soil moisture are the consequence of spatial correlations of the atmospheric forcings, spatial correlations of soil hydraulic parameters and their interaction with the land surface model. Atmospheric reanalysis data and the soil map provided a good basis for the imposed spatial correlation structure. The imposed spatial correlation structures on the perturbations determine to a large extend the soil moisture updates in space.

**5) MINOR: In sections 4.2 and 4.5, too many details are provided in the description of the results for each single site. I suggest focusing on the most important results to improve their readability. Also, discussion section is too generic, especially the first paragraph. In the specific comments, I added some corrections and suggestions that should be implemented. On this basis, I believe the paper deserves to be published only after a major revision.**

Results section and discussion section will be shortened respectively sharpened, also taking into account the detailed comments of reviewer #3.

**SPECIFIC COMMENTS (P: page, L: line or lines) Title: The paper, in the current version, demonstrates that the assimilation of soil moisture data from cosmic-ray probes is able to improve soil moisture modelling, not "land surface modelling" (e.g., evapotranspiration or discharge fluxes). Therefore, I suggest changing the title. Abstract: The abstract should include information on the location of the study area and on the employed data assimilation technique.**

**P4, L7: Formatting error for Kurtz et al. (2016). Please correct.**

Thank you. This will be corrected.

**P5, L19: It should be COSMIC in place of COMIC.**

Thank you. This will be corrected as well.

**P12, L11: Is sigma=0.5 considered for perturbing precipitation in all the assimilation**

**experiments? Please clarify.**

We changed:

*"For jackknife simulations, the perturbation of soil texture was set to 30 % and precipitation perturbation was set to σ= 0.5 for all experiments."*

**P12, L25: It should be "four data assimilation scenarios" in place of "six assimilation**

**scenarios".**

Thank you. This will be changed.

**P19, L18-19: This sentence is too broad, please modify.**

Thank you. We agree and changed the sentence to:

*"Hence, this study represents a way forward towards the integration of CRP information in the calibration or real-time updating of land surface models."*

---

## Referee Comment (RC4) · Anonymous Referee #4 · 10 Oct 2016

OVERALL QUALITY

The paper by Baatz et al. (2016-432) describes an effort to use soil moisture data from nine closely-spaced (2000 km2) cosmic-ray probes (CRPs) with data assimilation scheme to improve the assessment of soil moisture in land-surface models. The goal is worthy and the execution is thorough. The results are significant: (1) the joint state (soil moisture) and parameter (soil properties, like sand percentage) estimation within data assimilation scheme produces better results than just state estimation; (2) in absence of soil data and meteorological data, CRPs alone can improve data assimilation results. On that account, the paper is suitable for publication in HESS.

However, I am less certain about the significance of these results in light of the finding that the parameters change in time and in many cases never converge. This is only

possible if the parameters are fitting parameters rather than physical parameters. So we end up with better results, but possibly only by statistical manipulation rather than by improved understanding of the physics. Is this progress? I would like to see at least some discussion of this issue in the paper in its final form.

SPECIFIC COMMENTS

Why are RMSE and bias discussed separately if they are essentially the same information? One is computed on squares of differences and therefore has a positive sign; the other is computed on differences and therefore has a sign. Wouldn't the bias suffice? If you keep both, please explain why they are both needed and how they are different.

How are the results evaluated? What is the gold standard for soil properties? Pedotransfer functions? What is it for soil moisture? At some of the sites extensive networks of TDR probes exists. Would it be possible to include TDR data in the evaluation? While TDRs are hardly the gold standard in soil moisture measurements, they would provide independent soil moisture data. By the same token, have soil properties been measured at some of the sites? Using the pedotransfer functions to derive unsaturated hydraulic conductivities is hardly the gold standard, and hard to defend. I suggest that the authors make at least some effort to provide independent data on soil moisture and hydraulic properties.

TECHNICAL CORRECTIONS

Please, see the annotated pdf manuscript.

Please also note the supplement to this comment:
http://www.hydrol-earth-syst-sci-discuss.net/hess-2016-432/hess-2016-432-RC4-supplement.pdf

**Supplement:**

[revised manuscript text omitted]

---

## Author Comment (AC2) · 10 Oct 2016

Text formats:

**Reviewer – bold, non-italic**

Answer of the author – non-bold, non-italic

*Cited text – non-bold, italic*

**Anonymous Referee #2**

**GENERAL COMMENTS**

**The authors present an interesting and important study on investigating the benefits of integrating CRNP in data assimilation to improve atmospheric/land surface modeling. While I am not a data assimilation expert, this clearly seems to be a path forward on showing the importance and utilizing long-term monitoring networks for societal benefits. This is a novel study on using a network of CRNP to improve catchment water and energy balance. The authors show the utility of a network on CRNP on improving SWC states and soil parameter estimates in areas with poor or low meteorological coverage and soil information. While the paper is generally well written the authors missed some key references to put this work into proper context. In particular, the recent paper on the Plumber experiment of LSMs (Best 2015) and follow up paper on information content (Nearing 2016) should be discussed in light of this papers major findings. With these additional modifications the paper is appropriate for publication in HESS.**

Thank you for your positive evaluation and reviewing this manuscript. The additional references will provide valuable new aspects to the discussion of the results of this manuscript. In the revised manuscript, we will include the suggested references.

**Major Comments.**
**1. The recent paper by Best (2015) on the plumbing of LSMs needs to be discussed in the introduction and discussion. In addition, the follow up paper by Nearing (2016) on discussing the information content of LSMs is critical. Most notably, their findings on the importance parameterization, model physics, and boundary conditions affecting the partitioning of sensible and latent heat, and comparisons between a SWC benchmark are important. The authors need to discuss these results and how their findings agree or disagree with Best (2015) and Nearing (2016). (e.g. Pg. 3 L 12, Pg. 17 L 13, conclusions). Without this it is hard to place this work in its proper context for critical evaluation. 2. The work of Avery (2016) should also be discussed given the importance of soil and vegetation parameters discussed in this manuscript. This and point 1 will help update the referencing to be most up to date. (e.g. Pg. 10 L 25, Pg. 18 L 18).**

Thank you. The publications by Best et al. (2015) and Nearing et al. (2016) clearly show a strong relation to the experiment set up, results and discussions of the submitted manuscript. Similar to our study, these papers deal with soil moisture and latent heat flux as key variables in land surface modeling and will be addressed and discussed accordingly.
We will also include the reference of Avery et al. (2016) appropriately in the manuscript.

**Minor Comments:**

**Pg 4. L 6. The cases illustrate a way. . .**

**Pg 4. L 15-16. Sentence is awkward please revise.**

Thank you. This will be corrected and revised.

**Referecnes:**
**Avery, W., C. Finkenbiner, T. E. Franz, T. Wang, A. L. Nguy-Roberston, A. Suyker, T. Arkebauer, and F. Munoz-Arriola. 2016. Incorporation of globally available datasets into the roving cosmic-ray neutron probe method for estimating field-scale soil water content. HESS 20: 3859-3872. doi:10.5194/hess-20-3859-2016.**

**Best, M. J., G. Abramowitz, H. R. Johnson, A. J. Pitman, G. Balsamo, A. Boone, M. Cuntz, B. Decharme, P. A. Dirmeyer, J. Dong, M. Ek, Z. Guo, V. Haverd, B. J. J. Van den Hurk, G. S. Nearing, B. Pak, C. Peters-Lidard, J. A. Santanello, L. Stevens, and N. Vuichard. 2015. The Plumbing of Land Surface Models: Benchmarking Model Performance. J. Hydrometeorol. 16:3: 1425-1442. doi:10.1175/jhm-d-14-0158.1.**

**Nearing, G. S., D. M. Mocko, C. D. Peters-Lidard, S. V. Kumar, and Y. L. Xia. 2016. Benchmarking NLDAS-2 Soil Moisture and Evapotranspiration to Separate Uncertainty Contributions. J. Hydrometeorol. 17:3: 745-759. doi:10.1175/jhm-d-15-0063.1.**

---

## Author Comment (AC3) · 14 Oct 2016

Text formats:

**Reviewer – bold, non-italic**

Answer of the author – non-bold, non-italic

*Cited text – non-bold, italic*

**Anonymous Referee #3**

**GENERAL COMMENTS**

**The presented manuscript applies time-series data from 9 cosmic-ray neutron stations to the land-surface model CLM in the Rur catchment. The authors assimilate the data using an ensemble Kalman filter technique to update states and parameters of their model. The added value of training data in years 2011 to 2012 is assessed (1) by testing the model performance in year 2013, (2) by testing the model adaption capabilities to an invalid soil map, and (3) by jackknifing single stations from the training period. The application of the cosmic-ray neutron method in large-scale models is one of the challenges in state-of-the-art hydrology and thus the present study is worth to be published in the scope of HESS after major revision.**

Thank you for your positive evaluation. We sincerely appreciate your time reviewing the manuscript. In the forthcoming revision we will carefully address your suggestions and comments to further strengthen the manuscript.

**1 Evaluating the overall quality**
**Large parts of the manuscript are written in the style of a protocol, by listing lots of other publications who used similar approaches, by mentioning tools that were used, and by reporting every step of the performed analysis. However, I believe that scientific articles should be entitled to challenge their own strategy by discussing alternative methods, by justifying their selection of tools and decisions, and by explaining the corresponding implications. I would thus recommend to rewrite and extend major parts of the introduction and method section. Therein, some literature reviews are unnecessary and probably unrelated to the study and can be omitted or need further explanation (see line-by-line comments). Here, I would suggest to follow the guideline that cited papers should be discussed, and not just mentioned. Other parts concerning the data integration and the SWC model need to be described in more detail. I would further recommend to reduce the detail of the results section, which is hard to follow without proper discussion, and thus to merge it with the discussion section. The results of the study were well structured and decribed, but are not entirely novel, and not sufficient to provide answers to all research questions raised by the authors. For example, to assess the value of a CRNS network of certain density to the performance of a land-surface model, various fractions of the 9 stations should be tested as requested by Referee #1.**

We will shorten and sharpen the indicated parts of the manuscript to further increase its readability. Where appropriate, we will add additional discussion to the mentioned references. In the revised version we will state more clearly, how the research questions were answered by this study. As pointed out in the answer to Referee #1, new simulations will be run with a more limited number of cosmic-ray neutron sensors (4) to emphasize the additional value of a network of CRNS using five CRNS for evaluation.

**Furthermore, the study uses some questionable assumptions, like a constant error of soil moisture data (although neutron measurement uncertainty highly varies with wetness condition), or the assimilation of SWC data assuming homogeneous vertical profiles and no changes of seasonal biomass (see line-by-line comments for details). Another questionable approach is to allow static site-specific parameters to be variable in time, e.g., hydraulic conductivity or soil porosity. This is highly counter-intuitive and should be discussed with respect to uncertain data and/or model conceptualization. I agree to most of the general comments made by Referee #1 and will thus compensate the previous reviews by detailed line-by-line comments below.**

In the revision we will address the mentioned assumptions either when stated in the experiment set up or in the discussion section, if this has not been done already.

**2 Line-by-line comments, scientific questions/issues, and technical corrections**
Thank you for the detailed review of the manuscript and the line-by-line suggestions. We will address and change the line-by-line comments accordingly in the revised version of the manuscript.

**• Title ("Evaluating the value of a network of cosmic-ray probes for improving land**
**surface modelling"): I'd suggest to remove "the value of" to simplify the title. Furthermore, state observations usually do not improve a model, they rather improve model results, e.g. predictions.**
Thank you for this suggestion. We agree and suggest to change the title towards:
"Evaluation of a cosmic-ray neutron sensor network for improved land surface model prediction"

**Page 2**
**• L1: "Land surface models can model": bad phrasing, replace "can model" e.g. by "describe".**
This will be changed.

**• L3: "CRP", please use the newly accepted abbreviation CRNS (cosmic-ray neutron sensing/sensor) with regards to the recent 5th COSMOS workshop.**
We agree and will change this.

**• L14: improve readability, split in two sentences.**
This will be changed.

**• L18: please add a statement about the impact of your findings for the scientific community.**
This will be added.

**• L21-22: this sentence needs a reference.**
This will be added.

**• L24: " and is "→ "while it is"**
This will be considered.

**• L27-28: the given number of references here appears to overwhelm the statement and its low relevance to your paper. Please use only the 1 or 2 most important citations.**
This will be addressed.

**• L30-31: Please discuss the alternatives in more detail to strengthen your decision to use CRNS technology. Were space-borne remote-sensing products assimilated to LSMs before? Why wasn't it successful? What about the use of airborne products with higher resolution and depth? You could also mention point-scale or large-scale soil moisture monitoring networks which have been used for evaluation of land surface models.**
This will be addressed and we will discuss in more detail the limitation of assimilating remotely sensed soil moisture products.

**• L31: "not reliable for areas with dense vegetation": a paper by the same first author recently found that CRNS is also influenced by dense vegetation. Is it more reliable?**
Yes. CRNS is influenced by vegetation and more reliable. This will be addressed.

**• L33: the selection of citations for this statement appears to be random/unrelated. If you want to provide references for the "intermediate scale", Zreda 2008 and Köhli 2015 might be appropriate.**
This will be addressed.

**• L34: "desired application scale of land surface models": please make the reader happy by finally providing concrete information. What is the scale? Are you talking about centimeters or lightyears? Please do not use citations inflationary and do not keep them untouched. How do the three citations help you to support your argumentation?**
We will provide the concrete information earlier in the revision.

**Page 3**
**• L1: omit "fast" as it repeats with the next sentence.**
Okay.

**• L3: add "fast" to make clear that the sensor measures the non-moderated neutrons.**
We will change this-

**• L4: "15 ha", your SWC range seems to be 10 to 40%, which leads to an approximate CRNS footprint of 7 to 14 ha following Köhli 2015, excluding vegetation and altitude influence. You could write "maximum area of 15 ha" to circumvent mentioning this variability.**
Thank you for the suggestion. We will change this.

• **L6: omit "Desilets and Zreda, 2013" as it does only marginally address heterogeneous averaging. Franz 2013a is already a great reference to this topic, Köhli 2015 also touched this.**
We will address this comment.

• **L8: Bogena et al. 2013 did not perform simulations to the penetration depth. Instead, Franz et al. 2012 (doi:10.1029/2012WR01187) and Köhli et al. 2015 provided simulations that both support these values.**
We will address this comment.

• **L11: add a reference for COSMOS-UK, Evans et al. 2016, 10.1002/hyp.10929**
Thank you. We will add this new reference.

• **L13-15: please rephrase to make clear what data assimilation is and is not.**
We will change this.

• **L15: It is not clear why you choose EnKF. Please at least mention other techniques and provide reasons for your choice. The sentence further should be moved to the end of the paragraph after you have introduced the history of DA.**
Thank you, we will add this.

• **L16-34: This historical overview appears to be unnecessary in the context of your study. Neither do you explain what things like " four-dimensional variational DA" are, nor is the relation to your work described. Furthermore, citations are used inflationary again. Please reduce this paragraph to the key publications which support your study. Also think about moving certain studies about ensemble size, multiple time steps, and other filtering approaches to the methodology section, where you need justification for your approach.**
Thank you. We will add motivation and consider moving the overview to the methodology section. This is not clear yet.

• **L30-32: Just to emphasize the previous comment, these lines particularly carry no information for non-experts due to the lack of explanation.**
Thank you. Same as above. We will make this more clear in the resubmission.

**Page 4**
• **L1-19: As stated before, the whole literature review appears to be random and irrelevant to your work. Or at least the relations are not explained. For example, work from Montzka 2011;2013 and Han 2014b appear to be of some relevance for you, prior to others.**
We will restructure the literature review and explain this better. Nevertheless, we disagree that the literature review is random or irrelevant and think that the large majority of the cited papers is relevant. This will be motivated better in the revision.

• **L23-24: "Its capability to propagate surface soil moisture information into the deeper soil column was analyzed by Rosolem et al. (2014)", what does this sentence mean?**
We will clarify this in the resubmission.

• **L26: "The COSMIC operator", third repetition as a sentence starter.**
We will reformulate sentences here.

• **L27-29: combine those sentences: "neutron observations have been used to update states (. . . ) and hydraulic parameters (. . . )"**
The sentences will be combined.

• **L29: "showed" → "demonstrated"**
We will revise the use.

• **L29-30: be more correct in phrasing. Rephrase that Villarreyes 2014 used a different model, but also estimated hydr. parameters by inversion. Han 2016 did so too, using support from neutron data, but neutron assimilation alone does not "update" a hydraulic parameter.**
Thank you. We will rephrase these sentences to be more precise.

• **L31: "This work further explores", omit "further". Until now it is not clear what this work does, you only told stories about work of others. Please summarize which of the presented approaches you are picking up and what scientific novelty you add.**
We will clarify this in the revision.

**Page 5**
• **L3-4: "the soil moisture characterization at the larger catchment scale", what exactly is meant by these terms, and how do you measure improvement?**
We will specify this in the revision.

• **L4: "how dense the CRP network should be", do you answer this question?**
We will specify to which extent this question is answered and what are the limitations to further answer this question.

• **L7-8: "soil maps and atmospheric forcings show spatial correlations over larger distances", this is an interesting point, please provide reference. Isn't the largescale heterogeneity of soil maps only an artefact of soil data scarcity?**
We will provide a reference and think the large scale correlation is not an artefact.

• **L9: "10 stations", do you assimilate all 10, or just 9?**
In fact we assimilate 9 stations. This will be clarified/corrected.

• **L15: "feasibility of the updated large scale soil hydraulic parameters", how can a parameter be feasible? Please clarify your novel research question.**

We will rephrase the sentence to become clearer.

**• L18-19: The sentences can be omitted as being obvious.**
The sentence introduces the topics of the following paragraphs. As such, the sentence is not redundant, but may seem obvious for some readers.

**Page 6**
**• L6: correct wording, a "process" can not be "solved"**
This will be corrected.

**• L10: "Oleson et al. (2013) provide further details on CLM4.5", redundant information with regard to L5-6.**
We will remove this sentence.

**• L10-12: provide reasons why you artificially limit the scope and complexity of your study. What process would a "biogeochemical module" have added and why are they not important here compared to a prescribed LAI?**
This would be beyond the scope of this module as invoking the biogeochemical module would require a mode spin-up of 1000 years for the catchment.

**• L14: please finally (after lots of references in the introduction) provide concrete information about the grid size in your study (the reader is still lost between centimeters and lightyears)**
We will explicitly state the grid cell size. It is however common practice to introduce the grid cell size after the introduction and after the methodology.

**• L23: use standard format for functions, $k[z] \rightarrow k(z)$**
We will change this.

**• L24: format $z \rightarrow z$**
This will be modified.

**• L24: what is the difference between "soil moisture" and SWC? Why are you using the expression $\theta$ here, while SWC is used elsewhere (e.g., eqs. 24 and 25)?**
Thank you. Agreed. We will change the term SWC in Eqs. 24 and 25.

**• L25: use the more convenient expression $k_{sat}(z)$,**
We will do this.

**• L26 (eq. 1):**
**– format $k[z] \rightarrow k(z)$,**
**– rewrite $k_{sat,z} \rightarrow k_{sat}(z)$ as this is a functional relationship. In contrast, indexing a state variable $\theta_i$ is ok.**
**– omit occurrences of 0.5 since $\frac{0.5}{0.5} \approx 1$,**

– case conditions (e.g., $1 < i < N \ldots$ ) are usually preceded by a comma ineach line

– the curly bracket on the right is not common in multi-case equations.

Thank you. We will recheck the formulations.

**Page 7**

**• eq. 3 and 5: reformat** *sand* → sand**, same for clay.**

We will format the terms as suggested.

**• L6: "whereas", split sentence here.**

Thank you. We will split the sentence.

**• eqs. 9 and 10: this is a single equation, requiring only a single equation number, and a multi-case alignment using a curly bracket**

We will modify this as suggested by the reviewer.

**• L12: reformat** *mm* → mm**,**

This will be corrected

**• whole page: please motivate the reader why these details are important for your research question. Also provide information where all these empirical (fixed) parameters (or regression coefficients) are coming from. Is the underlying theory so well understood that no uncertainties or further dependencies are required?**

This is important and we will formulate the motivation more clearly to inform the reader on the use of these pedotransfer functions. Here we will point out uncertainties and further readings which are related to the uncertainty of these particular pedotransfer functions.

**Page 8**

**• L4: "COSMIC parameterizes interactions". The interactions are parameterized by the underlying physical cross-section data. COSMIC rather parameterizes the neutron transport.**

Thank you. We will be more precise in the description of COSMIC.

**• L7-8: Repetition from the introduction.**

We will remove the sentence.

**• L10: "high energy neutrons are reduced" → "the number of high energy neutrons is reduced"**

We will modify this as suggested.

**• L11: "with less energy in each soil layer", misleading/unphysical. Fast neutrons typically evaporate with constant energy.**

We will modify this.

**• L12: rewrite "soil interaction", as fast neutrons predominantly interact with the water.**

Agreed. It will be changed to something like "soil moisture interaction".

**• L16-22 and eq. 14: this part can be omitted, since it is already well described in papers from Shuttleworth and Baatz, and does not add to the message of this paper. If you decide not to omit it, replace $\theta$ in eq. 14 to avoid confusion with soil moisture.**
Thank you. We will replace $\theta$ accordingly.

**• L22: explain to the reader how the 300 soil layers in COSMIC communicate with the 10 soil layers from CLM.**
We will explain this in the revised version.

**• L26: what is a "COSMIC soil surface"?**
We will rephrase or explain this in the revised version.

**• L25ff: it looks like you are not assimilating neutrons, but reiterating SWC from neutron data. The whole paragraph creates a great confusion about what the difference is between SWC, CLM SWC, weighted CLM SWC, and CRP SWC. In contrast to other less relevant paragraphs in this section, this part is highly unclear and simultaneously highly important to understand the most important part of your model. Please rephrase the whole paragraph and clarify to the reader what exactly you do, and why (i.e., why not assimilating N directly?)**
Thank you. Obviously this paragraph needs to be revised. We will rephrase the paragraph to make clearer what is assimilated and how COSMIC and CLM communicate.

**Page 9**
**• L5-10: Your paper is not a protocol. Again, it is described what you are using and who else used it, but the reader is left with the question why you (and others) made this decision. Shortly explain advantages of your strategy and why it serves your research question better than others.**
Thank you. We will put the motivation and explanation forward.

**• L11: what is $f$ in $\vec{x}^f$ ?**
We will add an explanation. $f$ marks the model state at which data is assimilated whereas $a$ marks the new model state after assimilation.

**• L19: confusing typesetting. Is it $\vec{H}$ as a function of the COSMIC model, or is $\vec{H}$ identical with the COSMIC operator?**

The sentence will be reformulated to clarify this.

**• eq. 19: do not use T as a symbol for transposition, there is a reserved symbol for this: $\vec{Y}^{\top}$.**
Thank you. We change T to the reserved symbol.

**Page 10**

**• L18: redundant sentence.**
The sentence will be deleted.

**• L20: why these values? is it comparable with the catchment-mean texture? If your question is, what impact a rough and uncertain soil map in data scarce region would have, wouldn't it be more reasonable to smooth out the existing soil map to a very rough degree, rather than using a completely arbitrary soil map?**
Thank you for your questions. We will do calculations with a third soil map for the revised version of the manuscript (see responses to reviewer #1).

We will also add information on catchment wide average sand and clay content, and what was the motivation to select the biased soil map as initial soil map in part of the simulation experiments:

*"The BK50 soil map provides the initial high resolution soil texture for the catchment and is the most detailed soil map available for the defined region. Average sand and clay content of the catchment are 22.5% and 21.4%, respectively. As an alternative, simulations were also performed for a biased soil texture distribution with a fixed sand content of 80 % and clay content of 10 % (S80 soil map). This represents a large error with respect to the expected soil properties. However, perturbations of 10 and 30 % guarantee some variability in the initial soil properties. The S80 soil map simulations allow evaluating the joint state-parameter estimation approach because given the expected bias, we can evaluate whether and to what extend the soil properties are modified by the data assimilation to be closer to the available high resolution soil map."*

**Page 11**
**• L3-5: omit physical units (they are irrelevant in this context).**
We will omit physical units here.

**• L8-11: How do you justify the perturbation of physical soil parameters like porosity and texture? Does the uncertainty of the soil map justify the huge variation ranges applied in this work? Are models allowed to adapt their physical basement to hydrological data (which also show uncertainty)?**
Concerning your first question: little information is available, but tests were made with different degrees of uncertainty and the sensitivity of the model outcomes to the imposed perturbation was limited. Concerning your second question: this is a standard procedure in data assimilation and we can refer to numerous papers where this strategy was followed. We will provide additional motivation in the revised version of the manuscript.

**• L15: omit "="**
We agree.

**• How was the CRP SWC uncertainty determined? Assuming a constant CRNS error is not physical and might have substantial influence on the results (to be tested). For example, the error of neutron observations N is pN, while N can almost double from very wet to very dry conditions, which leads to a**

**variation of the neutron uncertainty by 30%. This can propagate through the non-linear relation to soil moisture in such a way that your observed SWC is significantly more uncertain in wet periods compared to dry periods. Consequently, the DA approach should give more weight to dry periods during assimilation.**

This is an interesting point. We will add explanation in the manuscript and acknowledge the approximation that was made here. The error you are referring to is the Poisson noise in the measurement. Averaging measurements over a sufficiently long time windows (24 hours in our case) reduces the error in the measurement substantially. Naturally, not all sources of uncertainty can be identified and ruled out. Previous work has shown that there is still an error in the SWC estimation by CRNS (Baatz et al. 2014). Based on this work we assume a fixed error of 0.03 cm3/cm3.

**Page 12**
**• L16: Why do you use RMSE, although many alternative measures are accepted as state-of-the-art measures for time series evaluation, e.g., KGE or NSE, in order to assess bias, deviation, and correlation simultaneously?**

RMSE is a standard measure used in the data assimilation community, whereas NSE is not standard.

**• eqs. 23 and 24: reformat** *SWC* → SWC**, same with** *RMSE* **and** *bias***, as those are single multi-letter variables, not products of multiple single-letter variables. Following this style guide, rewrite** $E_{RMS}$ → $E_{\mathrm{RMS}}$**. You can even omit "RMS" since** E **is the only error used in this work. This would improve readability of the results section.**

Thank you for the suggestion. We will reformat the equation and adapt the equation to the style guide.

**Page 16**
**• L17-26: It is argued that changes in SWC states have impact to simulated ET flux. However, only for state-parameter updates (L19). Why is ET not affected by (SWC) state updates only?**

This .

**• L32: "to** $E_{RMS}$**-values", omit "-"**

Thank you, agreed.

**Page 17**
**• L9: if precipitation data from COSMO_DE was used, why was this information omitted in the method section (only mentioning DWD)?**

This information will be added in the revised version of the manuscript.

**• L26: replace "fast" with "quickly".**

Thank you.

**Page 18**
**• L10-23: This question already needs an answer in the method section, I'd suggest to move the whole paragraph.**

We agree and we will integrate this paragraph in the method section.

**• L10-23: I cannot follow the argumentation. Baatz et al. 2014 suggested a correction function for neutron counts based on vegetation estimates. In your model, you already have LAI data every month, implementation of the correction functions in the model would probably be straight forward. Furthermore, to convert neutron data to SWC, some vegetation correction would be necessary, too. Third, assimilating CRP SWC assumes homogeneous vertical SWC profiles (before iteration), this assumption would be unnecessary if neutrons would be assimilated directly. I am afraid that this topic is more complex and needs further discussions and tests. It would be most convincing if you could show that neutron assimilation indeed gives different results than CRP SWC assimilation.**

We will clarify why leaf area index data is not sufficient to estimate biomass in a catchment. There is no vegetation correction necessary to convert neutron data to SWC. We did on-site calibration which circumvents the need for a vegetation correction if e.g. a rover is applied in a region with high biomass differences (e.g. forest and crops).

**• L27: "neutron flux intensity", do you mean flux or intensity or both?**
Thank you. We will remove "flux".

**• L27: "Although . . . only available at few locations", write more positively. Neutron data was available at up to 9 locations, which was intended to be the amazing novelty compared to other catchments!**
Thank you. We will rephrase the sentence.

**Figures**
**1. South → south, same with North.**
This will be changed.

**2. Please add grid lines**
We will add horizontal grid lines.

**3. Please add grid lines**
We will add horizontal grid lines.

**4. Please add grid lines**
We will add horizontal grid lines.

**5. It is hard to distinguish two black lines with different meaning. Further indication of the expected "true" sand content (given by the soil map or soil samples) would be helpful to evaluate these plots.**
We will add the sand content of the BK50 soil map.

**6. This figure is not understandable without the text. Please shortly provide information about the B parameter in the caption to understand the message of this figure.**
We will add information to the caption.

**7. replace** k(sat) → $k_{sat}$**. It would be interesting to also show the evolution of the soil porosity parameter together with an indication of its measured value. Why does hydraulic conductivity (and probably also porosity) vary over time at individual sites? Those are expected to be constant physical parameters of the sites. In my opinion this is a serious flaw of the DA approach used here.**

Thank you. This issue is well known. This is a standard procedure in data assimilation and we can refer to numerous papers where this strategy was followed. We will provide additional motivation in the revised version of the manuscript. Notice also that simulations were made for a verification period with constant parameters (estimated in the assimilation period).

**8. The purpose of this figure is not clear, as no observation data is provided to evaluate the model performance with respect to simulated latent heat.**

Thank you. The issue you point out is in agreement with the desire of referee #1 for additional latent heat flux observations. We will add or modify Figure 8 in the revision with some LE observations in comparison to modeled ET.

**Tables**

**1. what is C3? Replace "non arctic" with "non-arctic", probably add a citation to the caption for plant functional types.**

Thank you. We will add information, a reference for plant functional types and replace "non arctic".

**3. improve readability by increasing font weight (boldness) for particularly good cases below an RMSE threshold, which is a common strategy in many journals.**

Thank you for the suggestion. We will mark particularly good cases with bold fonts.

**4. same as 3. Rephrase the last sentence.**

Thank you. We will rephrase the sentence.

**5. same as 4.**

Thank you. We will rephrase the sentence.

---

## Author Comment (AC4) · 19 Oct 2016

Q: GENERAL COMMENTS OVERALL QUALITY The paper by Baatz et al. (2016-432) describes an effort to use soil moisture data from nine closely-spaced (2000 km2) cosmic-ray probes (CRPs) with data assimilation scheme to improve the assessment of soil moisture in land-surface models. The goal is worthy and the execution is thorough. The results are significant: (1) the joint state (soil moisture) and parameter (soil properties, like sand percentage) estimation within data assimilation scheme produces better results than just state estimation; (2) in absence of soil data and meteorological data, CRPs alone can improve data assimilation results. On that account, the paper is suitable for publication in HESS.

A: Thank you for the positive overall evaluation of the manuscript. We appreciate your

effort in reviewing the manuscript. In the forthcoming revision we will consider each of your suggestions and expand the discussion to answer the questions raised.

Q: However, I am less certain about the significance of these results in light of the finding that the parameters change in time and in many cases never converge. This is only possible if the parameters are fitting parameters rather than physical parameters. So we end up with better results, but possibly only by statistical manipulation rather than by improved understanding of the physics. Is this progress? I would like to see at least some discussion of this issue in the paper in its final form.

A: We will expand the discussion on this issue in the revised version of the manuscript. This is a standard procedure in data assimilation and we can refer to numerous papers where this strategy was followed. We will provide additional motivation in the revised version of the manuscript. Notice also that simulations were made for a verification period with constant parameters (estimated in the assimilation period).

Q: SPECIFIC COMMENTS Why are RMSE and bias discussed separately if they are essentially the same information? One is computed on squares of differences and therefore has a positive sign; the other is computed on differences and therefore has a sign. Wouldn't the bias suffice? If you keep both, please explain why they are both needed and how they are different.

A: We will add some clarifying remark on the differences and the advantage of using both measures, RMSE and bias.

Q: How are the results evaluated? What is the gold standard for soil properties? Pedo-transfer functions? What is it for soil moisture? At some of the sites extensive networks of TDR probes exists. Would it be possible to include TDR data in the evaluation? While TDRs are hardly the gold standard in soil moisture measurements, they would provide independent soil moisture data. By the same token, have soil properties been measured at some of the sites? Using the pedotransfer functions to derive unsaturated hydraulic conductivities is hardly the gold standard, and hard to defend. I suggest that

the authors make at least some effort to provide independent data on soil moisture and hydraulic properties.

A: Thank you for the suggestions. TDR probes work on a scale of few dm3 which is significantly smaller than the scale of the land surface model (1 km2). Hence, TDR probes are not suitable for a direct evaluation whereas cosmic-ray neutron sensors (CRNS) measure soil moisture at an equivalent scale. We note that soil moisture measurements by CRNS were already independent from the prediction of the land surface model during the verification period. Soil hydraulic properties at the desired scale were not measured. However, we will provide additional data on latent heat flux observations for comparison with predicted latent heat flux.

Q: TECHNICAL CORRECTIONS Please, see the annotated pdf manuscript.

A: Technical corrections were addressed in the annotated pdf manuscript.

Please also note the supplement to this comment:
http://www.hydrol-earth-syst-sci-discuss.net/hess-2016-432/hess-2016-432-AC4-supplement.pdf

**Supplement:**

[revised manuscript text omitted]

---

## Author Response (AR2)

Response to the referee comments on the manuscript originally titled "Evaluating the value of a network of cosmic-ray probes for improving land surface modelling" by Baatz et al. on behalf of all co-authors submitted to *Hydrology and Earth System Sciences – Discussion*.

We thank the Editor for his guidance through the revision process.

We thank the reviewers for their time reviewing the manuscript and their suggestions for improving the manuscript. We appreciate their constructive comments and changed the manuscript in the following main points:

- All comments of the reviewers were addressed throughout the manuscript
- Three new experiments with four instead of nine cosmic ray neutron sensors were conducted
- Two new experiments with the FAO soil map were conducted
- These additional five experiments were addressed/included in Abstract, Introduction, Methods, Results & Discussion, Conclusion
- "Discussion" was merged into results section; now "Results and discussion"
- The Tables with detailed site-wise results were moved into the Annex, and two shorter summarizing Tables were included instead. This reduces length and increases readability of the results section.
- Introduction and Results were shortened, as suggested by the reviewers
- Discussion was improved including results on evapotranspiration and new important references

The title of the manuscript was changed as suggested by Referee #1 and #3 to "Evaluation of a cosmic-ray neutron sensor network for improved land surface model prediction".

Sincerely,

Roland Baatz and co-authors

Text formats:

      **Referee – bold**

      Answer of the authors – non-bold

**Reviewer 1**

**GENERAL COMMENTS**

**The manuscript is well written and clear. The topic is of interest for the HESS readership as cosmic-ray probes represent a relatively new technology for ground measuring soil moisture over large areas. Therefore, we need to assess the impact of this new technology for improving land surface modelling. The paper describes several assimilation experiments in which soil moisture data from cosmic-ray probes are used for improving soil moisture modelling through CLM land surface model. Results are (quite) well described and clearly structured. However, in my opinion, several aspects should be improved/changed before the publication. I reported below a list of the general comments to be addressed with also the specification of their relevance.**

Thank you for your positive evaluation and for your time reviewing this manuscript. We carefully address your comments in the following.

**1) MAJOR: Some of the results shown in the paper are well-known. I am aware that it is important to show real-world experiments, mainly by considering new technology, but the main results given in the paper were already reported in several previous studies: a) the assimilation of ground-based soil moisture data is able to improve soil moisture modelling, b) the joint state-parameter assimilation is better than the state assimilation only, and c) the assimilation is more effective when soil texture information are wrong (i.e., there's larger room for improvement). I believe that the paper results need to be published, but I would like to see some new findings that can be obtained by using the same material (data and modelling) presented in the paper.**

Thank you, we understand that the novelty of this paper needs to be more clearly pointed out. The main novelty is the use of the cosmic ray probe to measure soil moisture at the intermediate scale and update soil moisture at the larger catchment scale. Although it is true that points a, b and c have been demonstrated in other papers, this has never been done for the cosmic ray probe at the larger catchment scale. We added now begin the abstract with:

Page 1 line 5:"In this study, the potential of a network of CRNS installed in the 2354 km$^2$ Rur catchment (Germany) for estimating soil hydraulic parameters and improving soil moisture states was tested."

CRNS, catchment wide, used for parameter updates, this is clearly novel.

Another key finding was pointed out in the abstract, where we quantified the possible improvement of soil moisture prediction using a land surface model:

Page 2 line 11-14: "For the FAO soil map and the biased soil map soil moisture predictions improved strongly to a root mean square error of 0.03 $cm^3/cm^3$ for the assimilation period and 0.05 $cm^3/cm^3$ for the evaluation period. Improvements were limited by the measurement error of CRNS (0.03 $cm^3/cm^3$). "

Demonstrating a way to propagate CRNS measurements into horizontal space using the local ensemble transform Kalman Filter is novel. The improved parameterization in catchment space was not demonstrated before. Therefore we state in the abstract:

Page 2 line 15-17: "The results demonstrate that assimilated data of a CRNS network can improve the characterization of soil moisture content at the catchment scale by updating spatially distributed soil hydraulic parameters of a land surface model."

**For instance:**

**A) What are the results if only one (or two) cosmic-ray probes are assimilated? In the real world it is expected that the number of probes will be limited and, hence, the use of a limited number of probes is surely of great interest.**

Thank you for the suggestion. Many further (synthetic) studies are possible, but a publication generally has limited space available. However, we add to the manuscript three additional simulation experiments with a smaller number (four) of cosmic ray neutron sensors for assimilation. This allows verification at the five remaining cosmic ray neutron sensors. Please see the manuscript for further details.

**B) What is the impact in terms of fluxes? In section 4.6 a comparison of annual evapotranspiration maps without and with the assimilation is carried out, but simply showing that the resulting maps are different.**

Thank you. We agree that the figure demonstrates an impact of soil moisture and soil parameter updates on latent heat flux during the evaluation period. We added more information to the caption:

Page 35 line 1:"Annual evapotranspiration (ET) is shown in the year 2013 (evaluation period, no assimilation). This figure demonstrates the impact of parameter updates (PAR-S80-10 and PAR-BK50-10) in comparison to open loop (OL-S80) and reference soil map (OL-BK50). ET changes in the North but not as much in the South."

**However, it is obvious that changing soil moisture will change evapotranspiration. Is it possible to perform an independent validation by using data about the actual evapotranspiration in the basin? Or likely by using discharge observations? I believe that some new results should be included in the paper (even though I am aware that authors are usually reluctant to perform additional analyses). Moreover, the results in terms of soil moisture simulation should be synthetized (see Comment 5).**

We agree that a comparison to evapotranspiration (ET) observations would be beneficial. However, a comparison of catchment ET or observed ET at a representative number of sites is beyond the scope of this paper because uncertainties in upscaling of ET are large. Impacts on ET are discussed in a paragraph:

Page 20 line 25-page21: "Additionally, the impact of soil parameter estimates on ET is different in the North of the catchment compared to the South. While ET in the North of the catchment was impacted by

the estimated soil properties during the evaluation period 2013 for PAR-S80-10, ET in the South was not as much impacted by estimated soil properties. This is related to the fact that in the North ET is moisture limited in summer, whereas in the South this is not moisture limited but energy limited. Therefore, ET in the North is sensitive to variations in soil hydraulic parameter values, whereas in the South this is not the case. In the South, ET is sensitive to model forcings like incoming shortwave radiation. Nearing et al. (2016) came to the conclusion that soil parameter uncertainty dominates soil moisture uncertainty and forcing uncertainty dominates ET uncertainty. Our findings in the southern part of the catchment support their conclusion, but in the northern part of the catchment soil parameter uncertainty strongly affect ET. Hence particularly in the northern part of the catchment, further observations such as ET measurements are desirable for further improving the land surface model. These additional observations could be used for future land surface model benchmarking (Best et al., 2015) or for more constrained parameter estimates (Shi et al., 2015)."

**2) MAJOR: The description of the data assimilation experiments should be improved. As usually, a number of subjective choices were made in the setup of the data assimilation experiments, and these choices may have a significant impact on the results. For instance, a fixed error for soil moisture estimates from cosmic-ray probes is considered (0.03 cm3/cm3). Similarly, the perturbation factors for input data (precipitation and shortwave radiation) and parameters (10 and 30%) are arbitrarily selected. A sensitivity analysis on these choices should be carried out. It might be that different choices produce very different results.**

Thank you for these constructive suggestions. We agree that a number of assumptions needed to be made. The assumption on the measurement error for the soil water content was thoroughly evaluated and more details can be found in our earlier papers on this (e.g., Bogena et al., 2013; Baatz et al., 2014; Baatz et al., 2015). We feel that there is no need to repeat experiments with other values for the measurement error given the earlier work:

Page 13 line 24: "Based on previous work (Baatz et al., 2015), the SWC retrieval uncertainty for CRNS was estimated to be 0.03 cm$^3$/cm$^3$ while fluctuations in the measurement standard deviation, related to the non-linear relation between observed neutron intensity and SWC, were assumed negligible."

However, we agree that perturbation of meteorological forcings and soil hydraulic parameters are subject to larger uncertainty. We applied a 10% and 30% perturbation and think covers adequately the uncertainty with respect to texture. Simulations were not strongly affected by the magnitude of the soil perturbations:

Page 15 line 25-28:"The $E_{RMS}$ and bias for simulations with 10 % and 30 % perturbation of soil texture only showed very small differences (smaller than 0.01 cm$^3$/cm$^3$)."

 We feel that the applied perturbations are realistic and the difference in the two applied perturbations was already large.

We also already tested two different magnitudes of perturbation of precipitation, including a 50% and 100% error but this did not affect the simulation results:

Page 13 line 23: "In this work, only results for precipitation perturbation with $\sigma = 0.5$ will be shown as results for $\sigma = 1.0$ were similar."

The simulations are also CPU-intensive. We expect that an extension of the experiments for further magnitudes of perturbation would not supply significant additional insights.

**3) MODERATE: Similarly as above, the selection of one single biased soil texture map is arbitrary. Why only one soil map? Why 80% of sand content and 10% of clay content is selected? What is the average sand and clay content percentage in the basin? Again, a sensitivity analysis is needed. Otherwise, it might be that a very large error in the soil map is used to highlight the positive impact of assimilating soil moisture data. What happens for a less biased map? This aspect should be clarified.**

Thank you for this suggestion. In the revised version, we include a simulation with a third (FAO) soil map that may be close to the expected values of the BK50 soil map. Table 1 shows the sand content at the nine sites for the BK50 soil map. We added information on catchment wide average sand and clay content, and what was the motivation to select the biased soil map as initial soil map in part of the simulation experiments:

Page 12 line 15-18: "Alternative simulations were also performed with the FAO soil map of the global Harmonized World Soil Database (FAO, 2012) and with a biased soil texture with a fixed sand content of 80 % and clay content of 10 % (S80 soil map). Average sand and clay content are 22.5% and 21.4% for the BK50 soil map and 39% and 22% for the FAO soil map. The FAO soil map and the biased soil map represent large error with respect to the soil properties of the BK50 soil map."

The new simulation results with the FAO soil map are discussed in detail.

**4) MINOR: In CLM the subsurface lateral flow is not considered. It has an impact on soil moisture simulation and, mainly, on the capability to modify soil moisture simulations at unmonitored locations. Therefore, I expect that the assimilation of in situ soil moisture data will have a local effect. However, the jackknifing data assimilation experiments show that the assimilation produces significant changes also at unmonitored locations. Why does it happen? I believe it should be clarified in the paper.**

Thank you for this detail. It is correct that CLM does not consider subsurface lateral flow. The updates of soil moisture in space depend on spatial correlations of soil moisture. In this study, spatial correlations of soil moisture are the consequence of spatial correlations of the atmospheric forcings, spatial correlations of soil hydraulic parameters and their interaction with the land surface model. Atmospheric reanalysis data and the soil map provided a good basis for the imposed spatial correlation structure. The imposed

spatial correlation structures on the perturbations determine to a large extend the soil moisture updates in space. We added:

Page 17 lin 27-29: "Spatial improvements are possible by spatial correlation structures of atmospheric forcings, soil hydraulic parameters and soil moisture which are taken into account by the local ensemble transform Kalman filter."

**5) MINOR: In sections 4.2 and 4.5, too many details are provided in the description of the results for each single site. I suggest focusing on the most important results to improve their readability. Also, discussion section is too generic, especially the first paragraph. In the specific comments, I added some corrections and suggestions that should be implemented. On this basis, I believe the paper deserves to be published only after a major revision.**

Results section and discussion section were shortened and sharpened, also taking into account the detailed comments of reviewer #3. We also merged the Results and Discussion section into one section for more fluent reading.

**SPECIFIC COMMENTS (P: page, L: line or lines) Title: The paper, in the current version, demonstrates that the assimilation of soil moisture data from cosmic-ray probes is able to improve soil moisture modelling, not "land surface modelling" (e.g., evapotranspiration or discharge fluxes). Therefore, I suggest changing the title. Abstract: The abstract should include information on the location of the study area and on the employed data assimilation technique.**

Thank you.

We changed the title to:

"Evaluation of a cosmic-ray neutron sensor network for improved land surface model prediction."

We now include the employed data assimilation technique and the location of the study area in the abstract:

Page 2 line 4-9: "In this study, the potential of a network of CRNS installed in the 2354 km$^2$ Rur catchment (Germany) for estimating soil hydraulic parameters and improving soil moisture states was tested. Data measured by the CRNS were assimilated with the local ensemble transform Kalman filter in the Community Land Model v. 4.5."

**P4, L7: Formatting error for Kurtz et al. (2016). Please correct.**

Thank you. We corrected this.

**P5, L19: It should be COSMIC in place of COMIC.**

Thank you. We corrected this.

**P12, L11: Is sigma=0.5 considered for perturbing precipitation in all the assimilation**

**experiments? Please clarify.**

Yes. We agree that this was slightly unclear and moved the sentence to the right position in the text.

**P12, L25: It should be "four data assimilation scenarios" in place of "six assimilation**

**scenarios".**

Thank you. We changed this:

Page 15 line 10: "Presented are results for the open loop scenarios with the BK50, FAO and S80, and data assimilation scenarios."

**P19, L18-19: This sentence is too broad, please modify.**

Thank you. We agree and changed the sentence to:

Page 22 line 6-8: "Hence, this study represents a way forward towards the integration of CRNS information in the calibration or real-time updating of land surface models. "

**Anonymous Referee #2**

GENERAL COMMENTS

The authors present an interesting and important study on investigating the benefits of integrating CRNP in data assimilation to improve atmospheric/land surface modeling. While I am not a data assimilation expert, this clearly seems to be a path forward on showing the importance and utilizing long-term monitoring networks for societal benefits. This is a novel study on using a network of CRNP to improve catchment water and energy balance. The authors show the utility of a network on CRNP on improving SWC states and soil parameter estimates in areas with poor or low meteorological coverage and soil information. While the paper is generally well written the authors missed some key references to put this work into proper context. In particular, the recent paper on the Plumber experiment of LSMs (Best 2015) and follow up paper on information content (Nearing 2016) should be discussed in light of this papers major findings. With these additional modifications the paper is appropriate for publication in HESS.

Thank you for your positive evaluation and reviewing this manuscript. We added two suggested references relevant to the manuscript. Please see below.

Major Comments.
1. The recent paper by Best (2015) on the plumbing of LSMs needs to be discussed in the introduction and discussion. In addition, the follow up paper by Nearing (2016) on discussing the information content of LSMs is critical. Most notably, their findings on the importance parameterization, model physics, and boundary conditions affecting the partitioning of sensible and latent heat, and comparisons between a SWC benchmark are important. The authors need to discuss these results and how their findings agree or disagree with Best (2015) and Nearing (2016). (e.g. Pg. 3 L 12, Pg. 17 L 13, conclusions). Without this it is hard to place this work in its proper context for critical evaluation. 2. The work of Avery (2016) should also be discussed given the importance of soil and vegetation parameters discussed in this manuscript. This and point 1 will help update the referencing to be most up to date. (e.g. Pg. 10 L 25, Pg. 18 L 18).

Thank you. We agree that the study of Nearing et al. (2016) and Best et al. (2015) is related to this work. We add to the discussion:
Page 20 line 31-page21: "Nearing et al. (2016) came to the conclusion that soil parameter uncertainty dominates soil moisture uncertainty and forcing uncertainty dominates ET uncertainty. Our findings in the southern part of the catchment support their conclusion, but in the northern part of the catchment soil parameter uncertainty strongly affect ET. Hence particularly in the northern part of the catchment, further observations such as ET measurements are desirable for further improving the land surface model. These additional observations could be used for future land surface model benchmarking (Best et al., 2015) or for more constrained parameter estimates (Shi et al., 2015)."

We also found the reference of Avery et al. (2016) being useful and added:
Page 10 line 19: "This would require calibration data throughout the catchment which is only feasible using spatially distributed data sets (e.g. Avery et al., 2016)."

**Minor Comments:**

**Pg 4. L 6. The cases illustrate a way. . .**
Thank you. We changed this.

**Pg 4. L 15-16. Sentence is awkward please revise.**

Thank you. We rephrased:
Page 4 line 32-page5: "They showed that the nonlinear character of the soil moisture retention characteristic is critical for joint state-parameter estimation in data assimilation systems and showed that the Particle Filter is an interesting alternative for soil hydraulic parameter estimation for 1D problems."

**Referecnes:**
Avery, W., C. Finkenbiner, T. E. Franz, T. Wang, A. L. Nguy-Roberston, A. Suyker, T. Arkebauer, and F. Munoz-Arriola. 2016. Incorporation of globally available datasets into the roving cosmic-ray neutron probe method for estimating field-scale soil water content. HESS 20: 3859-3872. doi:10.5194/hess-20-3859-2016.

Best, M. J., G. Abramowitz, H. R. Johnson, A. J. Pitman, G. Balsamo, A. Boone, M. Cuntz, B. Decharme, P. A. Dirmeyer, J. Dong, M. Ek, Z. Guo, V. Haverd, B. J. J. Van den Hurk, G. S. Nearing, B. Pak, C. Peters-Lidard, J. A. Santanello, L. Stevens, and N. Vuichard. 2015. The Plumbing of Land Surface Models: Benchmarking Model Performance. J. Hydrometeorol. 16:3: 1425-1442. doi:10.1175/jhm-d-14-0158.1.

Nearing, G. S., D. M. Mocko, C. D. Peters-Lidard, S. V. Kumar, and Y. L. Xia. 2016. Benchmarking NLDAS-2 Soil Moisture and Evapotranspiration to Separate Uncertainty Contributions. J. Hydrometeorol. 17:3: 745-759. doi:10.1175/jhm-d-15-0063.1.

**Anonymous Referee #3**

**GENERAL COMMENTS**

**The presented manuscript applies time-series data from 9 cosmic-ray neutron stations to the land-surface model CLM in the Rur catchment. The authors assimilate the data using an ensemble Kalman filter technique to update states and parameters of their model. The added value of training data in years 2011 to 2012 is assessed (1) by testing the model performance in year 2013, (2) by testing the model adaption capabilities to an invalid soil map, and (3) by jackknifing single stations from the training period. The application of the cosmic-ray neutron method in large-scale models is one of the challenges in state-of-the-art hydrology and thus the present study is worth to be published in the scope of HESS after major revision.**

Thank you for your positive evaluation. We sincerely appreciate your time reviewing the manuscript. We carefully addressed your suggestions and comments to further strengthen the manuscript.

**1 Evaluating the overall quality**
**Large parts of the manuscript are written in the style of a protocol, by listing lots of other publications who used similar approaches, by mentioning tools that were used, and by reporting every step of the performed analysis. However, I believe that scientific articles should be entitled to challenge their own strategy by discussing alternative methods, by justifying their selection of tools and decisions, and by explaining the corresponding implications. I would thus recommend to rewrite and extend major parts of the introduction and method section. Therein, some literature reviews are unnecessary and probably unrelated to the study and can be omitted or need further explanation (see line-by-line comments). Here, I would suggest to follow the guideline that cited papers should be discussed, and not just mentioned. Other parts concerning the data integration and the SWC model need to be described in more detail. I would further recommend to reduce the detail of the results section, which is hard to follow without proper discussion, and thus to merge it with the discussion section. The results of the study were well structured and decribed, but are not entirely novel, and not sufficient to provide answers to all research questions raised by the authors. For example, to assess the value of a CRNS network of certain density to the performance of a land-surface model, various fractions of the 9 stations should be tested as requested by Referee #1.**
We shortened and sharpened the indicated parts of the manuscript to further increase its readability. Where appropriate, we added additional discussion to the mentioned references. In the revised version we state more clearly, how the research questions were answered by this study.
As pointed out in the answer to Referee #1, new simulations were run with a more limited number of cosmic-ray neutron sensors (4) to emphasize the additional value of a network of CRNS using five CRNS for evaluation and with a third soil map.

**Furthermore, the study uses some questionable assumptions, like a constant error of soil moisture data (although neutron measurement uncertainty highly varies with wetness condition), or the**

**assimilation of SWC data assuming homogeneous vertical profiles and no changes of seasonal biomass (see line-by-line comments for details). Another questionable approach is to allow static site-specific parameters to be variable in time, e.g., hydraulic conductivity or soil porosity.**

Thank you. We address these questions exhaustively in the line-by-line comments in the following pages.

**This is highly counter-intuitive and should be discussed with respect to uncertain data and/or model conceptualization. I agree to most of the general comments made by Referee #1 and will thus compensate the previous reviews by detailed line-by-line comments below.**

**2 Line-by-line comments, scientific questions/issues, and technical corrections**

**• Title ("Evaluating the value of a network of cosmic-ray probes for improving land surface modelling"): I'd suggest to remove "the value of" to simplify the title. Furthermore, state observations usually do not improve a model, they rather improve model results, e.g. predictions.**

Thank you for this suggestion. We agree and suggest to change the title towards:

"Evaluation of a cosmic-ray neutron sensor network for improved land surface model prediction"

**Page 2**
**• L1: "Land surface models can model": bad phrasing, replace "can model" e.g. by "describe".**

We changed this to "describe".

**• L3: "CRP", please use the newly accepted abbreviation CRNS (cosmic-ray neutron sensing/sensor) with regards to the recent 5th COSMOS workshop.**

We agree and changed this throughout the manuscript.

**• L14: improve readability, split in two sentences.**

Thank you, we did split the sentence.

**• L18: please add a statement about the impact of your findings for the scientific community.**

The impact on the scientific community can only crystalize after publication. The expected scientific impact is highly speculative and in my opinion not suitable for the abstract.

We conclude the abstract with the key finding summarizing the detailed results of the study in a general way and think this is sufficient for the abstract:

"The results demonstrate that assimilated data of a CRNS network can improve the characterization of soil moisture content at the catchment scale by updating spatially distributed soil hydraulic parameters of a land surface model."

**• L21-22: this sentence needs a reference.**

We added: Brutsaert, W.: Hydrology : an introduction, Cambridge University Press, Cambridge ; New York, xi, 605 p. pp., 2005.

**• L24: " and is "→ "while it is"**

We changed this.

• **L27-28: the given number of references here appears to overwhelm the statement and its low relevance to your paper. Please use only the 1 or 2 most important citations.**
We agree and removed some references of less relevance.

• **L30-31: Please discuss the alternatives in more detail to strengthen your decision to use CRNS technology. Were space-borne remote-sensing products assimilated to LSMs before? Why wasn't it successful? What about the use of airborne products with higher resolution and depth? You could also mention point-scale or large-scale soil moisture monitoring networks which have been used for evaluation of land surface models.**
These methods were successful in some cases, but have deficiencies particularly in respect to dense vegetation as mentioned here:
Page 2 line 31-page3: "Soil moisture measured by space-borne remote sensing technologies provides information over large areas but is strongly affected by vegetation and surface roughness (e.g. Temimi et al., 2014). Therefore, in this paper an alternative source for soil moisture information is explored which can measure soil moisture more accurately under dense vegetation (Bogena et al., 2013)."

We mention successful applications of remote sensing data e.g. here:
Page 3 line 27: "Reichle et al. (2002) performed a synthetic experiment using L-band microwave observations of the Southern Great Plains Hydrology Experiment (Jackson et al., 1999) to analyse the effect of ensemble size and forecast errors."
And here:
Page 3 line 32-page4: "More recently, state updates with the EnKF were tested for the Soil Moisture Ocean Salinity (SMOS, Kerr et al., 2012) mission. De Lannoy and Reichle (2016) assimilated SMOS temperature brightness and soil moisture retrievals into a land surface model with large improvements in surface soil moisture."

Following your previous argumentation on limiting the introduction to relevant publications, we constrain the introduction to cosmic-ray neutron sensors, mentioning alternative techniques. Additionally, we give a limited overview on important literature of earlier work done particularly on methods used in this study.

• **L31: "not reliable for areas with dense vegetation": a paper by the same first author recently found that CRNS is also influenced by dense vegetation. Is it more reliable?**
Yes, CRNS are more accurate under dense vegetation than remote sensing products. We added:
Page 2 line 31: "Soil moisture measured by space-borne remote sensing technologies provides information over large areas but is strongly affected by vegetation and surface roughness (e.g. Temimi et al., 2014). Therefore, in this paper an alternative source for soil moisture information is explored which can measure soil moisture more accurately under dense vegetation (Bogena et al., 2013)."

• **L33: the selection of citations for this statement appears to be random/unrelated. If you want to provide references for the "intermediate scale", Zreda 2008 and Köhli 2015 might be appropriate.**
We state this more precise now:

Page 3 line 3-5: "Cosmic-ray neutron sensors (CRNS) measure fast neutron intensity at an intermediate scale of ~15 ha (Kohli et al., 2015;Zreda et al., 2008) which is the desired application scale of land surface models (Ajami et al., 2014;Chen et al., 2007;Shrestha et al., 2014).".

**• L34: "desired application scale of land surface models": please make the reader happy by finally providing concrete information. What is the scale? Are you talking about centimeters or lightyears? Please do not use citations inflationary and do not keep them untouched. How do the three citations help you to support your argumentation?**
Thank you, we are precise by now stating the observation scale (15ha, see comment before), but the model resolution is stated in the methods section which is common sense in literature of hydrologic modelling.

**Page 3**
**• L1: omit "fast" as it repeats with the next sentence.**
In accordance with your following comment, we prefer to stay clear and keep "fast":
Page 3 line 5-8: "Fast neutrons originate from collisions of secondary cosmic particles from outer space with terrestrial atoms. Fast neutrons in turn are moderated most effectively by hydrogen because the mass of a neutron is similar to that of a nucleus of the hydrogen atom."

**• L3: add "fast" to make clear that the sensor measures the non-moderated neutrons.**
We added "fast".

**• L4: "15 ha", your SWC range seems to be 10 to 40%, which leads to an approximate CRNS footprint of 7 to 14 ha following Köhli 2015, excluding vegetation and altitude influence. You could write "maximum area of 15 ha" to circumvent mentioning this variability.**
Thank you for the suggestion. We see your point, however, the newest research is not always carved into stone and figures are evolving as research continues and is discussed. Take for example the all very recent publications that address the cosmic ray neutron sensor footprint defined to 300m radius (Zreda et al., 2008), 300m radius and less (Desilets and Zreda, 2013, on the footprint specifically) and 200m radius and less (Kohli et al, 2015 on the footprint specifically).
However, this is not a major focus of this study. We changed to:
Page 3 line 4: "intensity at an intermediate scale of ~15 ha (Kohli et al., 2015;Zreda et al., 2008)"

**• L6: omit "Desilets and Zreda, 2013" as it does only marginally address heterogeneous averaging. Franz 2013a is already a great reference to this topic, Köhli 2015 also touched this.**
We agreed and changed to:
Page 3 line 11: "are averaged over a larger area (Franz et al., 2013a; Kohli et al., 2015)."

**• L8: Bogena et al. 2013 did not perform simulations to the penetration depth. Instead, Franz et al. 2012 (doi:10.1029/2012WR01187) and Köhli et al. 2015 provided simulations that both support these values.**
We changed this:

Page 3 line 12-15: "Vertical measurement depth ranges from a maximum of ~70 cm under completely dry conditions and decreases to roughly ~12 cm under wet conditions (e.g. 40 vol. % soil moisture) (Kohli et al., 2015;Franz et al., 2012)."

**• L11: add a reference for COSMOS-UK, Evans et al. 2016, 10.1002/hyp.10929**
Thank you. We added:
Page 3 line 17: "and the British COSMOS-UK (Evans et al., 2016)."

**• L13-15: please rephrase to make clear what data assimilation is and is not.**
Thank you. We rephrased:
Page 2 line 28: "Data assimilation of soil moisture provides a way to improve imperfect land surface model predictions. Here, soil moisture measurements are used to update model predictions by optimally considering the uncertainty of model initial conditions, model parameters and model forcings. "

**• L15: It is not clear why you choose EnKF. Please at least mention other techniques and provide reasons for your choice. The sentence further should be moved to the end of the paragraph after you have introduced the history of DA.**
We added:
Page 3 line 23: "The EnKF is much less CPU intensive compared to alternative methods such as the particle filter (e.g. Montzka et al., 2011) because for high dimensional problems the EnKF requires a much smaller ensemble size to achieve reasonable good predictions. "
We considered moving the 'sentence further', but find it is located well at that spot.

**• L16-34: This historical overview appears to be unnecessary in the context of your study. Neither do you explain what things like "four-dimensional variational DA" are, nor is the relation to your work described. Furthermore, citations are used inflationary again. Please reduce this paragraph to the key publications which support your study. Also think about moving certain studies about ensemble size, multiple time steps, and other filtering approaches to the methodology section, where you need justification for your approach.**
Thank you. We shortened the literature review slightly, but include previous work on state updates and joint state-parameter updates.

**• L30-32: Just to emphasize the previous comment, these lines particularly carry no information for non-experts due to the lack of explanation.**
Thank you. We removed this section out.

**Page 4**
**• L1-19: As stated before, the whole literature review appears to be random and irrelevant to your work. Or at least the relations are not explained. For example, work from Montzka 2011;2013 and Han 2014b appear to be of some relevance for you, prior to others.**

The literature review puts this work into the context of soil moisture data assimilation and the joint state-parameter estimation. We may be too broad for you in this literature review but may not be too broad for other reviewers who desire a literature review.

**• L23-24: "Its capability to propagate surface soil moisture information into the deeper soil column was analyzed by Rosolem et al. (2014)", what does this sentence mean?**
We rephrase:
Page 5 line 21-25:"The surface soil moisture information was propagated into greater soil depth than only the measurement depth using COSMIC in combination with data assimilation (Rosolem et al., 2014)."

**• L26: "The COSMIC operator", third repetition as a sentence starter.**
We reformulated

**• L27-29: combine those sentences: "neutron observations have been used to update states (. . . ) and hydraulic parameters (. . . )"**
We considered to combine the sentences but it is better as is:
Page 5 line 24-28:"Neutron counts measured by CRNS have been used in data assimilation studies to update model states (Han et al., 2015;Rosolem et al., 2014). Soil hydraulic parameters were also updated by assimilation of neutron counts in one synthetic study (Han et al., 2016), showing its feasibility."

**• L29: "showed" → "demonstrated"**
We changed.

**• L29-30: be more correct in phrasing. Rephrase that Villarreyes 2014 used a different model, but also estimated hydr. parameters by inversion.**
We state:
Page 5 line 27: **"**CRNS were also used for inverse estimation of soil hydraulic parameters of the Hydrus-1D model (Villarreyes et al., 2014)."

**Han 2016 did so too, using support from neutron data, but neutron assimilation alone does not "update" a hydraulic parameter.**
We rephrase:
Page 5 line 24: "Soil hydraulic parameters were also updated by assimilation of neutron counts in one synthetic study (Han et al., 2016), showing its feasibility."

**• L31: "This work further explores", omit "further". Until now it is not clear what this work does, you only told stories about work of others. Please summarize which of the presented approaches you are picking up and what scientific novelty you add.**
"This work further…" links to the previous paragraph and the meaning of "further" means to expand the work that has been done by the community, not necessarily by the authors themselves. It is placed correctly in this context.

**Page 5**

• **L3-4: "the soil moisture characterization at the larger catchment scale", what exactly is meant by these terms, and how do you measure improvement?**

Thank you. We removed "larger" and hope this is clearer now.

• **L4: "how dense the CRP network should be", do you answer this question?**

Thank you. We removed this question, as this should be addressed in a synthetic study and is most probably catchment specific.

• **L7-8: "soil maps and atmospheric forcings show spatial correlations over larger distances", this is an interesting point, please provide reference. Isn't the largescale heterogeneity of soil maps only an artefact of soil data scarcity?**

These correlations are generated by large scale atmospheric processes, geomorphology, vegetation, and last but not least anthropogenic activities such as land use. We added three relevant references:

Page 6 line 6-9: "On the other hand, soil moisture, soil maps and atmospheric forcings show spatial correlations over larger distances (Kirkpatrick et al., 2014;Korres et al., 2015) which suggests that CRNS measurements potentially carry important information to update soil moisture contents for larger regions (e.g. Han et al., 2012)."

• **L9: "10 stations", do you assimilate all 10, or just 9?**

Thank you. We corrected this, it is nine stations.

• **L15: "feasibility of the updated large scale soil hydraulic parameters", how can a parameter be feasible? Please clarify your novel research question.**

We removed "of the feasibility".

• **L18-19: The sentences can be omitted as being obvious.**

We prefer to keep these sentences.

**Page 6**

• **L6: correct wording, a "process" can not be "solved"**

We rephrased:

Page 7 line 10: "Some of the key processes which are modelled by CLM are radiative transfer…"

• **L10: "Oleson et al. (2013) provide further details on CLM4.5", redundant information with regard to L5-6.**

Thank you. We removed this sentence.

• **L10-12: provide reasons why you artificially limit the scope and complexity of your study. What process would a "biogeochemical module" have added and why are they not important here compared to a prescribed LAI?**

This would be beyond the scope of this study as invoking the biogeochemical module would require a model spin-up of 1000 years for the catchment. It would have allowed to model vegetation development

dynamically and model the biomass development. It was however already explained elsewhere in this response letter that little additional gain is expected from this, which would not have balanced the great additional complexity introduced in the modelling process. We added:

Page 7 line 15-19: "To limit the scope and complexity of this study, CLM was run using satellite phenology e.g. prescribed leaf area index data and the biogeochemical module turned off. The biogeochemical module allows CLM to model the vegetation development dynamically, but it requires a large spin-up of 1000 years and little additional gain is expected for this study from these additionally modelled processes."

**• L14: please finally (after lots of references in the introduction) provide concrete information about the grid size in your study (the reader is still lost between centimeters and lightyears)**

In hydrologic literature it is common to provide the numbers for the model setup in the corresponding section. We added there:

Page 12line 10-11: "The model of the Rur catchment was spatially discretized by rectangular grid cells of 0.008 degree size (~750 m)."

**• L23: use standard format for functions,** $k[z] \rightarrow k(z)$

Thank you, changed

**• L24: format** $z \rightarrow z$

We modified.

**• L24: what is the difference between "soil moisture" and SWC? Why are you using the expression** $\theta$ **here, while SWC is used elsewhere (e.g., eqs. 24 and 25)?**

Thank you. Agreed. We changed the term SWC in Eqs. 24 and 25.

**• L25: use the more convenient expression** $k_{sat}(z)$**,**

We agree and modified it.

**• L26 (eq. 1):**

**– format** $k[z] \rightarrow k(z)$**,**

Agreed.

**– rewrite** $k_{sat,z} \rightarrow k_{sat}(z)$ **as this is a functional relationship. In contrast, indexing a state variable** $\theta_i$ **is ok.**

We agree and modified it.

**– omit occurrences of 0.5 since** $\frac{0.5}{0.5} \approx 1$**,**

Agreed.

**– case conditions (e.g.,** $1 < i < N \ldots$ **) are usually preceded by a comma in each line**

We added comma.

**– the curly bracket on the right is not common in multi-case equations.**

Thank you. We removed it.

**Page 7**

• **eq. 3 and 5: reformat** sand → sand**, same for clay.**
This was changed.

• **L6: "whereas", split sentence here.**
Thank you. We will split the sentence.

• **eqs. 9 and 10: this is a single equation, requiring only a single equation number, and a multi-case alignment using a curly bracket**
Thank you, this was modified.

• **L12: reformat** mm → mm**,**
This was corrected

• **whole page: please motivate the reader why these details are important for your research question. Also provide information where all these empirical (fixed) parameters (or regression coefficients) are coming from. Is the underlying theory so well understood that no uncertainties or further dependencies are required?**
We agree that this is important. We added motivation to this section:
Page 9 line 24-29:"The joint state-parameter estimation used in this study updates soil texture and organic matter in CLM. Hence, parameter estimates directly determine soil hydraulic properties in CLM. The following equations describe how soil texture and organic matter define the soil hydraulic properties in CLM such as porosity, hydraulic conductivity, the empirical exponent $B$ and soil matric potential."

**Page 8**
• **L4: "COSMIC parameterizes interactions". The interactions are parameterized by the underlying physical cross-section data. COSMIC rather parameterizes the neutron transport.**
We rephrased:
Page 9 line 9:"COSMIC parameterizes neutron transport within the soil subsurface and was calibrated against the more complex Monte Carlo Neutron Particle model MCNPx (Pelowitz, 2005)."

• **L7-8: Repetition from the introduction.**
We removed.

• **L10: "high energy neutrons are reduced" → "the number of high energy neutrons is reduced"**
We modified as suggested.

• **L11: "with less energy in each soil layer", misleading/unphysical. Fast neutrons typically evaporate with constant energy.**
We modified:
Page 9 line 14:" In the soil, the number of high energy neutrons is reduced by interactions within the soil leading to generation of fast neutrons in each soil layer."

• **L12: rewrite "soil interaction", as fast neutrons predominantly interact with the water.**

Agreed. However, bulk density plays a major role in parameterizing COSMIC.
We changed to:
"...interactions within the soil..."

**• L16-22 and eq. 14: this part can be omitted, since it is already well described in papers from Shuttleworth and Baatz, and does not add to the message of this paper. If you decide not to omit it, replace $\theta$ in eq. 14 to avoid confusion with soil moisture.**
Thank you. We replaced $\theta$ accordingly.

**• L22: explain to the reader how the 300 soil layers in COSMIC communicate with the 10 soil layers from CLM.**
We rephrased:
Page 10 line 13-14: "In this study, the contribution of each CLM soil layer was used to calculate the weighted CLM SWC retrieval corresponding to the vertical distribution of simulated SWC in each grid cell."
 And
Page 10 line 16-18: "Measured neutron intensity of CRNS was used to inversely determine a CRNS SWC retrieval as by Baatz et al. (2014) assuming a homogeneous vertical SWC distribution. Then, the weighted CLM SWC retrieval is used in the data assimilation scheme to relate the CRNS SWC retrieval to the model state."

Further explanation is given in the next section on data assimilation.

**• L26: what is a "COSMIC soil surface"?**
We rephrase: "the soil surface in COSMIC"

**• L25ff: it looks like you are not assimilating neutrons, but reiterating SWC from neutron data. The whole paragraph creates a great confusion about what the difference is between SWC, CLM SWC, weighted CLM SWC, and CRP SWC. In contrast to other less relevant paragraphs in this section, this part is highly unclear and simultaneously highly important to understand the most important part of your model. Please rephrase the whole paragraph and clarify to the reader what exactly you do, and why (i.e., why not assimilating N directly?)**
It is stated several times that SWC retrievals are assimilated updated and not neutron data e.g.:
Page 10 line 16-18: "Then, the weighted CLM SWC retrieval is used in the data assimilation scheme to relate the CRNS SWC retrieval to the model state."

The first sentence of this paragraph states what observation is assimilated:
Page 10 line 26-27:"To further expand the work of Han et al. (2016), this study uses the local ensemble transform Kalman filter (LETKF) (Hunt et al., 2007) to assimilate SWC retrievals by CRNS into the land surface model CLM. "

We then clearly state what is updated in CLM:

Page 10 line 28-29: "Updates were calculated either for SWC states or jointly for SWC states and soil parameters depending on the experiment setup."

We also added to the previous paragraph why SWC retrievals were assimilated, and not neutron flux:
Page 10 line 16-24: "Measured neutron intensity of CRNS was used to inversely determine a CRNS SWC retrieval as by Baatz et al. (2014) assuming a homogeneous vertical SWC distribution. Then, the weighted CLM SWC retrieval is used in the data assimilation scheme to relate the CRNS SWC retrieval to the model state. Alternatively, neutron flux data could be assimilated directly within the catchment. This would require calibration data throughout the catchment which is only feasible using spatially distributed data sets (e.g. Avery et al., 2016). However, high stands of biomass are a major factor for calibration in the Rur catchment (Baatz et al., 2015) and estimates of biomass come along with high uncertainties. To circumvent introducing these additional uncertainties, SWC retrievals are assimilated in this study. Changes in on-site biomass were assumed negligible."

**Page 9**
• **L5-10: Your paper is not a protocol. Again, it is described what you are using and who else used it, but the reader is left with the question why you (and others) made this decision. Shortly explain advantages of your strategy and why it serves your research question better than others.**
Thank you. We now state:
Page 10 line 26: "To further expand the work of Han et al. (2016), this study uses the local ensemble transform Kalman filter (LETKF) (Hunt et al., 2007) to assimilate SWC retrievals by CRNS into the land surface model CLM. "

We also argue:
Page 3 line 22-25: "The EnKF is much less CPU intensive compared to alternative methods such as the particle filter (e.g. Montzka et al., 2011) because for high dimensional problems the EnKF requires a much smaller ensemble size to achieve reasonable good predictions."

• **L11: what is $f$ in $\vec{x}^f$ ?**
We adde at page 11 line 3: "f marks the model prediction or forecast before the update"

• **L19: confusing typesetting. Is it $\overline{H}$ as a function of the COSMIC model, or is $\overline{H}$ identical with the COSMIC operator?**
We now state:
Page 11 line 11-13: "the observation operator **H** (COSMIC) is applied on the measured neutron intensity in order to obtain the expected weighted SWC retrieval at each of the observation locations for each of the stochastic realizations" to explain that H is neither identical nor a function.

• **eq. 19: do not use T as a symbol for transposition, there is a reserved symbol for this: $\vec{Y}^{\top}$.**
Thank you. We change T to the reserved symbol.

**Page 10**

**• L18: redundant sentence.**

The sentence was removed.

**• L20: why these values? is it comparable with the catchment-mean texture? If your question is, what impact a rough and uncertain soil map in data scarce region would have, wouldn't it be more reasonable to smooth out the existing soil map to a very rough degree, rather than using a completely arbitrary soil map?**

Thank you for your questions. We did calculations with the globally available but coarse FAO soil map for the revised version of the manuscript (see responses to reviewer #1).

We add information on catchment wide average sand and clay content, and what was the motivation to select the biased soil map as initial soil map in part of the simulation experiments:

Page 12 line 15-24: "Alternative simulations were also performed with the FAO soil map of the global Harmonized World Soil Database (FAO, 2012) and with a biased soil texture with a fixed sand content of 80 % and clay content of 10 % (S80 soil map). Average sand and clay content are 22.5% and 21.4% for the BK50 soil map and 39% and 22% for the FAO soil map. The FAO soil map and the biased soil map represent large error with respect to the soil properties of the BK50 soil map. The FAO soil map and S80 soil map simulations allow evaluating the joint state-parameter estimation approach because given the expected bias, we can evaluate to what extent the soil properties are modified by the data assimilation. This is important because in many regions across the Earth a high resolution soil map is not available. Land surface models are applied for those regions, for example in the context of global simulations, and hence might be strongly affected by the error in soil properties."

The new simulation results and discussion on the FAO soil map simulations were added.

**Page 11**

**• L3-5: omit physical units (they are irrelevant in this context).**

Omitted as suggested.

**• L8-11: How do you justify the perturbation of physical soil parameters like porosity and texture? Does the uncertainty of the soil map justify the huge variation ranges applied in this work? Are models allowed to adapt their physical basement to hydrological data (which also show uncertainty)?**

Perturbation of physical soil parameters like porosity and texture is a general standard procedure in data assimilation and numerous papers use this approach mentioned in the introduction of joint state-parameter estimation.

With the perturbation we account for uncertainty in these soil parameters. One of the main relevant sources of uncertainty in our modelling here is the uncertainty with respect to the soil parameters. The variation ranges are not so huge, considering that CLM assumes a single set of pedotransfer functions being valid throughout the globe. We added:

Page 13 line 12: "Soil texture perturbation considers that in CLM a single set of pedotransfer functions is assumed to be valid throughout the globe while usually pedotransfer functions are specific for regions

(e.g. Patil and Singh, 2016). In other words, the perturbation of soil texture also covers the uncertainty in the pedotransfer function itself."

**• L15: omit "="**
We agree and modifed.

**• How was the CRP SWC uncertainty determined? Assuming a constant CRNS error is not physical and might have substantial influence on the results (to be tested). For example, the error of neutron observations N is pN, while N can almost double from very wet to very dry conditions, which leads to a variation of the neutron uncertainty by 30%. This can propagate through the non-linear relation to soil moisture in such a way that your observed SWC is significantly more uncertain in wet periods compared to dry periods. Consequently, the DA approach should give more weight to dry periods during assimilation.**
This is a good point. The uncertainty of the neutron count intensity can be modelled by the Poisson distribution, and sampling this distribution and processing it through the non-linear equation which links neutron count intensity and weighted soil water content, allows deriving the measurement error in terms of SWC, also as function of SWC. This was not done here and we used results from earlier work that focused explicitly on estimating the measurement standard deviation of SWC. Based on this work we assume a fixed measurement standard deviation of 0.03 $cm^3/cm^3$ here. We added:
Page 13 line 24-27: "Based on previous work (Baatz et al., 2015), the SWC retrieval uncertainty for CRNS was estimated to be 0.03 $cm^3/cm^3$ while fluctuations in the measurement standard deviation, related to the non-linear relation between observed neutron intensity and SWC, were assumed negligible."

**Page 12**
**• L16: Why do you use RMSE, although many alternative measures are accepted as state-of-the-art measures for time series evaluation, e.g., KGE or NSE, in order to assess bias, deviation, and correlation simultaneously?**
RMSE is a standard measure used in the data assimilation community, whereas NSE is not standard. KGE is indeed an interesting measure which was however not considered here.

**• eqs. 23 and 24: reformat** SWC $\rightarrow$ SWC**, same with** RMSE **and** bias**, as those are single multi-letter variables, not products of multiple single-letter variables. Following this style guide, rewrite** $E_{RMS}$ $\rightarrow$ $E_{RMS}$**. You can even omit "**RMS**" since** E **is the only error used in this work. This would improve readability of the results section.**
Thank you for the suggestion. We reformatted the equation.

**Page 16**
**• L17-26: It is argued that changes in SWC states have impact to simulated ET flux. However, only for state-parameter updates (L19). Why is ET not affected by (SWC) state updates only?**
No, we did not argue this. The impact on ET was explored in less detail, partly because ET is most of the year close to potential ET, especially in the southern part of the catchment (hills). For illustration

purposes, we showed here the impact on ET by comparing open loop simulations and simulations with joint state-parameter updating.

**• L32: "to** $E_{RMS}$**-values", omit "-"**
Thank you, agreed.

**Page 17**
**• L9: if precipitation data from COSMO_DE was used, why was this information omitted in the method section (only mentioning DWD)?**
We added this now to the method section.

**• L26: replace "fast" with "quickly".**
Thank you. Done.

**Page 18**
**• L10-23: This question already needs an answer in the method section, I'd suggest to move the whole paragraph.**
We agree and moved the paragraph to the method section.

**• L10-23: I cannot follow the argumentation. Baatz et al. 2014 suggested a correction function for neutron counts based on vegetation estimates.**
Baatz et al. (2016) suggested such a correction function.

**In your model, you already have LAI data every month, implementation of the correction functions in the model would probably be straight forward. Furthermore, to convert neutron data to SWC, some vegetation correction would be necessary, too.**
LAI is not modelled dynamically (predicted) by CLM, but is a model forcing. In addition, LAI data from satellite are not as accurate (bias, scale mismatch) and the conversion from LAI to biomass is also affected by significant uncertainty. We do not expect additional gain from assimilating neutron count intensity instead of converted neutron count intensity.
The conversion of neutron data to SWC does not require vegetation correction per-se. If on-site calibration is done (which was the case for all nine locations) seasonal biomass changes are normally negligible. If biomass changes are not negligible (fast growing crops), at the sites more precise information on biomass is needed than at other catchment locations. Therefore the translation of neutron counts into SWC is reliable at the sites, and SWC can be interpolated dynamically afterwards with data assimilation. It is unclear what extra information can be gained if neutron count intensity is assimilated, given the less precise information on for example biomass at other locations in the catchment.

**Third, assimilating CRP SWC assumes homogeneous vertical SWC profiles (before iteration), this assumption would be unnecessary if neutrons would be assimilated directly. I am afraid that this topic**

**is more complex and needs further discussions and tests. It would be most convincing if you could show that neutron assimilation indeed gives different results than CRP SWC assimilation.**

This would be ideal but should be subject of another study. We clarify the paragraph, put it into the method section and rephrased:

Page 10 line 16: "Measured neutron intensity of CRNS was used to inversely determine a CRNS SWC retrieval as by Baatz et al. (2014) assuming a homogeneous vertical SWC distribution. Then, the weighted CLM SWC retrieval is used in the data assimilation scheme to relate the CRNS SWC retrieval to the model state. Alternatively, neutron flux data could be assimilated directly within the catchment. This would require calibration data throughout the catchment which is only feasible using spatially distributed data sets (e.g. Avery et al., 2016). However, high stands of biomass are a major factor for calibration in the Rur catchment (Baatz et al., 2015) and estimates of biomass come along with high uncertainties. To circumvent introducing these additional uncertainties, SWC retrievals are assimilated in this study. Changes in on-site biomass were assumed negligible."

**• L27: "neutron flux intensity", do you mean flux or intensity or both?**

Thank you. We removed intensity.

**• L27: "Although . . . only available at few locations", write more positively. Neutron data was available at up to 9 locations, which was intended to be the amazing novelty compared to other catchments!**

Thank you. We removed this part of the sentence.

**Figures**

**1. South → south, same with North.**

Thank you, we considered but according to Bryan A. Garner "The Oxford Dictionary of Usage and Style" (2000) Oxford University Press:

"The words north, south, east and west should not be capitalized when used to express directions <we went north>. They are properly capitalized when used as nouns denoting regions of the world or of a country <Far East> <the South>.

But when a directional word appears as an adjective before a geographic proper name, it is lower case <eastern United States> <southern Italy>. If, however, the adjective is part of the proper name, it should be capitalized <North Dakota> <East Anglia>."

In this study, the North of the catchment is a geographic region. Hence, we keep "the North" and "the South".

**2. Please add grid lines**

We added horizontal grid lines.

**3. Please add grid lines**

We added horizontal grid lines.

**4. Please add grid lines**

We added horizontal grid lines.

**5. It is hard to distinguish two black lines with different meaning. Further indication of the expected "true" sand content (given by the soil map or soil samples) would be helpful to evaluate these plots.**
We added a marker for the value of the BK50 soil map.

**6. This figure is not understandable without the text. Please shortly provide information about the B parameter in the caption to understand the message of this figure.**
We added information to the caption.

**7. replace** k(sat) → k$_{sat}$**. It would be interesting to also show the evolution of the soil porosity parameter together with an indication of its measured value. Why does hydraulic conductivity (and probably also porosity) vary over time at individual sites? Those are expected to be constant physical parameters of the sites. In my opinion this is a serious flaw of the DA approach used here.**
Thank you. We replaced ksat. We rephrase the figure caption to be more clear:
Page 32 line 3-6: "At nine sites, estimates of percentage sand content are shown for simulations with parameter update: PAR-S80-30 (green), PAR-S80-10 (light green), PAR-BK50-30 (red), PAR-BK50-10 (light red), jk8-S80-* (black) and jk8-BK50-* (black). The value of the BK50 soil map is marked at the second y-axis."
The period with parameter updates should be considered as calibration period where parameters are allowed to change. Notice also that simulations were made for a verification period with constant parameters (estimated during the assimilation period). We would like therefore to stress the importance of the evaluation period with temporally constant (estimated) parameter values.

**8. The purpose of this figure is not clear, as no observation data is provided to evaluate the model performance with respect to simulated latent heat.**
Thank you. We rephrased the caption:
Page 35 line 2-6: "Annual evapotranspiration (ET) is shown in the year 2013 (evaluation period, no assimilation). This figure demonstrates the impact of parameter updates (PAR-S80-10 and PAR-BK50-10) in comparison to open loop (OL-S80) and reference soil map (OL-BK50). ET changes in the North but not as much in the South."

**Tables**
**1. what is C3? Replace "non arctic" with "non-arctic", probably add a citation to the caption for plant functional types.**
C3 is a type of carbon fixation.
We added:
Page 36 line 1: "CLM plant functional type (Bonan et al., 2002)"
And removed "non" as there is not "arctic grass" in the catchment.

**3. improve readability by increasing font weight (boldness) for particularly good cases below an RMSE threshold, which is a common strategy in many journals.**
Thank you for the suggestion. We added:

Page 40 line 5: "The best cases are marked bold."

**4. same as 3. Rephrase the last sentence.**
Thank you. We rephrased.

**5. same as 4.**
Thank you. We rephrased.

Anonymous Referee #4

GENERAL COMMENTS

OVERALL QUALITY
**The paper by Baatz et al. (2016-432) describes an effort to use soil moisture data from nine closely-spaced (2000 km2) cosmic-ray probes (CRPs) with data assimilation scheme to improve the assessment of soil moisture in land-surface models. The goal is worthy and the execution is thorough. The results are significant: (1) the joint state (soil moisture) and parameter (soil properties, like sand percentage) estimation within data assimilation scheme produces better results than just state estimation; (2) in absence of soil data and meteorological data, CRPs alone can improve data assimilation results. On that account, the paper is suitable for publication in HESS.**

Thank you for the positive overall evaluation of the manuscript. We appreciate your effort in reviewing the manuscript. In the forthcoming revision we will consider each of your suggestions and expand the discussion to answer the questions raised.

**However, I am less certain about the significance of these results in light of the finding that the parameters change in time and in many cases never converge. This is only possible if the parameters are fitting parameters rather than physical parameters. So we end up with better results, but possibly only by statistical manipulation rather than by improved understanding of the physics. Is this progress? I would like to see at least some discussion of this issue in the paper in its final form.**

We will expand the discussion on this issue in the revised version of the manuscript. Regarding the change of parameters, those should be considered as parameter estimates. Notice also that simulations were made for a verification period with constant parameters (estimated in the assimilation period). Regarding uncertainty of parameters and the fact that those are not
We added possible reasons and discussion:
Page 19 line 26: "Temporally not stable parameter estimates imply that there may be multiple or seasonal optimal parameter values. This is also supported by the findings of the temporal behaviour of site average $E_{RMS}$ (**Error! Reference source not found.**) e.g. during the evaluation period when in the dry summer 2013 the $E_{RMS}$ peaks for the PAR-S80-30 simulation. In this context, it is important to mention that many possible error sources were not subject to calibration in this study but could be crucial for an even better modelled soil moisture and more reliable soil parameter estimation. In this study we only considered uncertainty of soil parameters, but also vegetation parameters are uncertain. Also a number of other CLM-specific hydrologic parameters (e.g. decay factor for subsurface runoff and maximum subsurface drainage) strongly influence state variables in CLM and hence show potential for optimization (Sun et al., 2013). Considering this uncertainty from multiple parameters could give a better parameter uncertainty characterization (Shi et al., 2014). Precipitation is also an important forcing for hydrologic modelling. For this study, precipitation data from the COSMO_DE re-analysis were used. A product which optimally combines precipitation estimates from radar and gauge measurements is expected to give better precipitation estimates than the reanalysis. This could improve the soil moisture characterization and also potentially lead to better parameter estimates. Further improvements and constraining of parameter uncertainty is also possible using multivariate data assimilation with observations such as latent heat flux (e.g. Shi et al., 2014). Also other error sources related to the model structure play a significant role. These options should be subject of future investigations. "

**SPECIFIC COMMENTS**
**Why are RMSE and bias discussed separately if they are essentially the same information? One is computed on squares of differences and therefore has a positive sign; the other is computed on differences and therefore has a sign. Wouldn't the bias suffice? If you keep both, please explain why they are both needed and how they are different.**

We add a motivation for using bias as second error measurement:
Page 15 line 4-5: "The second evaluation measurement in this study is the bias which is, in contrast to the $E_{RMS}$, a measure for systematic deviation:"

**How are the results evaluated? What is the gold standard for soil properties? Pedotransfer functions? What is it for soil moisture? At some of the sites extensive networks of TDR probes exists. Would it be possible to include TDR data in the evaluation?**
Thank you for the suggestions.
TDR probes work on a scale of few $dm^3$ which is significantly smaller than the scale of the land surface model (1 $km^2$). Hence, TDR probes are not suitable for a direct evaluation whereas cosmic-ray neutron sensors (CRNS) measure soil moisture at an equivalent scale.

**While TDRs are hardly the gold standard in soil moisture measurements, they would provide independent soil moisture data. By the same token, have soil properties been measured at some of the sites? Using the pedotransfer functions to derive unsaturated hydraulic conductivities is hardly the gold standard, and hard to defend.**
Using pedotransfer functions is common sense in land surface modeling such as the land surface model used in this study (CLM) or the Noah land surface model. Those models need to be calibrated by some sort of measurements which are scale consistent data measured by cosmic-ray neutron sensors in this study.

**I suggest that the authors make at least some effort to provide independent data on soil moisture and hydraulic properties.**
We note that soil moisture measurements by CRNS were already independent from the prediction of the land surface model during the verification period. Soil hydraulic properties at the desired scale of the land surface model were not measured.
We also acknowledge that measurements on evapotranspiration would be desirable, but those are not in the scope of this paper which focuses on soil moisture predictions:
Page 21 line 2-6: "Hence particularly in the northern part of the catchment, further observations such as ET measurements are desirable for further improving the land surface model. These additional observations could be used for future land surface model benchmarking (Best et al., 2015) or for more constrained parameter estimates (Shi et al., 2015)."

**TECHNICAL CORRECTIONS**
**Please, see the annotated pdf manuscript.**

For technical corrections we refer to the annotated pdf manuscript.

[revised manuscript text omitted]

**Evaluating the valueEvaluation of a network of cosmic-ray probes neutron sensor network for improvingimproved land surface modellingmodel prediction**

Roland Baatz[1,2], Harrie-Jan Hendricks Franssen[1,2], Xujun Han[1,2], Tim Hoar[3], Heye R. Reemt Bogena[1] and Harry Vereecken[1,2]

[1]Agrosphere (IBG-3), Forschungszentrum Jülich GmbH, 52425 Jülich, Germany.

[2]HPSC-TerrSys , 52425 Jülich, Germany.

[3]NCAR Data Assimilation Research Section, Boulder, CO, USA.

*Correspondence to*: Roland Baatz (r.baatz@fz-juelich.de)

**Abstract:**  In-situ soil moisture sensors provide highly accurate but  very local soil moisture measurements while remotely sensed soil moisture is strongly affected by vegetation and surface roughness. In contrast, Cosmic-Ray Neutron Sensors (CRNS) allow highly accurate soil moisture estimation at the field scale which could be valuable to improve land surface model predictions. In this study, the potential of a network of CRNS installed in the 2354 km$^2$ Rur catchment (Germany) for estimating soil hydraulic parameters and improving soil moisture states was tested . Data measured by the CRNS were assimilated with the local ensemble transform Kalman filter in the Community Land Model  v. 4.5. Data of four, eight and nine CRNS were assimilated for the years 2011 and 2012 (with and without soil hydraulic parameter estimation), followed by a verification year 2013 without data assimilation. This was done using (i) a regional high resolution soil map, (ii) the FAO soil map and (iii) an erroneous, biased soil map as input information for the simulations. For the regional soil map, soil moisture characterization was only improved in the assimilation period but not in the verification period. For the biased soil map soil moisture predictions improved strongly to a $E_{RMS}$root mean square error of 0.03 cm$^3$/cm$^3$ assimilation period and 0.05 cm$^3$/cm$^3$ for the evaluation period. Improvements were limited by the measurement error of CRNS (0.03 cm$^3$/cm$^3$). The positive results obtained with data assimilation of nine CRNS were confirmed by the jackknife experiments with four and eight CRNS used for assimilation. The results demonstrate that assimilated data of a CRNS network can improve the characterization of soil moisture content at the catchment scale by updating spatially distributed soil hydraulic parameters a land surface model.

**1    Introduction**

Soil water content (SWC) is a key variable of land surface hydrology and has a strong control on the partitioning of net radiation between latent and sensible heat flux. (Brutsaert, 2005). Knowledge of SWC is relevant for the assessment of plant water stress and agricultural production, as well as runoff generation as a response to precipitation events (Vereecken et al., 2008;Robinson et al., 2008). In atmospheric circulation models, SWC is important as a lower boundary condition andwhile it is calculated as a state variable in land surface models. Coupling of atmospheric circulation models and land surface models allows quantifying the role of soil moisture on atmospheric processes such as soil moisture-precipitation feedbacks (Koster et al., 2004;Eltahir, 1998) and summer climate variability and drought (Seneviratne et al., 2006;Oglesby and Erickson, 1989;Sheffield and Wood, 2008;Seneviratne et al., 2006;Bell et al., 2015). It is therefore important to improve the modelling and prediction of SWC, but this is hampered by. Data assimilation of soil moisture provides a way to improve imperfect land surface model deficiencies andpredictions. Here, soil moisture measurements are used to update model predictions by optimally considering the uncertainty of model initial conditions, model parameters and model forcings. However, there is a lack of high quality soil moisture data (Vereecken et al., 2016). Soil moisture measured by space-bornborne remote sensing technologies provides information over large areas (e.g. Temimi et al., 2014). However, space born remote sensing supplies only information on the upper few centimeters,but is strongly affected by vegetation, and data are not reliable for areas with dense vegetation.surface roughness (e.g. Temimi et al., 2014). Therefore, in this paper an alternative source for soil moisture information is explored. which can measure soil moisture more accurately under dense vegetation (Bogena et al., 2013). Cosmic-ray probes (CRPsneutron sensors (CRNS) measure fast neutron intensity which allows estimating SWC at an intermediate scale of ~15 ha (Kohli et al., 2015;Zreda et al., 2008;Desilets et al., 2010;Cosh et al., 2016;Lv et al., 2014) which is closer to the desired application scale of land surface models (Ajami et al., 2014;Chen et al., 2007;Shrestha et al., 2014). Fast neutrons originate from moderationcollisions of secondary cosmic particles from outer space bywith terrestrial atoms. These particles are mainly fast Fast neutrons, which 
[revised manuscript text omitted]

Merged Cells
Merged Cells
Merged Cells
Merged Cells
Merged Cells
Merged Cells
Merged Cells
Merged Cells
Merged Cells
Merged Cells
Inserted Cells
Inserted Cells
Deleted Cells
Inserted Cells
Inserted Cells
Deleted Cells
Deleted Cells
Inserted Cells
Inserted Cells

**Annex**

Annex 1: $E_{RMS}$ (cm$^3$/cm$^3$) at CRNS  sites for open loop runs and different data assimilation scenarios, for the assimilation period (2011 and 2012). For jackknife experiments (21 in total) only the error of the omitted sites is reported. The best cases are marked bold.

| | 2011 & 2012 | Rolles-broich | Merzen-hausen | Geve-nich | Heins-berg | Kall | RurAue | Wueste-bach | | | |
|---|---|---|---|---|---|---|---|---|---|---|---|
| **Soil map** | | | | | | | | | | | |
| **BK50** | OL-BK50 | 0.058 | 0.060 | 0.039 | 0.039 | 0.046 | 0.034 | 0.056 | | | |
| - | Stt-BK50 | 0.031 | 0.039 | 0.021 | 0.021 | 0.030 | 0.024 | 0.039 | | | |
| - | PAR-BK50-10 | 0.033 | 0.036 | 0.020 | 0.019 | 0.032 \|  | 0.025 | 0.035 | | | |
| - | PAR-BK50-30 | 0.030 | 0.032 | 0.018 | 0.018 | 0.028 | 0.024 | 0.040 | | | |
| - | jk8-BK50--1 to 9 | 0.067 | 0.056 | 0.065 | 0.033 | 0.047 | 0.051 | 0.062 | | | |
| **FAO** | OL-FAO | 0.097 | 0.033 | 0.029 | 0.056 | 0.082 | 0.096 | 0.079 | | | |
| - | PAR-FAO-30 | 0.029 | 0.033 | 0.018 | 0.019 | 0.028 | 0.025 | 0.042 | | | |
| **Biased (S80)** | OL-S80 | 0.169 | 0.054 | 0.082 | 0.119 | 0.152 | 0.161 | 0.110 | | | |
| - | Stt-S80 | 0.098 | 0.019 | 0.036 | 0.050 | 0.082 | 0.054 | 0.083 \|  0.060 | | | |
| - | PAR-S80-10 | 0.031 | 0.035 | 0.023 | 0.023 | 0.033 | 0.024 | 0.041 | | | |
| - | PAR-S80-30 | 0.029 | 0.032 | 0.018 | 0.019 | 0.028 | 0.024 | 0.042 | | | |
| - | jk8-S80--1 to 9 | 0.081 | 0.038 | 0.060 | 0.035 | 0.068 | 0.043 | 0.057 | | | |
| - | jk4-S80-A | 0.064 | 0.038 | 0.059 | 0.076 | - | 0.157 | - | - | - | 0.079 |
| - | jk4-S80-B | 0.077 | 0.041 | - | 0.051 | 0.062 | 0.079 | - | - | - | 0.062 |
| - | jk4-S80-C | - | 0.073 | 0.056 | - | 0.051 | - | - | 0.078 | 0.109 | 0.073 |

Merged Cells
Merged Cells
Merged Cells
Merged Cells
Merged Cells
Merged Cells
Merged Cells
Merged Cells
Merged Cells
Inserted Cells
Inserted Cells
Inserted Cells

Annex 2: $E_{RMS}$ (cm$^3$/cm$^3$) at CRNS-sites for open loop, data assimilation and jackknife simulations on the basis of a comparison with CRNS SWC retrievals for the verification period (2013). For jackknife experiments (21 in total) only the error of the omitted sites is reported. The best cases are marked bold.

| Soil map | 2013 | Rolles-broich | Merzen-hausen | Geve-nich | Heins-berg | Kall | RurAue | Wueste-bach | Aachen | Wilden-rath | Average $E_{RMS}$ |
|---|---|---|---|---|---|---|---|---|---|---|---|
| BK50 | OL-BK50 | 0.04404 | 0.06507 | 0.03604 | 0.02703 | 0.04805 | 0.03804 | 0.04805 | 0.04204 | 0.01702 | **0.04104** |
|  | Stt-BK50 | 0.04104 | 0.05405 | 0.03403 | 0.02703 | 0.04905 | 0.03804 | 0.04805 | 0.04104 | 0.01802 | **0.039** |
|  | PAR-BK50-10 | 0.06807 | 0.06206 | 0.03604 | 0.03804 | 0.05606 | 0.05606 | 0.04304 | 0.05806 | 0.01702 | 0.048 |
|  | PAR-BK50-30 | 0.05205 | 0.06106 | 0.03504 | 0.03303 | 0.06807 | 0.04805 | 0.04304 | 0.04805 | 0.03504 | 0.047 |
|  | jk8-BK50-*-1 to 9 | 0.03604 | 0.04705 | 0.02804 | 0.02503 | 0.04205 | 0.03104 | 0.04005 | 0.05406 | 0.10611 | 0.04505 |
| FAO | OL-FAO | 0.08 | 0.04 | 0.04 | 0.05 | 0.09 | 0.09 | 0.07 | 0.09 | 0.07 | 0.068 |
|  | PAR-FAO-30 | 0.06 | 0.06 | 0.04 | 0.04 | 0.06 | 0.03 | 0.05 | 0.07 | 0.04 | **0.049** |
| Biased (S80) | OL-S80 | 0.15716 | 0.06206 | 0.10611 | 0.11512 | 0.16016 | 0.15415 | 0.09910 | 0.16717 | 0.01902 | 0.115 |
|  | Stt-S80 | 0.10010 | 0.06306 | 0.10711 | 0.10611 | 0.09910 | 0.14615 | 0.09710 | 0.15816 | 0.02002 | 0.100 |
|  | PAR-S80-10 | 0.06006 | 0.03904 | 0.04304 | 0.04004 | 0.06406 | 0.04304 | 0.05205 | 0.06006 | 0.01902 | **0.047** |
|  | PAR-S80-30 | 0.04905 | 0.05906 | 0.03704 | 0.03604 | 0.05305 | 0.03203 | 0.04605 | 0.04705 | 0.03504 | **0.044** |
|  | jk8-S80-*-1 to 9 | 0.07908 | 0.04605 | 0.04204 | 0.03604 | 0.059 | 0.03804 | 0.06306 | 0.04404 | 0.10511 | 0.057 |
|  | jk4-S80-A | 0.05 | 0.03 | 0.05 | 0.04 | - | 0.16 | - | - | - | 0.065 |
|  | jk4-S80-B | 0.05 | 0.07 | - | 0.04 | 0.07 | 0.07 | - | - | - | 0.061 |
|  | jk4-S80-C | - | 0.05 | 0.04 | - | 0.07 | - | - | 0.06 | 0.13 | 0.069 |

Table

Merged Cells ...
Merged Cells ...
Merged Cells ...
Merged Cells ...
Merged Cells ...
Merged Cells ...
Merged Cells ...
Merged Cells ...
Merged Cells ...
Merged Cells ...
Inserted Cells ...
Inserted Cells ...
Deleted Cells ...
Deleted Cells ...
Inserted Cells ...
Inserted Cells ...
Deleted Cells ...
Deleted Cells ...
Deleted Cells ...
Inserted Cells ...
Inserted Cells ...
Inserted Cells ...
Inserted Cells ...
Inserted Cells ...
Deleted Cells ...
Deleted Cells ...
Deleted Cells ...
Inserted Cells ...
Inserted Cells ...
Inserted Cells ...
Deleted Cells ...
Deleted Cells ...
Inserted Cells ...

Annex 53: Bias (cm$^3$/cm$^3$) at CRPCRNS-sites for open loop, data assimilation and jackknife simulations compared to CRPCRNS SWC retrievals duringfor the verificationdata assimilation period (20132011 and 2012). For each jackknife simulationexperiments (21 in total) only onethe bias of the omitted sites is reported:. The bias of the location that is meant for evaluationbest cases are marked bold.

| Year 2013 Soil map | 2011 & 2012 | Rolles-broich | Merzen-hausen | Geve-nich | Heins-berg | Kall | Rur... Aue... |
|---|---|---|---|---|---|---|---|
| BK50 | OL-BK50 | -0.0305 | 0.0605 | 0.0102 | -0.0001 | -0.02 | |
| | Stt-BK50 | -0.0102 | 0.0403 | 0.00 | -0.0001 | -0.01 | |
| | PAR-BK50-10 | 0.0600 | 0.0503 | 0.0100 | 0.0200 | 0.0400 | |
| | PAR-BK50-30 | -0.02 0.03 0.05 | | 0.00 | -0.0201 | -0.0401 0.03 | |
| | jk8-BK50-1 to 9 | -0.05 | 0.05 | 0.05 | -0.01 | 0.00 | |
| FAO jk-BK50 * | OL-FAO | -0.09 | 0.02 | -0.01 | -0.05 | -0.07 | |
| | -0.02PAR-FAO-30 | -0.0401 0.03 | | 0.00 | -0.01 | -0.01 | -0.0... |
| Biased (S80) | OL-S80 | -0.17 | -0.05 | -0.08 | -0.12 | -0.15 | |
| | Stt-S80 | -0.09 | -0.0500 | -0.1003 | -0.1004 | -0.0807 | |
| | PAR-S80-10 | 0.00 | 0.03 | 0.00 | -0.01 | 0.00 | |
| | PAR-S80-1030 | 0.03 0.05 -0.01 -0.02 0.03 | | 0.0100 | -0.01 | -0.01 | |
| | PARjk8-S80-301 to 9 | -0.0407 0.02 -0.03 0.02 | | 0.05 | -0.01 | -0.06 | |

| | | | | | | |
|---|---|---|---|---|---|---|
| jk4-S80-A | -0.05 | -0.02 | -0.03 | -0.06 | - | -0.16 | - |
| jk4-S80-B | -0.07 | 0.02 | - | -0.04 | -0.05 | -0.07 | - |
| jkjk4-S80-*-C | -0.07 | 0.0304 | 0.02 | 0.02- 0.04 | -0.02 | | |

Deleted Cells

Inserted Cells

Annex 4: Bias (cm$^3$/cm$^3$) at CRNS-sites for open loop, data assimilation and jackknife simulations compared to CRNS SWC retrievals for the data assimilation period (2011 and 2012). For jackknife experiments (21 in total) only the bias of the omitted sites is reported. The best cases are marked bold.

| Soil map | 2013 | Rolles-broich | Merzen-hausen | Geve-nich | Heins-berg | Kall | RurAue | Wueste-bach | Aachen | Wilden-rath | Mean absolute bias |
|---|---|---|---|---|---|---|---|---|---|---|---|
| BK50 | OL-BK50 | -0.03 | 0.06 | 0.01 | 0.00 | -0.02 | 0.00 | -0.02 | 0.01 | 0.00 | **0.02** |
| | Stt-BK50 | -0.01 | 0.04 | 0.00 | 0.00 | -0.01 | -0.01 | -0.02 | 0.00 | -0.01 | **0.01** |
| | PAR-BK50-10 | 0.06 | 0.05 | 0.01 | 0.02 | 0.04 | 0.04 | 0.02 | -0.04 | 0.00 | 0.03 |
| | PAR-BK50-30 | 0.03 | 0.05 | 0.00 | 0.02 | 0.04 | 0.03 | -0.01 | -0.03 | 0.03 | 0.03 |
| | jk8-BK50-1 to 9 | -0.02 | 0.04 | 0.01 | -0.01 | -0.03 | -0.02 | -0.04 | -0.05 | 0.11 | 0.04 |
| FAO | OL-FAO | -0.08 | 0.02 | -0.02 | -0.04 | -0.08 | -0.08 | -0.05 | -0.08 | 0.06 | 0.06 |
| | PAR-FAO-30 | 0.04 | 0.05 | 0.00 | 0.02 | 0.03 | 0.00 | -0.02 | -0.06 | 0.03 | **0.03** |
| Biased (S80) | OL-S80 | -0.15 | -0.05 | -0.10 | -0.11 | -0.16 | -0.15 | -0.09 | -0.16 | -0.01 | 0.11 |
| | Stt-S80 | -0.09 | -0.05 | -0.10 | -0.10 | -0.08 | -0.14 | -0.09 | -0.15 | -0.01 | 0.09 |
| | PAR-S80-10 | 0.04 | 0.03 | -0.03 | 0.03 | 0.05 | 0.03 | 0.03 | -0.04 | -0.01 | **0.03** |
| | PAR-S80-30 | 0.03 | 0.05 | -0.01 | 0.02 | 0.03 | 0.01 | -0.01 | -0.03 | 0.03 | **0.02** |
| | jk8-S80-1 to 9 | -0.07 | 0.03 | 0.02 | 0.02 | -0.05 | -0.02 | -0.04 | -0.03 | 0.10 | 0.04 |
| | jk4-S80-A | 0.00 | 0.01 | 0.03 | -0.03 | - | -0.15 | - | - | - | 0.04 |
| | jk4-S80-B | -0.03 | 0.06 | - | -0.03 | -0.06 | -0.06 | - | - | - | 0.05 |
| | jk4-S80-C | - | 0.04 | 0.02 | - | -0.05 | - | - | -0.05 | 0.13 | 0.06 |